# FuseMoE: Mixture-of-Experts Transformers for Fleximodal Fusion

**Xing Han**

Department of Computer Science

Johns Hopkins University

xhan56@jhu.edu

**Huy Nguyen**[*]

Department of Statistics and Data Sciences

The University of Texas at Austin

huynm@utexas.edu

**Carl Harris**[*]

Department of Biomedical Engineering

Johns Hopkins University

charr165@jhu.edu

**Nhat Ho**[+]

Department of Statistics and Data Sciences

The University of Texas at Austin

minhnhat@utexas.edu

**Suchi Saria**[+]

Department of Computer Science

Johns Hopkins University

ssaria@cs.jhu.edu

## Abstract

As machine learning models in critical fields increasingly grapple with multimodal data, they face the dual challenges of handling a wide array of modalities, often incomplete due to missing elements, and the temporal irregularity and sparsity of collected samples. Successfully leveraging this complex data, while overcoming the scarcity of high-quality training samples, is key to improving these models' predictive performance. We introduce "FuseMoE", a mixture-of-experts framework incorporated with an innovative gating function. Designed to integrate a diverse number of modalities, FuseMoE is effective in managing scenarios with missing modalities and irregularly sampled data trajectories. Theoretically, our unique gating function contributes to enhanced convergence rates, leading to better performance in multiple downstream tasks. The practical utility of FuseMoE in the real world is validated by a diverse set of challenging prediction tasks.

## 1 Introduction

Multimodal fusion is a critical and extensively studied problem in many significant domains [78, 94, 90, 9], such as sentiment analysis [26, 54], image and video captioning [42, 41], and medical prediction [33, 83]. Previous research has shown that embracing multimodality can improve predictive performance by capturing complementary information across modalities, outperforming single-modality approaches in similar tasks [71, 32]. However, an ongoing challenge lies in the creation of scalable frameworks for fusing multimodal data under a variety of conditions, and in creating reliable models that consistently surpass their single-modal counterparts.

---

[*]Equal Contribution, [+]Equal Advising.

38th Conference on Neural Information Processing Systems (NeurIPS 2024).

Table 1: We evaluated the characteristics of `FuseMoE` against various benchmarks. The pipeline approach [83] relies on a simple feature extraction scheme for each modality, followed by concatenation and classification. It doesn't incorporate irregularities or missingness in its process, but its use of concatenation and zero-imputation for missing modalities allows it to be adapted to FlexiModal settings. Both [101] and [96] tackle multi-modality fusion, but as modalities increase, their method demands exponentially more cross-modal computations and significant model architecture modifications. Finally, [58] presents MoE for language-image alignment, yet it also requires substantial adjustments for the more intricate and universal FlexiModal context we explore.

| Method | Type | Irregularity | Missingness | Num of Mods | Theory | FlexiModal Adaptive? |
|---|---|---|---|---|---|---|
| Soenksen et al. [83] | Data Pipeline | ✗ | ✗ | ≥4 | ✗ | ✓ |
| Zhang et al. [101] | Modality Fusion | ✓ | ✗ | 2 | ✗ | ✗ |
| Zadeh et al. [96] | Modality Fusion | ✗ | ✗ | 3 | ✗ | ✗ |
| Mustafa et al. [58] | Multimodal MoE | ✗ | ✗ | 2 | ✗ | ✗ |
| FuseMoE | This Paper | ✓ | ✓ | ≥4 | ✓ | Adapted |

Handling a variable number of input modalities remains an open challenge in multimodal fusion, due to challenges with scalability and lack of unified approaches for addressing missing modalities. Many existing multimodal fusion methods are designed for only two modalities [25, 102, 101], rely on costly pairwise comparisons between modalities [90], or employ simple concatenation approaches [83], rendering them unable to scale to settings with a large number of input modalities or adequately capture inter-modal interactions. Similarly, existing works are either unable to handle missing modalities entirely [101, 98] or use imputation approaches [89, 51, 83] of varying sophistication. The former methods restrict usage to cases where all modalities are completely observed, significantly diminishing their utility in settings where this is often not the case (such as in clinical applications); the latter can lead to suboptimal performance due to the inherent limitations of imputed data. In addition, the complex and irregular temporal dynamics present in multimodal data have often been overlooked [101, 88], with existing methods often ignoring irregularity entirely [83] or relying on positional embedding schemes [90] that may not be appropriate when modalities display a varying degree of temporal irregularity. Consequently, there is a pressing need for more advanced and scalable multimodal fusion techniques that can efficiently handle a broader set of modalities, effectively manage missing and irregular data, and capture the nuanced inter-modal relationships necessary for robust and accurate prediction. We use the term **FlexiModal Data** to capture several of these key aspects, which haven't been well-addressed by prior works:

> *"Flexi" suggests flexibility, indicating the possibility of having any combination of modalities, even with arbitrary missingness or irregularity.*

FlexiModal data is most evident in clinical scenarios, where extensive monitoring results in the accumulation of comprehensive electronic health records (EHRs) for each patient. A typical EHR encompasses diverse data types, including tabular (e.g., age, demographics, gender), images (X-rays, magnetic resonance imaging, and photographs), clinical notes, physiological time series (ECG and EEG), and vital signs (blood chemistry, heart rate). In this setting, we observe a variety of modalities, sampled with varying irregularity and a high degree of missingness and sparsity.

**Contributions** In this paper, we introduce a novel mixture-of-experts (MoE) framework, which we call `FuseMoE`, specifically designed to enhance the multimodal fusion of FlexiModal data. `FuseMoE` incorporates sparsely gated MoE layers in its fusion component, which are adept at managing distinct tasks and learning optimal modality partitioning. In addition, `FuseMoE` surpasses previous cross-attention-based methods in scalability, accommodating an unlimited array of input modalities. Furthermore, `FuseMoE` routes each modality to designated experts that specialize in those specific data types. This allows `FuseMoE` to effectively handle scenarios with missing modalities by dynamically adjusting the influence of experts primarily responsible for the absent data, while still utilizing the available modalities. Lastly, `FuseMoE` integrates a novel Laplace gating function, which is theoretically proven to ensure better convergence rates compared to traditional Softmax functions, thereby enhancing predictive performance. We have conducted comprehensive empirical evaluations of `FuseMoE` across a range of application scenarios to validate its effectiveness.

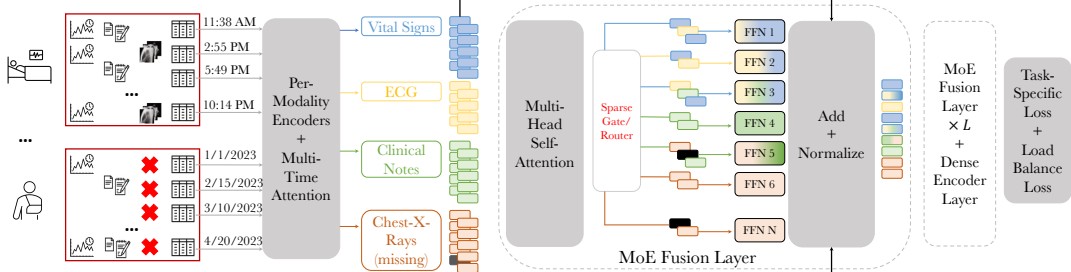

Figure 1: An example of addressing the challenge of FlexiModal Data: patients in ICUs often have extensive and irregular health status measurements over time; patients with milder conditions only require monitoring across fewer categories. `FuseMoE` is adept at handling inputs featuring any combination of modalities, including those with missing elements. It starts by encoding inputs using modality-specific feature extractors, followed by employing a multi-time attention mechanism [82] to address temporal irregularities. The core of `FuseMoE` lies the MoE Fusion Layer, where a routing mechanism is trained to categorize multimodal inputs and direct them to the appropriate combinations of MLPs. The outputs from these MLPs are weighted through a gating function, resulting in fused embeddings, which are subsequently utilized for further processing.

## 2 FuseMoE: Enhance Predictive Performance for FlexiModal Data

In this section, we delve into the fundamental components of `FuseMoE`, illustrated in Figure 1. We focus on two critical elements: the modality and irregularity encoder, and the MoE fusion layer.

### 2.1 Sparse MoE Backbone

The main components of a sparse MoE layer are a network $G$ as a sparse gate and an expert network $E$. [79] proposed a Top-$K$ gating function that takes as an input a token representation $x \in \mathbb{R}^D$ and then routes it to the Top-$K$ experts out of the set $\{E_i\}_{i=1}^S$. The gating network parameter $W \in \mathbb{R}^{D \times N}$ produces logits $h_s(x) = \text{Top K}(x \cdot W)$, which are normalized via Softmax:

$$G(x)_i = \frac{\exp(h_s(x)_i)}{\sum_j^K \exp(h_s(x)_j)}. \tag{1}$$

Each expert network ($E_i : \mathbb{R}^D \to \mathbb{R}^D$) contains a feed-forward layer (FFN) and its parameters are independent of other models. The final output of the expert network $y$ is the linearly weighted combination of each expert's output on the token by the gate's output: $y = \sum_{i=1}^S G(x)_i E_i(x)$.

**Gating Network Design** The gating network's advantage lies in its capacity to be concurrently trained with FFNs, facilitating the learning of an optimal sparse combination of experts. Essentially, by evaluating the similarity between the input token and the experts, the gating network/router optimally matches the input partition with the most suitable experts. In many cases, variations in the routing mechanism can greatly influence performance across diverse applications [48]. The Softmax gating is the most widely adopted across domains [77, 79]. We introduce a novel Laplace gating function that offers enhanced convergence guarantees and delivers superior predictive performance, particularly in FlexiModal applications. The function is formulated as follows:

$$h_l(x) = \text{Top K}(-\|W - x\|_2). \tag{2}$$

The Laplace gating function, characterized by its Euclidean term $\exp(-\|W - x\|_2)$, is less prone to converge towards extreme weight distributions due to the bounded nature of this term. In subsequent sections, we will illustrate how this gating function facilitates faster parameter estimation rates compared to Softmax gating. Moreover, our empirical findings indicate that the Laplace gating exhibits enhanced performance in managing FlexiModal data.

### 2.2 Modality and Irregularity Encoder

To encode the irregularity of sampling in each modality, we utilize a discretized multi-time attention (mTAND) module [82], which leverages a time attention mechanism [43, 92] to discretize irregularly sampled observations into discrete intervals. Specifically, given a set of $l_k$ continuous time points,

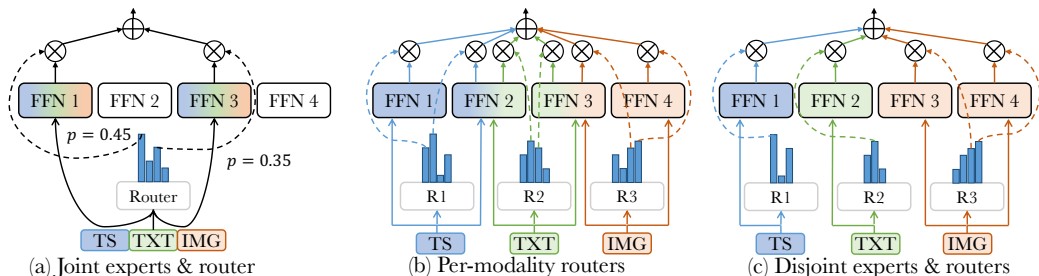

(a) Joint experts & router      (b) Per-modality routers      (c) Disjoint experts & routers

Figure 2: We present three exemplary designs of the Top-$K$ router for effective multimodal fusion, considering an input scenario with three modalities: Time-Series (TS), Text (TXT), and images (IMG). (a) The joint router design utilizes a concatenated embedding of all modalities, directing this combined input to selected experts. (b) In the modality-specific router design, each modality's embedding is independently assigned to a shared pool of experts. (c) The third design variant also uses modality-specific routers but assigns each modality's embedding to separate pools of experts, each pool uniquely tailored to process a specific modality type.

$t \in \mathbb{R}^{l_k}$, corresponding to the $k^{\text{th}}$ dimensionality of a given modality, we employ $H$ embedding functions $\phi_h(\tau)$ to embed each $\tau_k \in t_k$ in a $d_h$ dimensional vector space (detailed definition and examples can be found in Appendix B and E). The $i^{\text{th}}$ dimension of the $h^{\text{th}}$ embedding is defined as

$$\phi_h(\tau)[i] = \begin{cases} w_i \tau_k, & \text{if } i = 1 \\ \sin(w_i \tau_k + \phi_i), & \text{if } 1 < i \le d_h, \end{cases}$$

where $\{w_i, \phi_i\}_{i=1}^{d_h}$ are learnable parameters. By performing this for each continuous time point in $t_k$, we create a $d_h$ dimensional representation of each time point in $H$ different embedding spaces. We then leverage these embeddings to discretize the irregularly sampled observations into discretized bins. Specifically, we seek to discretize $x_k$ (with $l_k$ corresponding observation times $t_k$) into $\gamma$ regularly sampled intervals $\gamma$. We do this via an attention mechanism, which, for each embedding function $\phi_h(\tau)$, takes $\gamma$ as queries, $t_k$ as keys, and $x_k$ as values and produces $\hat{x}_{k,h} \in \mathbb{R}^\gamma$ embeddings for each sequence. Formally,

$$\hat{x}_{k,h} = \text{Softmax}\left(\frac{\phi_h(\gamma)\mathbf{Q}_h \mathbf{K}_h^\top \phi_h(t_k)^\top}{\sqrt{l_k}}\right) x_k,$$

where $\mathbf{Q}_h$ and $\mathbf{K}_h$ are learnable parameters. This formulation allows us to discretize univariate observations $x_k$ into $\gamma$ regularly-sampled bins. To model irregularity across a multivariate set of observations for a given modality with $d_m$ dimensions, we repeat this process for each dimension of the input. This allows us to obtain an interpolation matrix $\hat{X}_h = [\hat{x}_{1,h}, \hat{x}_{2,h}, ..., \hat{x}_{d_m,h}] \in \mathbb{R}^{\gamma \times d_m}$ for each of the $h$ embedding functions. We then concatenate the interpolation matrices across all $H$ embedding functions (i.e., $I = [\hat{X}_1, \hat{X}_2, ..., \hat{X}_h] \in \mathbb{R}^{\gamma \times (H \cdot d_m)}$) and employ a linear projection to achieve a final, discretized embedding for each modality, $Z \in \mathbb{R}^{\gamma \times d_e}$, where $d_e$ denotes the desired dimensionality of each modality's representation. The discretization procedure offers a standardized approach to managing irregularly sampled time series across various input types; however, it can inevitably result in information loss. On the other hand, relying solely on the mTAND module may yield suboptimal performance due to the potentially varying sampling rates of different variables [31], especially in scenarios where the sample sizes are small. To mitigate this, we combine discretized outputs with continuous representations learned through the mTAND module.

**Encoding Multiple Modalities** The process described above allows us to discretize an arbitrarily long irregular, multivariate sequence into a regularly sampled, discretized embedding with length $\gamma$ and dimensionality $d_e$. We repeat this for each of the $M$ modalities, to create $M$ embeddings, $\{Z_j\}_{j=1}^M$, which are then combined to generate predictions.

## 2.3 MoE Fusion Layer

**Router Design Study** Upon obtaining embeddings from each of the $j$ modalities, we propose multiple complementary approaches for processing multimodal inputs. Figure 2 illustrates a range of router design options. The most straightforward strategy involves employing a common router that handles the concatenated embeddings of all $j$ modalities, without imposing any gating constraints.

As the complexity increases with additional modalities, we consider more sophisticated alternatives: deploying separate routers for each modality's embedding and assigning these embeddings to a shared pool of experts. This allows for distinct processing while maintaining a unified expert framework. Additionally, we further segregate these common expert pools, allowing each router to direct its respective embedding to dedicated experts skilled in handling such specific inputs. These varied router design choices offer users enhanced flexibility, enabling more fine-grained control of both inter-modal and intra-modal relationships. Details of the respective advantages and challenges of these router design mechanisms can be found in Appendix C.

We implement an entropy regularization loss to ensure balanced and stable expert utilization, a concept supported by various previous studies [58, 57, 22]. It maximizes the mutual information between modalities and experts and serves as an auxiliary loss function in addition to task-specific loss. Given a total of $M$ modalities, and denoting $\mathcal{H}$ as the entropy, we define the loss function $\mathcal{E}$ as

$$\mathcal{E}(x) = \frac{1}{M}\sum_{j=1}^{M}\mathcal{H}(\hat{p}_{m_j}(E)) - \mathcal{H}(\frac{1}{M}\sum_{j=1}^{M}\hat{p}_{m_j}(E)), \tag{3}$$

where $\hat{p}_{m_j}(E)$ is the distribution over the experts $\{E_i\}_{i=1}^{S}$ for the $j^{\text{th}}$ modality. This distribution can be approximated by $\hat{p}_{m_j}(E) = \frac{1}{l^j}\sum_{i=1}^{l^j}p_{m_j}(E \mid x_i^{m_j})$, where $l^j$ is the number of observations of the $j^{\text{th}}$ modality. Intuitively, we actively encourage the input embeddings to diminish the uncertainty in selecting experts. By incorporating the loss $\mathcal{E}$, we aim to stabilize the experts' preferences within each modality, while promoting a diverse range of expert selections across different modalities.

**Missing Modalities**   In scenarios where certain modalities are missing throughout the data trajectories, we substitute the original embedding $Z_{\text{missing}}$ with a learnable embedding $\mathcal{Z}$, acting as a generic "missing indicator". This strategy is facilitated by employing per-modality routers, which, in conjunction with entropy regularization, guide $\mathcal{Z}$ predominantly toward a specific group of less-utilized experts. The new embeddings $\mathcal{Z}$ are dynamically adjusted throughout the model training process to minimize the task-specific loss and the entropy regularization loss. As a result, the router will assign lower weights to the experts responsible for processing these embeddings.

## 3   Theoretical Contribution

In this section, we provide a theoretical guarantee of the benefits of the Laplace gating over the standard Softmax gating in MoE. In particular, we conduct a convergence analysis for maximum likelihood estimation (MLE) under the Lapace gating Gaussian MoE, and demonstrate that the MLE under this model has better convergence behaviors than that under the softmax gating Gaussian MoE.

**Problem Setup.**   Since the convergence analysis of MLE under the Top-K sparse gating MoE has been studied in [63], we will focus on examining the Laplace gating solely in the sequel. Assume that $(X_1, Y_1), \ldots, (X_n, Y_n) \in \mathbb{R}^d \times \mathbb{R}$ are i.i.d. samples drawn from the Laplace gating Gaussian MoE of order $k_*$ whose conditional density function $p_{G_*}(Y|X)$ is

$$p_{G_*}(Y|X) = \sum_{i=1}^{k_*}\text{softmax}(-\|W_i^* - X\| + \beta_i^*) \cdot f(Y|(a_i^*)^\top X + b_i^*, \nu_i^*), \tag{4}$$

where we define for any vectors $v = (v_i)_{i=1}^{k_*}$ that $\text{softmax}(v_i) := \frac{\exp(v_i)}{\sum_{j=1}^{k_*}\exp(v_j)}$. Above, $f(\cdot|\mu, \nu)$ denotes a univariate Gaussian density function with mean $\mu$ and variance $\nu$. For ease of the presentation, we denote $G_* := \sum_{i=1}^{k_*}\exp(\beta_i^*)\delta_{(W_i^*, a_i^*, b_i^*, \nu_i^*)}$ as a true but unknown *mixing measure* associated with unknown parameters $(\beta_i^*, W_i^*, a_i^*, b_i^*, \nu_i^*)$ for $i \in \{1, 2, \ldots, k_*\}$. In the paper, we specifically consider two settings of the true number of experts $k_*$: (i) *Exact-specified setting*: when $k_*$ is known; (ii) *Over-specified setting*: when $k_*$ is unknown, and we over-specify the model in equation 4 by a Laplace gating MoE model with $k > k_*$ experts. However, due to the space limit, we present only the latter setting, and defer the former setting to Appendix J.

**Maximum Likelihood Estimation.** We use the maximum likelihood method to estimate the unknown mixing measure $G_*$ [91]. In particular, the MLE is given by

$$\widehat{G}_n \in \arg\max_{G \in \mathcal{G}_k(\Theta)} \frac{1}{n} \sum_{i=1}^{n} \log(p_G(Y_i|X_i)), \tag{5}$$

where $\mathcal{G}_k(\Theta) := \{G = \sum_{i=1}^{k'} \exp(\beta_i)\delta_{(W_i,a_i,b_i,\nu_i)} : 1 \leq k' \leq k, \ (W_i, a_i, b_i, \nu_i) \in \Theta\}$ denotes the set of all mixing measures with at most $k$ components. Given the MLE defined in equation 5, we are ready to present the main results. Before that, let us introduce some necessary notations for our analysis.

**Notations.** We denote $[n] := \{1, 2, \ldots, n\}$ for any $n \in \mathbb{N}$. For any vector $v \in \mathbb{R}^d$, $\|v\|$ stands for its 2-norm value. Additionally, the notation $|S|$ indicates the cardinality of a given set $S$, while $\delta$ denotes the Dirac delta measure. Finally, for any two probability densities $p, q$ dominated by the Lebesgue measure $\mu$, we denote $V(p, q) = \frac{1}{2} \int |p - q| d\mu$ as their Total Variation distance.

Firstly, we demonstrate in Theorem 3.1 that the convergence rate of density estimation under the Laplace gating Gaussian MoE is parametric on the sample size $n$.

**Theorem 3.1** (Density estimation)**.** *The density estimation $p_{\widehat{G}_n}(Y|X)$ converges to the true density $p_{G_*}(Y|X)$ under the Total Variation distance at the following rate:*

$$\mathbb{E}_X[V(p_{\widehat{G}_n}(\cdot|X), p_{G_*}(\cdot|X))] = \mathcal{O}(\sqrt{\log(n)/n}).$$

Proof of Theorem 3.1 is in Appendix K.2. The parametric rate $\mathcal{O}(\sqrt{\log(n)/n})$ of the conditional density function $p_{\widehat{G}_n}$ indicates that if there exists a loss function among parameters $\mathcal{D}$ such that $\mathbb{E}_X[V(p_{\widehat{G}_n}(\cdot|X), p_{G_*}(\cdot|X))] \gtrsim \mathcal{D}(\widehat{G}_n, G_*)$, then we will achieve the parameter and expert estimation rates via the bound $\mathcal{D}(\widehat{G}_n, G_*) = \mathcal{O}(\sqrt{\log(n)/n})$.

**Voronoi Loss.** Following the above implication, we now define a loss function among parameters based on a notion of Voronoi cells as in [55]. Given some mixing measure $G$, we distribute its components $\theta_i := (W_i, a_i, b_i, \nu_i)$ to the following Voronoi cells, which are generated by the components $\theta_j^* := (W_j^*, a_j^*, b_j^*, \nu_j^*)$ of the true mixing measure $G_*$:

$$\mathcal{A}_j \equiv \mathcal{A}_j(G) := \{i \in [k] : \|\theta_i - \theta_j^*\| \leq \|\theta_i - \theta_\ell^*\|, \ \forall \ell \neq j\}, \tag{6}$$

for any $1 \leq j \leq k_*$. Note that, the cardinality of the Voronoi cell $\mathcal{A}_j$ is exactly the number of fitted components approximating $\theta_j^*$. For ease of the presentation, let us denote $\Phi_{ij}(\rho_1, \rho_2, \rho_3, \rho_4) := \|W_i - W_j^*\|^{\rho_1} + \|a_i - a_j^*\|^{\rho_2} + |b_i - b_j^*|^{\rho_3} + |\nu_i - \nu_j^*|^{\rho_4}$, for any $(\rho_1, \rho_2, \rho_3, \rho_4) \in \mathbb{R}^4$. Then, the Voronoi loss function $\mathcal{D}_2(G, G_*)$ used for our analysis under the over-specified setting is given by:

$$\mathcal{D}_2(G, G_*) := \sum_{j=1}^{k_*} \Big| \sum_{i \in \mathcal{A}_j} \exp(\beta_i) - \exp(\beta_j^*) \Big| + \sum_{j \in [k_*] : |\mathcal{A}_j| = 1} \sum_{i \in \mathcal{A}_j} \exp(\beta_i)\Phi_{ij}(1, 1, 1, 1)$$

$$+ \sum_{j \in [k_*] : |\mathcal{A}_j| > 1} \sum_{i \in \mathcal{A}_j} \exp(\beta_i)\Phi_{ij}\Big(2, 2, \bar{r}(|\mathcal{A}_j|), \frac{\bar{r}(|\mathcal{A}_j|)}{2}\Big). \tag{7}$$

The notation $\bar{r}(|\mathcal{A}_j|)$ stands for the minimum value of $r \in \mathbb{N}$ such that the following system of polynomial equations does not have any non-trivial solutions for the unknown variables $\{(q_{1i}, q_{2i}, q_{3i})\}_{i=1}^{|\mathcal{A}_j|}$:

$$\sum_{i=1}^{|\mathcal{A}_j|} \sum_{\substack{m_1 + 2m_2 = s, \\ 1 \leq m_1 + m_2 \leq r}} \frac{q_{3i}^2 q_{1i}^{m_1} q_{2i}^{m_2}}{m_1! m_2!} = 0, \text{ for each } s = 1, 2, \ldots, r, \tag{8}$$

A solution to the above system is regarded as non-trivial if at least among variables $q_{1i}$ is different from zero, whereas all the variables $q_{3i}$ are non-zero. It is worth noting that the function $\bar{r}(\cdot)$ was previously studied in [28] to characterize the convergence behavior of parameter estimation under the location-scale Gaussian mixture models. [28] also gave some specific values of that function, namely $\bar{r}(2) = 4$ and $\bar{r}(3) = 6$. Meanwhile, they claimed that it was non-trivial to determine the

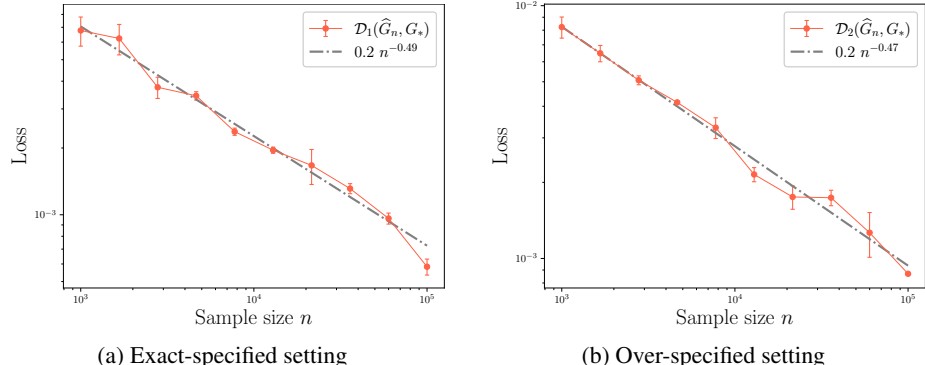

|   |   |
|:-:|:-:|
| (a) Exact-specified setting | (b) Over-specified setting |

Figure 3: Log-log scaled plots illustrating simulation results under the exact-specified (left) and the over-specified settings (right). The orange curves depict the mean discrepancy between the MLE $\widehat{G}_n$ and the true mixing measure $G_*$, accompanied by error bars signifying two empirical standard deviations. Additionally, the gray dash-dotted line represents the least-squares fitted linear regression line for these data points. Finally, the loss functions $\mathcal{D}_1$ and $\mathcal{D}_2$ are defined in equations equation 9 and equation 7, respectively. See Appendix I for the experimental details.

value of $\bar{r}(m)$ when $m \geq 4$, and further techniques should be developed for that purpose. Since Gaussian MoE models are generalization of the Gaussian mixture models, we also involve the function $\bar{r}(\cdot)$ in our convergence analysis. Now, we provide in the following theorem the convergence rate of parameter estimation under the over-specified setting of the Laplace gating Gaussian MoE model (see also Figure 3 for the empirical convergence rates justifying the theoretical rates in Theorem 3.2).

Table 2: Parameter estimation rates under the Softmax and Laplace gating Gaussian MoE models. The function $\widetilde{r}(\cdot)$ represents the solvability of a system of polynomial equations considered in [64] while $\widetilde{r}(\cdot) \leq \bar{r}(\cdot)$ and $\widetilde{r}(2) = 4, \widetilde{r}(3) = 6$. Additionally, $\mathcal{A}_j^n := \mathcal{A}_j(\widehat{G}_n)$ denotes a Voronoi cell defined in equation 6.

| **Gates** | $\exp(\beta_j^*)$ | $W_j^*$ | $a_j^*$ | $b_j^*$ | $\nu_j^*$ |
|:---:|:---:|:---:|:---:|:---:|:---:|
| Softmax [64] | $\mathcal{O}(n^{-1/2})$ | $\mathcal{O}(n^{-1/2\widetilde{r}(|\mathcal{A}_j^n|)})$ | $\mathcal{O}(n^{-1/\widetilde{r}(|\mathcal{A}_j^n|)})$ | $\mathcal{O}(n^{-1/2\widetilde{r}(|\mathcal{A}_j^n|)})$ | $\mathcal{O}(n^{-1/\widetilde{r}(|\mathcal{A}_j^n|)})$ |
| Laplace (Ours) | $\mathcal{O}(n^{-1/2})$ | $\mathcal{O}(n^{-1/4})$ | $\mathcal{O}(n^{-1/4})$ | $\mathcal{O}(n^{-1/2\bar{r}(|\mathcal{A}_j^n|)})$ | $\mathcal{O}(n^{-1/\bar{r}(|\mathcal{A}_j^n|)})$ |

**Theorem 3.2** (Parameter Estimation). *When $k > k_*$ becomes unknown, the following Total Variation bound holds true for any mixing measure $G \in \mathcal{G}_k(\Theta)$:*

$$\mathbb{E}_X[V(p_G(\cdot|X), p_{G_*}(\cdot|X))] \gtrsim \mathcal{D}_2(G, G_*).$$

*Consequently, we obtain that $\mathcal{D}_2(\widehat{G}_n, G_*) = \mathcal{O}(\sqrt{\log(n)/n})$.*

Proof of Theorem 3.2 is in Appendix K.3. The results of Theorem 3.2 together with the formulation of the loss function $\mathcal{D}_2$ in equation 7 reveal that (see also Table 2):

**(i)** The parameters $W_i^*, a_i^*, b_i^*, \nu_i^*$ which are fitted by exactly one component, i.e. $|\mathcal{A}_i^n| := |\mathcal{A}_i(\widehat{G}_n)| = 1$, enjoy the same estimation rate of order $\mathcal{O}(n^{-1/2})$ (up to some logarithmic factor), which match those in [64].

**(ii)** The rates for estimating the parameters $W_i^*, a_i^*, b_i^*, \nu_i^*$ which are fitted by more than one component, i.e. $|\mathcal{A}_i^n| > 1$, are no longer homogeneous. On the one hand, the estimation rates for the parameters $b_i^*$ and $\nu_i^*$ are of orders $\mathcal{O}(n^{-1/2\bar{r}(|\mathcal{A}_i^n|)})$ and $\mathcal{O}(n^{-1/\bar{r}(|\mathcal{A}_i^n|)})$, respectively, both of which are determined by the function $\bar{r}(\cdot)$ and vary with the number of fitted components $|\mathcal{A}_i^n|$. Those rates are comparable to their counterparts in [64]. On the other hand, the estimation rates for the gating parameters $W_i^*$ and the expert parameters $a_i^*$ are all of order $\mathcal{O}(n^{-1/4})$, which remains constant with respect to the number of fitted components. Meanwhile, those rates in [64] depend on a different system of polynomial equations from that in equation 8, which are significantly slower.

**Advantage of Laplace Gating on FlexiModal Setting** In the standard Softmax gating [64], the similarity score is computed as the inner product of a token's hidden representation and an expert

Table 3: MoE demonstrates improved performance averaged over 5 random experiments on the CMU-MOSI and MOSEI datasets; the best results are highlighted in **bold font** and the second best results are underlined.

| Method / Data | MOSI Dataset | | | | MOSEI Dataset | | | |
|---|---|---|---|---|---|---|---|---|
| | MAE↓ | Acc-2↑ | Corr↑ | F1↑ | MAE↓ | Acc-2↑ | Corr↑ | F1↑ |
| TFN | $0.90 \pm 0.02$ | $80.81 \pm 0.34$ | $0.70 \pm 0.04$ | $80.70 \pm 0.18$ | $0.59 \pm 0.03$ | $82.50 \pm 0.58$ | $0.68 \pm 0.02$ | $82.10 \pm 0.41$ |
| MulT | $0.86 \pm 0.01$ | $84.10 \pm 0.21$ | $0.71 \pm 0.02$ | $83.90 \pm 0.27$ | $0.58 \pm 0.02$ | $82.51 \pm 0.41$ | $0.71 \pm 0.04$ | $82.31 \pm 0.27$ |
| MAG | $0.71 \pm 0.04$ | $86.10 \pm 0.44$ | $0.80 \pm 0.03$ | $86.00 \pm 0.09$ | $0.57 \pm 0.07$ | $85.56 \pm 0.22$ | $0.79 \pm 0.02$ | $84.50 \pm 0.18$ |
| Softmax-MoE | $0.69 \pm 0.01$ | $87.09 \pm 0.18$ | $0.82 \pm 0.02$ | $87.29 \pm 0.22$ | $0.55 \pm 0.03$ | $86.34 \pm 0.23$ | $0.76 \pm 0.05$ | $84.97 \pm 0.32$ |
| Joint experts&router | $0.67 \pm 0.02$ | $87.28 \pm 0.35$ | $0.82 \pm 0.03$ | $87.35 \pm 0.24$ | **$0.54 \pm 0.01$** | **$86.41 \pm 0.36$** | $0.81 \pm 0.05$ | **$85.43 \pm 0.25$** |
| Per-mod router | **$0.65 \pm 0.04$** | **$88.23 \pm 0.57$** | **$0.84 \pm 0.01$** | **$87.39 \pm 0.13$** | $0.56 \pm 0.02$ | $86.12 \pm 0.19$ | $0.78 \pm 0.02$ | $85.07 \pm 0.14$ |
| Disjoint router | $0.73 \pm 0.02$ | $86.37 \pm 0.33$ | $0.81 \pm 0.04$ | $86.89 \pm 0.21$ | $0.55 \pm 0.02$ | $85.67 \pm 0.31$ | **$0.81 \pm 0.01$** | $85.21 \pm 0.22$ |

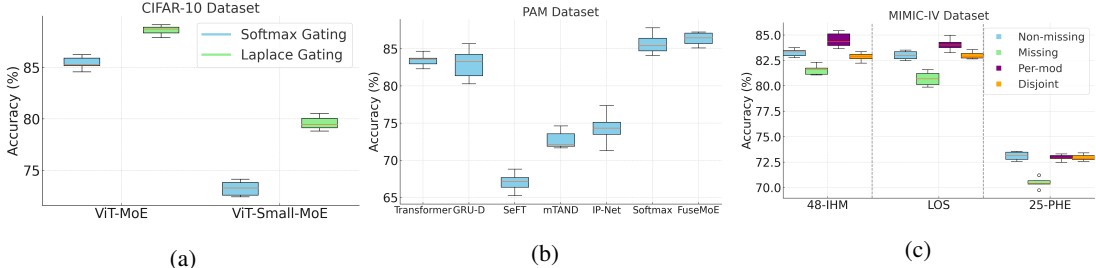

Figure 4: (a) The Laplace gating mechanism enhances CIFAR-10 classification when integrated into Vision-MoE [77]. We employed Vision Transformer (ViT) [17] and ViT-small as the backbone models and selectively replaced their FFN layers with MoE layers; (b) `FuseMoE` improves prediction on PAM dataset over baseline time series models; (c) Per-modality routers and the entropy loss $\mathcal{E}$ mitigate the impact of missing modalities.

embedding. However, this approach can lead to *representation collapse* [13, 69], where a subset of experts dominates the decision-making process, resulting in the redundancy of other experts. This issue likely contributes to the slow rates of estimating expert parameters $a_i^*$ in this setting (see Table 2). By contrast, the Laplace gating function partially alleviates this problem by computing the similarity score as the $L_2$-distance between token representations and expert embeddings. This approach does not inherently favor any expert based on magnitude, unlike inner product which can be biased towards experts with larger norms. The Laplace gating ensures that all experts have a more balanced opportunity to be selected based on how close they are to the token representation. Therefore, Laplace gating is beneficial when dealing with heterogeneous inputs, such as multimodal data, where its feature distributions can be very different across modalities. This is because it can handle these differences without being overly sensitive to the scale and variance of the input features. In addition, it can gracefully degrade in the presence of missing data, rather than causing abrupt changes in gating probabilities that might occur with inner product-based measures. The improved estimation rates for expert parameters $a_i^*$ under the Laplace gating Gaussian MoE, along with our empirical results on multiple large-scale datasets, substantiate these insights.

# 4  Experiments

**Overview**  We demonstrate that `FuseMoE` can provide accurate and efficient predictions when applied to the FlexiModal setting. We tested `FuseMoE` on a diverse set of benchmarks, including MIMIC-III [40] and MIMIC-IV [36], CMU-MOSI and MOSEI [97], the Physical Activity Monitoring (PAM) dataset [75], and CIFAR-10 [46]. Compared to CMU-MOSI and MOSEI, the MIMIC ecosystem exhibits irregular and missing modality patterns and includes distinct modalities unlike PAM and CIFAR-10. Evaluating `FuseMoE` across these diverse datasets provides various empirical insights into critical aspects of our model's performance. Comprehensive details on the datasets, metrics, parameters, and additional results are thoroughly presented in the Appendices.

## 4.1  Main Results

**CMU-MOSI and MOSEI Datasets**  We first apply our method to the CMU-MOSI and MOSEI datasets [97], which utilize visual, acoustic, and textual data for sentiment analysis and emotion recognition tasks. Our methodology employs pre-trained T5 [73] for text encoding, librosa [56] for audio feature extraction, and EfficientNet [84] for video feature encoding. Table 3 details the per-

Table 4: Comparison of `FuseMoE`-based methods (gray) and baselines, utilizing vital signs and clinical notes of MIMIC-IV [36]. The best results are highlighted in **bold font**, and the second-best results are underlined. All results are averaged across 5 random experiments.

| Task \ Method | | MISTS | MulT | MAG | TFN | HAIM | Softmax | Gaussian | Laplace |
|---|---|---|---|---|---|---|---|---|---|
| 48-IHM | AUROC | $75.06 \pm 1.03$ | $75.95 \pm 0.84$ | $75.82 \pm 0.73$ | $78.76 \pm 0.79$ | $79.65 \pm 0.00$ | $79.49 \pm 0.83$ | $80.76 \pm 0.56$ | **$81.03 \pm 0.25$** |
| | F1 | $45.61 \pm 0.34$ | $38.81 \pm 0.22$ | $42.55 \pm 0.82$ | $40.61 \pm 0.41$ | $39.79 \pm 0.00$ | $42.86 \pm 0.44$ | **$46.86 \pm 0.24$** | $46.53 \pm 0.57$ |
| LOS | AUROC | $80.56 \pm 0.33$ | $81.36 \pm 1.32$ | $81.13 \pm 0.66$ | $80.71 \pm 0.45$ | $82.58 \pm 0.00$ | $82.11 \pm 0.39$ | $81.92 \pm 0.73$ | **$82.91 \pm 1.02$** |
| | F1 | $73.01 \pm 0.52$ | $73.45 \pm 0.59$ | $72.51 \pm 0.27$ | $73.84 \pm 0.61$ | $73.18 \pm 0.00$ | $74.43 \pm 0.88$ | $74.46 \pm 0.52$ | **$74.58 \pm 0.63$** |
| 25-PHE | AUROC | $69.45 \pm 0.72$ | $66.58 \pm 0.41$ | $69.55 \pm 0.67$ | $69.18 \pm 0.32$ | $63.39 \pm 0.00$ | $70.54 \pm 0.47$ | $70.42 \pm 0.26$ | **$71.23 \pm 0.53$** |
| | F1 | $28.59 \pm 0.46$ | $28.55 \pm 0.31$ | $27.86 \pm 0.29$ | $28.52 \pm 0.22$ | **$42.13 \pm 0.00$** | $31.25 \pm 0.18$ | $30.44 \pm 0.27$ | $31.33 \pm 0.19$ |

formance of various router design mechanisms within our MoE architecture, utilizing the Laplace gating function, compared against representative baselines. The baselines include (1) the early fusion method, Tensor Fusion Network (TFN) [96]; (2) the Multimodal Transformer (MulT), which fuses modalities by modeling their interactions [90]; (3) the Multimodal Adaptation Gate (MAG), which focuses on the consistency and differences across modalities [74]; and (4) multimodal fusion using standard MoE with the Softmax gating function. Results indicate that employing an MoE backbone—regardless of the gating function chosen or whether utilizing per-modality routers or a joint experts & router configuration—significantly enhances performance on the multimodal task. This improvement is attributed to the MoE's ability to effectively allocate specific components to handle distinct input modalities, thus better addressing both inter- and intra-modal relationships.

**CIFAR-10 Dataset** Subsequently, we evaluate our method using the Vision-MoE framework [77] on the CIFAR-10 classification task [46], with results illustrated in Figure 4(a). In this experiment, we selectively replace the FFN layers with an even number in the Vision Transformer (ViT) models with MoE layers. These results, along with Table 3 on the CMU-MOSI and MOSEI datasets comparing Softmax-gating MoE, indicate that the Laplace gating function surpasses the standard Softmax gating function in performance. This outcome is consistent with our theoretical claims.

**MIMIC-IV and PAM Datasets** We then conduct comprehensive evaluations of `FuseMoE` on MIMIC-IV [36], and the Physical Activity Monitoring (PAM) dataset [75]. These datasets feature multiple input modalities, each *characterized by varying degrees of irregular sampling or significant levels of missingness*. Our tasks of interest for MIMIC datasets include the 48-hour in-hospital mortality prediction (48-IHM), 25-type phenotype classification (25-PHE), and length-of-stay (LOS) prediction. In addition to the previously mentioned baselines, we have incorporated the HAIM method [83], a data pipeline specifically designed for integrating multimodal data from the MIMIC-IV dataset. We also include the cross-attention combined with irregular sequences modeling approach (MISTS) [101]. Table 4 shows the outcomes of combining irregular vital signs and clinical notes from the MIMIC-IV dataset. In addition to the commonly used Softmax gating function, we also evaluated the Gaussian gating function [93] as a comparative benchmark. The `FuseMoE`-based methods surpass baselines in most scenarios, often by a non-trivial margin. Furthermore, we observe that HAIM shows considerable efficacy in extracting features from time series, resulting in a strong performance in the 48-IHM and LOS tasks, which are heavily reliant on such data. However, its performance appears more moderate on the 25-PHE task. The PAM dataset captures daily living activities through 17 sensors, with data from each sensor treated as a separate modality. These modalities are individually processed through time-series and irregularity encoders before being integrated into the FuseMoE framework. Our baselines include the Transformer [92], GRU-D [10], SeFT [31], a mTAND-only configuration, and IP-Net [81]. We use the Laplace gating and its joint experts & router structure in these experiments. The results in Figure 4(b) have again shown the efficacy of integrating the irregularity encoder with the MoE fusion layer.

## 4.2 Ablation Studies

**Scalability of FuseMoE with Increasing Modalities** Table 5 presents the revised outcomes of the MIMIC-IV dataset after integrating CXR and ECG of corresponding patients, employing the per-modality router and the entropy loss $\mathcal{E}$ within `FuseMoE`. This setup was chosen as it slightly outperformed the joint router with an increase in modalities. Relative to their two-modality versions, `FuseMoE` has effectively harnessed additional information (notably from CXR), resulting in

Table 5: Incorporating CXR and ECG into `FuseMoE` leads to a noticeable enhancement as compared to their two-modality counterparts. All results are averaged across 5 random experiments.

| Task \ Method | | Vital & Notes & CXR | | | | Vital & Notes & CXR & ECG | | | |
|---|---|---|---|---|---|---|---|---|---|
| | | HAIM | Softmax | Gaussian | Laplace | HAIM | Softmax | Gaussian | Laplace |
| 48-IHM | AUROC | 78.87 ± 0.00 | 83.13 ± 0.36 | 83.64 ± 0.47 | **83.87 ± 0.33** | 78.87 ± 0.00 | 82.92 ± 0.22 | 83.03 ± 0.85 | **83.55 ± 0.49** |
| | F1 | 39.78 ± 0.00 | **46.82 ± 0.28** | 38.87 ± 0.26 | 45.36 ± 0.46 | 39.78 ± 0.00 | 46.87 ± 0.17 | 44.04 ± 0.26 | **46.88 ± 0.42** |
| LOS | AUROC | 82.46 ± 0.00 | **83.76 ± 0.59** | 83.64 ± 0.52 | 83.51 ± 0.51 | 82.46 ± 0.00 | 83.53 ± 0.34 | 83.47 ± 0.37 | **83.58 ± 0.78** |
| | F1 | 72.75 ± 0.00 | 74.32 ± 0.44 | **76.59 ± 0.74** | 75.18 ± 0.77 | 72.75 ± 0.00 | 75.01 ± 0.63 | 74.43 ± 0.64 | **75.11 ± 0.65** |
| 25-PHE | AUROC | 63.57 ± 0.00 | **73.87 ± 0.71** | 72.68 ± 0.61 | 73.65 ± 0.39 | 63.82 ± 0.00 | 73.64 ± 0.89 | **73.74 ± 0.41** | 73.67 ± 0.71 |
| | F1 | **42.80 ± 0.00** | 35.96 ± 0.23 | 35.09 ± 0.15 | 36.01 ± 0.42 | **43.20 ± 0.00** | 36.06 ± 0.17 | 36.46 ± 0.55 | 35.81 ± 0.34 |

a significant enhancement in performance. Conversely, the addition of new modalities did not benefit the HAIM method, possibly due to its reliance on vital signs and clinical notes without adequately addressing the dynamics between different modalities. Furthermore, HAIM's notably high F1 scores on the 25-PHE task can be attributed to XGBoost's proficiency in managing missing minority classes. Note that, except for HAIM, other baselines were not designed to be agnostic to the quantity and variety of input modalities. Therefore, adapting them to manage extra and missing modalities requires considerable model changes, which might compromise their performance.

**Missing Modalities**  Figure 4(c) illustrates the effectiveness of utilizing per-modality routers and the entropy loss $\mathcal{E}$ in addressing missing modalities. Initially, we compare the performance of `FuseMoE` on patients with fully available modalities against those with missing components, employing a joint router mechanism with the importance loss function [79], to ensure load balancing. The inclusion of datasets with missing modalities, while expanding the sample size, resulted in a decrease in performance due to the compromised data quality. However, a performance enhancement was observed upon integrating per-modality or disjoint routers with $\mathcal{E}$. Notably, the outcomes for the 48-IHM and LOS tasks with missing modalities surpassed those obtained from datasets without any missingness. This is because the per-modality approach can better separate the present and missing modalities, reducing the influence of experts responsible for processing the absent inputs. Therefore, this leads to a more efficient exploitation of a broader array of samples.

## 5   Discussions and Limitations

In this paper, we introduced `FuseMoE`, a model adept at managing multimodal data characterized by random missingness or irregularity—a crucial yet relatively unexplored challenge. `FuseMoE` integrates MoE fusion layers with modal embeddings and offers multiple router configurations to adeptly handle multimodal inputs across different complexity levels. `FuseMoE` also employs an innovative Laplace gating function, which provides better theoretical results. Through empirical evaluation, `FuseMoE` has demonstrated superior performance across diverse scenarios. However, our current approach to encoding irregularities may potentially lead to over-parameterization when the input size is small. In our future work, we aim to identify simpler and more efficient methods to handle the irregularities of input samples while preserving the model's overall performance.

## Acknowledgement

Xing Han and Suchi Saria acknowledge support from the National Science Foundation (NSF) and the Gordon and Betty Moore Foundation. Carl Harris acknowledges support from the NSF Graduate Research Fellowship under Grant No. DGE 2139757. Nhat Ho acknowledges support from the NSF IFML 2019844 and the NSF AI Institute for Foundations of Machine Learning. Any opinion, findings, and conclusions or recommendations expressed in this material are those of the authors(s) and do not necessarily reflect the views of the NSF, Gordon and Betty Moore Foundation.

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

# Appendix for
# "FuseMoE: Mixture-of-Experts Transformers for Fleximodal Fusion"

## Contents

# A    Related Works

**Multimodal Fusion**    Initial approaches to multimodal fusion incorporated techniques such as kernel-based methods [7, 11, 70], graphical models [59, 21, 76], and neural networks [60, 20, 67]. With the diverse evolution of deep learning models, numerous advanced methods have now been employed in the fusion of multimodal data. In the realm of sentiment analysis, [96, 52] employ a low-rank Tensor Fusion method that leverages both language and video content. Attention-gating mechanisms are used by [74, 95] to generate displacement vectors through cross-modal self-attention, which are then added to the input vectors from the primary modality. [90] takes an alternative approach by integrating multiple layers of cross-modal attention blocks in a word-level vision/language/audio alignment task.

In the context of clinical prediction, [45, 16] adopt a late fusion approach to combining vital sign and text data by concatenating embeddings from pre-trained feature extractors. [83] developed a generalizable data preprocessing and modeling pipeline for EHR encompassing four data modalities, albeit through a direct concatenation of existing feature embeddings for each modality followed by an XGBoost classifier [12]. Recently, [101] expanded on the work of [90] by introducing a discretized multi-time attention (mTAND) module [82] to encode temporal irregularities in time series and text data. Their fusion approach involves layering sets of self- and cross-modal attention blocks. However, this approach is limited to just two modalities and is not easily extendable to include additional modal components or handle missing modalities. To the best of our knowledge, existing works are tailored to application-specific settings that necessitate the computation of pairwise cross-modal relationships, which are not scalable to more general settings with arbitrary modalities. Moreover, these studies typically do not account for scenarios where modalities are missing, or rely on imputation approaches based on observed data.

**Mixture-of-Experts**    MoE [35, 93] has gained significant popularity for managing complex tasks since its introduction three decades ago. Unlike traditional models that reuse the same parameters for all inputs, MoE selects distinct parameters for each specific input. This results in a sparsely activated layer, enabling a substantial scaling of model capacity without a corresponding increase in computational cost. Recent studies have demonstrated the effectiveness of integrating MoE with cutting-edge models across a diverse range of tasks [79, 19, 103]. These works have also tackled key challenges such as accuracy and training instability [66, 104, 72]. Given its ability to assign input partitions to specialized experts, MoE naturally lends itself to multimodal applications. This approach has been explored in fields such as vision-language modeling [58, 80] and dynamic image fusion [9]. However, the application of MoE in complex real-world settings, such as those involving FlexiModal Data, remains largely unexplored. This gap presents an opportunity to leverage MoE's potential in handling its intricate and multifaceted nature such as multimodal EHR, where reliable multimodal integration is crucial.

**MoE Theory**    While MoE has been widely employed to scale up large models, its theoretical foundations have remained nascent. Recently, [64] provided convergence rates for both density and parameter estimation of Softmax gating Gaussian MoE. They connected these rates to the solvability of systems of polynomial equations under Voronoi-based loss functions. Later, [63] extended these theories to the top-K sparse softmax gating Gaussian MoE. Their theories further characterize the effect of the sparsity of gating functions on the behaviors of parameter estimation and verify the benefits of using top-1 sparse softmax gating MoE in practice. Other theoretical results for MoE include estimation rates of parameters and experts for multinomial logistic MoE [62], for dense-to-sparse gating MoE [61], for Gaussian gating MoE [65], and for input-independent gating MoE [29].

# B    FlexiModal Data and Tasks of Interest

**Definition of FlexiModal Data**    We provide a generic definition for the FlexiModal Data as we used throughout the paper. Let $\mathcal{D} = \{(x_i^{m_1}, t_i^{m_1}), (x_i^{m_2}, t_i^{m_2}), \ldots, (x_i^{m_j}, t_i^{m_j}), y_i\}_{i=1}^{N}$ to be the FlexiModal dataset with $N$ units, where $x_i^{m_j}$ represents the input sequence from the $i^{\text{th}}$ unit of the $j^{\text{th}}$ modality, $t_i^{m_j}$ denotes the corresponding time points, and $y_i$ is the task-specific outcome. Take multimodal EHR as an example, each $j^{\text{th}}$ modality, which may vary from time-series data like heart

rate, blood pressure, and glucose levels to high-dimensional inputs such as clinical notes and X-rays, contains $l^{m_j}$ observations. Figure 5 is a more specific illustration of the FlexiModal example.

## B.1 MIMIC-IV and MIMIC-III Datasets

**Tasks** In the ICU, where rapid and informed decisions are crucial, accurate mortality prediction is essential to provide clinicians with advanced warnings of patient deterioration, aiding in critical decision-making processes [5]. Similarly, the prediction of patient length-of-stay is indispensable for optimizing treatment plans, resource allocation, and discharge processes [6]. Further, phenotyping of critical care conditions is highly relevant to comorbidity detection and risk adjustment and presents a more challenging task than binary classification, due to the heterogeneous presentation of conditions and the larger number of prediction tasks [101]. We concentrate on three critical care tasks as highlighted in [44], performing extensive empirical analysis on each building block of the proposed framework.

- **48-IHM** In this binary classification task, we predict in-hospital mortality based on the first 48 of the ICU stay for patients who stayed in the ICU for at least 48 hours.

- **LOS** We formulate our length-of-stay task similar to that of 48-IHM: for patients who spent at least 48 hours in the ICU, we predict ICU discharge without expiration within the following 48 hours.

- **25-PHE** In this multilabel classification problem, we attempt to predict one of 25 acute care conditions [18, 53] (e.g., congestive heart failure, pneumonia, shock, etc.) at the *end* each each patient's ICU stay. Because the original task was designed for diagnoses based on ICD-9 codes, but MIMIC-IV includes both ICD-9 and ICD-10 codes, we map patients with diagnoses coded using ICD-10 using the conversion database provided by [8].

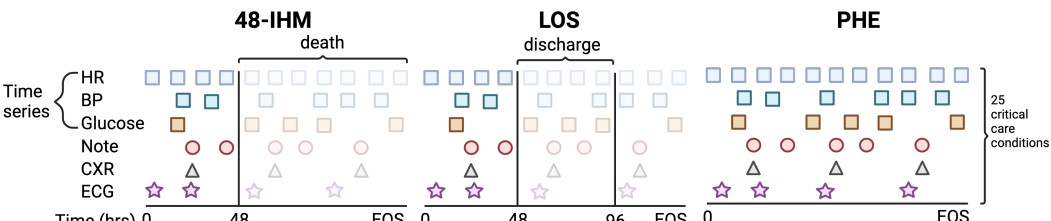

Figure 5: **Schematic of tasks of interest.** Plotted are example vitals/labs, radiological notes, X-rays, and ECGs sampled over the course of a patient's ICU stay. The first three rows represent example observations from a single modality consisting of three irregularly sampled vital signs (HR, BP), and lab values (Glucose). The following three rows represent irregularly sampled radiological notes, X-rays, and ECGs. Opaque shapes denote observations falling within the observation window (i.e., observations that are used to generate predictions), while translucent shapes are not used to generate predictions. For the **48-IHM** task, we use the first 48 hours of observations to predict death at any time during the ICU stay. For the **LOS** task, we use the first 48 hours of observations to predict whether the patient will be discharged (alive) during the following 48 hours. And in the phenotyping task (**PHE**), we use all observations to predict one of 25 critical care conditions.

We implement an in-hospital mortality prediction (**48-IHM**) task to evaluate our method's ability to predict short-term patient deterioration. Similarly, an accurate determination of patient discharge times is crucial for optimizing patient outcomes and hospital resource allocation [6], which motivates our length-of-stay (**LOS**) task. We frame 48-IHM and LOS as binary classification problems and use a 48-hour observation window (for patients who spent at least 48 hours in the ICU) to predict in-hospital mortality (48-IHM) and discharge (without expiration) within the 48 hours following the observation window (LOS). Lastly, identifying the presence of specific acute care conditions in patient records is essential for various clinical objectives, including the construction of cohorts for clinical studies and the detection of comorbidities [1]. Traditional methods, often reliant on manual chart reviews or simple billing code-based definitions, are increasingly being supplemented by machine learning techniques [27]; automating this process requires high-fidelity classifications, motivating our 25-type phenotype classification (**25-PHE**) task. In this multilabel classification problem, we attempt to predict one of 25 acute care conditions using data from the entire ICU stay.

**Evaluation**  In our initial analysis, we focused on patients with no missing modalities, resulting in a dataset comprised of 8,770 ICU stays for the 48-IHM and LOS tasks, and 14,541 stays for the 25-PHE task. For our analyses *with* missing observations, we include a total of 35,129 stays for 48-IHM and LOS, and 71.173 for 25-PHE. To evaluate the single-label tasks, 48-IHM and LOS, we employ the F1-score and AUROC as our primary metrics. In line with previous studies [101, 50, 3], we use macro-averaged F1-score and AUROC to assess the 25-PHE task.

**Dataset Information**  We leveraged data from MIMIC-IV [36], a comprehensive database with records from nearly $300k$ patients admitted to a medical center from 2008 to 2019, focusing on the subset of 73,181 ICU stays. We were able to link core ICU records (containing lab results and vital signs) to corresponding chest X-rays [39], radiological notes [38], and electrocardiogram (ECG) data [24] taking place during a given ICU stay. We allocated 70 percent of the data for model training, with the remaining 30 percent evenly split between validation and testing.

**Missingness Rates**  The total number of samples for each of our three tasks (i.e., those in which *at least one* vital sign was recorded in the specified observation window), along with the total number of observations per-modality, are shown in Table 6.

Table 6: We present the total number of ICU stays in each task, taking into account observations with missing modalities. The total number of stays with *at least one* observation of the corresponding modality are shown in the three right-most columns.

| Task(s) | Total | Text | CXR | ECG |
|---|---|---|---|---|
| 48-IHM & LOS | 35,129 | 32,038 | 8,781 | 18,271 |
| 25-PHE | 73,173 | 56,824 | 14,568 | 35,925 |

## B.2  MOSI and MOSEI Datasets

**Task**  We focus on the multimodal sentiment analysis (MSA) task which aims to predict sentiment polarity $\in$ {positive, negative, and neutral} and sentiment intensity, which is a real number ranging from -3 to +3 under a multimodal setting.

**Evaluation**  Following previous work such as [26], we adopt mean absolute error (MAE), Pearson correlation (Corr), binary classification accuracy, F1 score computed for non-negative/negative class as evaluation metrics.

**Dataset Information**  The CMU-MOSI dataset contains 1284/229/686 train/validation/test samples, and the CMU-MOSEI dataset contains 16326/1871/4659 train/validation/test samples. They are the largest dataset of multimodal sentiment analysis and emotion recognition to date. The datasets contain utterance videos from numerous online YouTube speakers, which are transcribed and properly punctuated, leading to multimodal input consisting of video frames, text, and audio signals.

## B.3  PAM Dataset

**Task**  Physical Activity Monitoring (PAM) dataset measures the daily living activities of 9 subjects with 3 inertial measurement units. PAM is labeled into 8 classes where each class represents an activity of daily living.

**Evaluation**  We choose common classification accuracy as the evaluation metric for this task.

**Dataset Information**  The processed PAM dataset contains 5,333 segments (samples) of sensory signals. Each sample is measured by 17 sensors and contains 600 continuous observations with the sampling frequency 100 Hz. PAM does not include static attributes and the samples are approximately balanced across all 8 categories.

Table 7: Dataset Summary

| Dataset | Research Area | Modalities | Sample Size | Tasks |
|---------|--------------|-----------|-------------|-------|
| MIMIC-III | Healthcare | Time-Series, Text | 36,212 | Mortality, length-of-stay, phenotyping |
| MIMIC-IV | Healthcare | Time-Series, Text, Images, ECG | 73,173 | Mortality, length-of-stay, phenotyping |
| CMU-MOSI | Affective Computing | Text, Video, Audio | 2,199 | Sentiment |
| CMU-MOSEI | Affective Computing | Text, Video, Audio | 22,777 | Sentiment, emotions |
| PAM | Healthcare | Time-Series | 5,333 | Activity recognition |
| CIFAR-10 | Multimedia | Images | 60,000 | Image classification |

## B.4 CIFAR-10 Dataset

CIFAR-10 [46] is an established computer-vision dataset used for object recognition. It consists of 60,000 32x32 color images containing one of 10 object classes ("plane", "car", "bird", "cat", "deer", "dog", "frog", "horse", "ship", "truck"), with 6000 images per class.

# C  Mechanisms of Different Router Designs

## C.1  Joint Experts & Routers

In this approach, a concatenated embedding of all modalities is created, and this combined input is directed to selected experts by the router. This method allows the model to capture interactions between modalities at the input level, as the concatenated embedding provides a unified representation that includes all modalities. The router and experts work with this comprehensive view, enabling the model to learn correlations and interactions directly from the fused data. However, this approach might not fully capture modality-specific nuances since the characteristics of each modality are blended into a single representation.

**Advantages**

1. Captures inter-modal relationships by considering all modalities together.
2. Simplifies the routing mechanism by treating the concatenated embedding as a single input.

**Challenges**

1. May overlook modality-specific features due to the blending of all modalities into one representation.
2. Could be less efficient if some modalities are irrelevant for certain tasks or experts.

## C.2  Modality-Specific Router

Each modality's embedding is independently assigned to a shared pool of experts by modality-specific routers. This setup allows the model to maintain the distinctiveness of each modality while still leveraging a common pool of expertise. By doing so, it can better capture modality-specific nuances and how they contribute independently to the overall task. However, this approach might be less effective in capturing complex inter-modal interactions since the initial routing is done independently for each modality.

**Advantages**

1. Preserves modality-specific information by routing each modality independently.
2. Flexible in directing modalities to the most relevant experts, potentially improving efficiency.
3. Captures interactions between modalities to some extent, but may not be as effective as joint routing approaches.

**Challenge**  needs additional coordination between independent routes to leverage cross-modal insights.

## C.3 Disjoint Experts & Routers

In this configuration, modality-specific routers assign each modality's embedding to separate pools of experts, with each pool uniquely tailored to process a specific modality type. This method maximizes the ability of the model to capture and exploit modality-specific features and relationships, as each pool of experts is optimized for a particular type of data. However, this setup might limit the model's ability to learn from the interactions between modalities, as each is processed in isolation.

**Advantages**

1. Allows for highly specialized processing of each modality, potentially improving performance on modality-specific tasks.

2. Modality-specific experts can develop deeper insights into the characteristics and patterns within their designated data type.

**Challenges**

1. Inter-modal relationships might be underutilized due to the segregated processing of each modality.

2. Requires additional coordination or subsequent integration stages to combine insights from different modality-specific experts.

Each router type offers unique benefits and faces specific challenges in capturing the subtle relationships between modalities. The choice among them depends on the specific requirements of the application, including the importance of preserving modality-specific information versus capturing inter-modal interactions, and the computational efficiency of managing multiple experts and routers. For example, we found that modality-specific routers are more effective in ameliorating the effect of missing modality in our experiments.

# D    Data Preprocessing

## D.1    MIMIC-IV

In the preprocessing stage, we focused on 30 pertinent lab and chart events from each patient's ICU record for vital sign measurements. For chest X-rays, we utilized a pre-trained DenseNet-121 model [14], which was fine-tuned on the CheXpert dataset [34], to extract 1024-dimensional image embeddings. For radiological notes, we obtained 768-dimensional embeddings using the BioClinicalBERT model [2]. ECG signals were processed using a convolutional autoencoder, adapted from [4], to generate a 256-dimensional embedding for each ECG.

**Time series**    We selected 30 time series events for inclusion, following [83]. Nine of these were vital signs: heart rate, mean/systolic/diastolic blood pressure, respiratory rate, oxygen saturation, and Glasgow Coma Scale (GCS) verbal, eye, and motor response. We also included 21 lab values: potassium, sodium, chloride, creatinine, urea nitrogram, bicarbonate, anion gap, hemoglobin, hematocrit, magnesium, platlet count, phosphate, white blood cell count, total calcium, MCH, red blood cell count, MCHC, MCV, RDW, platlet count, neutrophil count, and vancomycin. We standard scale each time series value to have mean 0 and standard deviation 1, based on the values in the training set.

**Chest X-rays**    To incorporate a medical imaging modality into our analyses, we use the MIMIC-CXR-JPG [37] module available from Physionet [23], which includes 377,110 JPG format images derived from the DICOM-based MIMIC-CXR database [39]. Following [83], for each image, we resize each JPG image to 224 × 224 pixels and then extract embeddings from the last layer of the Densenet121 model. We identify X-rays taken while the patient was in the ICU by first matching subject IDs in MIMIC-CXR-JPG with the core MIMIC-IV database, then limiting these matched X-rays to those with a chart time occuring between an ICU admission and discharge.

**Radiological notes**  To incorporate text data, we use the MIMIC-IV-Note module [38], which contains 2,321,355 deidentified radiology reports for 237,427 patients that can be matched with patients in the main MIMIC-IV via a similar approach to chest X-rays. We note that we were unable to obtain *intermediate* clinical notes (i.e., notes made by clinicians throughout a patient stay), as those have not yet been publicly released. We extract note embeddings using Bio-Clinical BERT [2].

**Electrocardiograms (ECGs)**  To include ECGs as an additional modality in our models, we utilize the MIMIC-IV-ECG [24] module, which includes approximately 800,000 ECGs (10 seconds, sampled at 500 Hz) collected from nearly 160,000 unique patients. To transform the ECGs so that they are suitable for input to our model, we adopt a convolutional autoencoder approach, adapted from [4], that compresses each ECG into a 256-dimensional vector. Specifically, each diagnostic ECG contains a $5000 \times 12$ dimensional vector (5000 time points $\times$ 12 ECG leads). To prepare the ECG for input to the autoencoder, we only include the first 4096 time points. We then train the autoencoder to compress the ECG into a 256-dimensional latent vector, and then reconstruct the original ECG using upsampling layers, using mean squared error as our loss function. The architecture is shown in Figure 6. We train the autoencoder with 90% of the ECGs available in the MIMIC-IV-ECG projection and use the rest for validation. We selected a batch size of 2048, and reduced the learning rate by a factor of $0.5$ if the validation loss had plateaued for 3 epochs. Training stopped if the validation loss had not decreased for 6 epochs. For our encoder, we use filter numbers of $[16, 16, 32, 32, 64, 64]$, kernel widths of $[5, 5, 5, 3, 3, 3]$ and a dropout rate of $0.1$. For the decoder, we use the same filter numbers and kernel widths in reverse, and maintain a dropout rate of $0.1$.

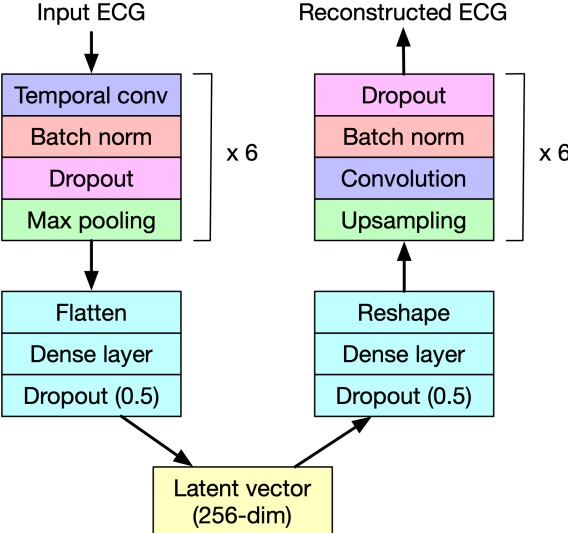

Figure 6: **CNN Autoencoder Architecture** The encoder consists of 6 convolutional blocks (temporal convolution, batch normalization, dropout, and max pooling layers), followed by a dense layer that reduces the dimensionality of the representation of 256. The decoder reconstructs the input ECG (dimensionality $4096 \times 12$) from this latent vector via a dense layer, followed by 6 upsampling convolutional blocks (upsampling, convolutional, batch normalization, and dropout layers).

### D.2  PAM Dataset

We follow the preprocessing procedure from [100] as published from their official GitHub repository[1].

---

[1]https://github.com/mims-harvard/Raindrop/tree/main/PAMdata

# E Modeling Irregularity

## E.1 Unified Temporal Discretization Embeddings

Unlike the embeddings in chest X-rays, clinical notes, and ECGs, vitals/lab/time-series values present temporal irregularity *across dimensions*. That is, for the former three modalities, each dimension of the corresponding is observed at each irregular time point $\tau$. By contrast, the sampling for vitals/labs is irregular in both *within* and *across* dimensions. For example, we might observe heart rate values sampled at times $\tau_{HR} = \{0, 0.2, 0.8, 1.2, 2.8\}$ and glucose values sampled at time $\tau_{Glu} = \{0.1, 0.7, 3.4\}$. Given this unique challenge present in vitals/labs, we adapt the Unified Temporal Discretization Embedding (UTDE) approach described in [101], which combines the mTAND approach described in Section 2.2 with a simpler imputation-based discretization scheme. Specifically, given a set of $t$ observations $\mathbf{x} \in \mathbb{R}^t$ observed at irregular times $\tau \in \mathbb{R}^t$, we a simple imputation scheme to discretize $\mathbf{x}$ into target bins $\gamma$ (e.g., $\gamma = \{0, 1, 2, ..., \gamma\}$). Specifically, given bin value $\gamma_i \in \gamma$, we apply the following rules:

- If there exists a previously observed value of $\mathbf{x}$ (i.e., $\exists \tau \in \tau \, st. \, \tau \leq \gamma_i$), we set the imputed value of $\mathbf{x}$ at time $\gamma$, $\hat{x}_{\gamma_i}$, to the closest previously observed value.
- If no previously observed value exists, we set the value of $\hat{x}_{\gamma_i}$ to the global mean of $\mathbf{x}$.

We do this for each possible vitals/lab, to generate a matrix of imputation embeddings $\mathbf{I} \in \mathbb{R}^{\gamma \times d_{\text{vitals}}}$, were $d_{\text{vitals}}$ is the number of vitals/labs. We then input this embedding into a 1D causal convolutional layer with stride 1 to obtain our final imputation embeddings with hidden dimension $d_h$, $\mathbf{E}_{\text{Imp}} \in \mathbb{R}^{\gamma \times d_h}$.

## E.2 Unifying imputation and mTAND embeddings

We combined simple imputation and mTAND embeddings via a gating function $\mathbf{g}$. Following [101], we let $\mathbf{E}_{\text{mTAND}} \in \mathbb{R}^{\gamma \times d_h}$ denote the mTAND embeddings for vitals/labs derived from the process described in Section 2.2 and let $\mathbf{E}_{\text{Imp}} \in \mathbb{R}^{\gamma \times d_h}$ denote the simple imputations from the process described above. We use each of these discretization embeddings to derive a final set of embeddings for vitals/labs $\mathbf{E}_{\text{vitals}}$ via a one-layer MLP gating function $f$. Specifically, we let $\mathbf{g} = f(\mathbf{E}_{\text{Imp}} \oplus \mathbf{E}_{\text{mTAND}})$, where $\oplus$ denotes the concatenation operator. We then calculate $\mathbf{E}_{\text{vitals}}$ as

$$\mathbf{E}_{\text{vitals}} = \mathbf{g} \odot \mathbf{E}_{\text{Imp}} + (1 - \mathbf{g}) \odot \mathbf{E}_{\text{mTAND}} \in \mathbb{R}^{\gamma \times d_h},$$

where $\odot$ denotes point-wise multiplication.

# F Baseline Comparison

## F.1 MISTS

This approach, from [101], casts time series and clinical notes as multivariate, irregularly-sampled time series (MISTS) and uses layers of self- and cross-attention to fuse modalities. The method uses a Time2Vec [43] encoding scheme to represent the irregularity of observation times. We use the same hyperparameters as in the original paper (e.g., 3 self- and cross-attention blocks, 128-dimensional time embedding, etc.).

## F.2 MulT

This model from [90] relies on multiple stacks of pairwise and bidirectional cross-modal attention blocks (without a self-attention mechanism) to attend to low-level features. The results of cross-modal attention are then sent to modality-specific transformers, concatenated, and used to make predictions.

## F.3 MAG

This method introduces the Multimodal Adaptation Gate (MAG) as an extension to BERT and XL-Net, allowing these pre-trained models to incorporate visual and acoustic data during fine-tuning.

By generating a modality-conditioned shift in their internal representations, MAG enables enhanced sentiment analysis performance on multimodal datasets, achieving human-level accuracy in the field [74].

### F.4 TFN

The proposed Tensor Fusion Network approach (TFN) integrates three core components: Modality Embedding Subnetworks for generating rich embeddings from unimodal inputs, a Tensor Fusion Layer for capturing all levels of modality interactions through a 3-fold Cartesian product, and a Sentiment Inference Subnetwork tailored to perform sentiment analysis based on the fusion layer's output [96].

### F.5 HAIM

The multimodal fusion approach detailed by [83] extracts a single set of features for each ICU stay, and uses this to predict the outcome of interest (in-hospital mortality, etc.). For vitals/lab values, the authors extract a set of 11 generic time series features: signal length, maximum, minimum, mean, median, SD, variance, number of peaks, and average time-series slope and piece-wise change over time of these metrics. This is done independently for each of the 30 events, leading to $30 \times 11 = 330$ vital/lab features per ICU stay. To provide a fair comparison with our method, we only include the most recent five notes and 128 vitals measurements in calculating embeddings. We only include entries for which all modalities are observed. For note/X-ray/ECG embeddings, we compute the mean embedding across all observations occurring during the specified time frame (i.e., the first 48 hours of 48-IHM and LOS, the entire stay for PHE). As with our method, we standardize scale values based on the training set. [83] uses an XGBoost [12] classifier to predict the outcomes of interest. We follow the hyperparameter optimization approach described in the paper. Specifically, we conduct a grid search across the following sets of hyperparameters: max depth $= \{5, 6, 7, 8\}$, number of estimators $= \{200, 300\}$, learning rate $= \{0.3, 0.1, 0.05\}$. Hyperparameters are selected based on the maximum AU-ROC from five-fold cross-validation.

### F.6 Implementation

We integrate F.1 through F.4 into our workflow using the implementation provided by [101]. For F.5, we adapt the time series (e.g., series variance, mean, etc.) feature extraction and model fitting code from the repository released by the corresponding paper. The original paper doesn't use ECG waveforms, so we adopt a similar approach to ECG embeddings as with image and note embeddings, and take the mean value of the latent vector across all included observations.

## G   Computational Resources and Hyper-Parameters

### G.1   Computational Resources

We train models using a Lambda Workstation with four A550 GPUs with 24 GB of memory. We are able to train models using a single GPU. An analysis of computation time and memory requirements is shown in Figure 12.

### G.2   Hyper-Parameters

The set of parameters we used for experiments can be found in Table 8.

## H   Additional Results

### H.1   FlexiModal Experiments

We present additional results comparing FuseMoE to baselines using the MIMIC-III dataset, which includes only vital signs and clinical notes (Table 9), and the MIMIC-IV dataset, featuring vital signs and CXR (Table 10). All experiments utilize the "joint experts and router" configuration. In these settings, FuseMoE demonstrates noticeable advantages.

Table 8: Hyperparameters used for MoE framework and general architecture.

| Hyper-Parameter Type | Parameter Name | Value |
|---|---|---|
| | Number of experts | 16 |
| | FFN hidden size | 512 |
| MoE | Top k | 4 |
| | Disjoint top k | 2 |
| | Hidden activation function | GeLU |
| | Number of MoE layers | 3 |
| | Random seed | [32, 42, 52, 62, 72] |
| | Training epochs | 8 |
| | Training batch size | 2 |
| | Eval batch size | 8 |
| | CNN kernel size | 1 |
| | Gradient accumulation steps | 16 |
| | BERT update epochs | 2 |
| Other Parameters | BERT learning rate | 2.00E-05 |
| | Time series encoder learning rate | 4.00E-04 |
| | Number of notes to include for a patient | 5 |
| | Get notes from beginning or last | Last |
| | Attention embedding dimension | 128 |
| | Number of attention heads | 8 |
| | Maximum time for irregular time series | 48 |
| | Time embedding dimension | 64 |

Table 9: Comparison of `FuseMoE`-based methods (gray) and baselines, utilizing vital signs and clinical notes of MIMIC-III. The best results are highlighted in bold font, and the second-best results are underlined. All results are averaged across 5 random runs. Since HAIM is not designed for the MIMIC-III dataset, we use the concatenation method from e.g. [45] as a replacement.

| Task \ Method | | MISTS | MulT | MAG | TF | Concat | Softmax | Gaussian | Laplace |
|---|---|---|---|---|---|---|---|---|---|
| 48-IHM | AUROC | $89.14 \pm 0.57$ | $87.26 \pm 0.35$ | $86.53 \pm 1.21$ | $87.22 \pm 0.89$ | $86.72 \pm 0.76$ | $90.25 \pm 0.74$ | $\underline{90.77 \pm 0.18}$ | $\mathbf{91.19 \pm 0.52}$ |
| | F1 | $\underline{56.45 \pm 1.30}$ | $54.13 \pm 1.20$ | $53.20 \pm 2.13$ | $51.44 \pm 0.66$ | $52.77 \pm 0.70$ | $56.41 \pm 0.98$ | $56.21 \pm 0.17$ | $\mathbf{57.36 \pm 0.73}$ |
| 25-PHE | AUROC | $86.06 \pm 0.06$ | $85.96 \pm 0.07$ | $85.94 \pm 0.07$ | $84.74 \pm 0.16$ | $85.94 \pm 0.21$ | $\underline{86.41 \pm 0.75}$ | $85.96 \pm 0.64$ | $\mathbf{86.72 \pm 0.27}$ |
| | F1 | $54.84 \pm 0.31$ | $54.20 \pm 0.33$ | $53.73 \pm 0.37$ | $49.84 \pm 0.83$ | $53.30 \pm 0.35$ | $55.02 \pm 0.23$ | $\underline{55.29 \pm 0.45}$ | $\mathbf{55.38 \pm 0.69}$ |

## H.2 Ablation Study on FuseMoE Building Blocks

In Figure 7, we evaluate the impact of various irregularity encoders on the performance of the FuseMoE framework. Our baseline approaches include the following methods:

1. employing only the imputation (discretization) module from the time-series irregularity encoder, as detailed in Appendix E

2. utilizing solely the mTAND module [82] within the time-series irregularity encoder

3. implementing the SeFT method [31] as an irregularity encoder

4. adopting the RAINDROP method [100] as an irregularity encoder

Table 10: Comparison of `FuseMoE`-based methods (gray) and baselines, utilizing vital signs and CXR of MIMIC-IV. The best results are highlighted in bold font, and the second-best results are underlined. All results are averaged across 5 random experiments.

| Task \ Method | | MISTS | MulT | MAG | TF | HAIM | Softmax | Gaussian | Laplace |
|---|---|---|---|---|---|---|---|---|---|
| 48-IHM | AUROC | $81.36 \pm 0.24$ | $77.70 \pm 0.44$ | $81.19 \pm 1.25$ | $76.92 \pm 0.65$ | $80.87 \pm 0.00$ | $\underline{82.08 \pm 0.26}$ | $81.26 \pm 0.18$ | $\mathbf{82.97 \pm 0.49}$ |
| | F1 | $43.35 \pm 0.39$ | $28.40 \pm 0.75$ | $39.59 \pm 0.43$ | $\underline{46.59 \pm 0.33}$ | $40.88 \pm 0.00$ | $38.14 \pm 0.31$ | $44.59 \pm 0.24$ | $\mathbf{47.48 \pm 0.23}$ |
| LOS | AUROC | $82.07 \pm 0.82$ | $81.94 \pm 0.26$ | $81.86 \pm 0.76$ | $81.47 \pm 0.89$ | $81.69 \pm 0.00$ | $\underline{82.96 \pm 0.47}$ | $82.74 \pm 0.85$ | $\mathbf{83.22 \pm 0.68}$ |
| | F1 | $74.07 \pm 0.18$ | $74.46 \pm 0.17$ | $73.89 \pm 0.93$ | $73.39 \pm 0.14$ | $72.93 \pm 0.00$ | $\mathbf{75.67 \pm 0.59}$ | $75.16 \pm 0.42$ | $\underline{75.43 \pm 0.19}$ |
| 25-PHE | AUROC | $\mathbf{71.50 \pm 0.22}$ | $71.20 \pm 0.76$ | $70.89 \pm 0.47$ | $70.55 \pm 0.29$ | $63.43 \pm 0.00$ | $71.38 \pm 0.31$ | $70.87 \pm 0.67$ | $\underline{71.44 \pm 0.24}$ |
| | F1 | $33.52 \pm 0.39$ | $32.80 \pm 0.18$ | $33.14 \pm 0.61$ | $33.56 \pm 0.74$ | $\mathbf{42.45 \pm 0.00}$ | $33.49 \pm 0.15$ | $31.94 \pm 0.09$ | $\underline{34.13 \pm 0.56}$ |

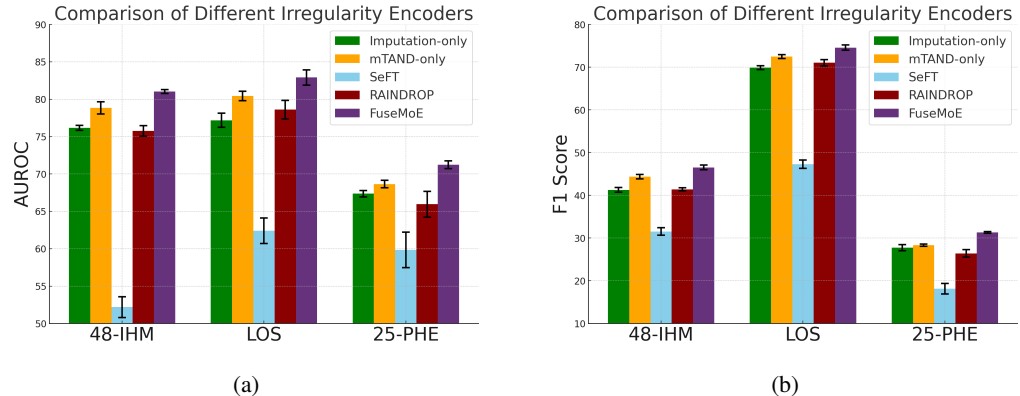

(a)                            (b)

Figure 7: The irregularity encoder employed by `FuseMoE` achieves the best average results compared with 4 baseline irregularity encoders. The performance of these approaches is averaged over 5 random runs. We utilized the vital signs and clinical notes components of the MIMIC-IV dataset.

In Figure 8, we evaluate the impact of various time-series encoders on the performance of the Fuse-MoE framework. The original FuseMoE framework feeds time-series embeddings obtained from the irregularity encoder into the Transformer [92] and extracts the last hidden states of the Transformer output to pass through fully connected layers to make predictions. Our baseline approaches include CNN [47] and LSTM [30] to encode time-series embeddings from the irregularity encoder.

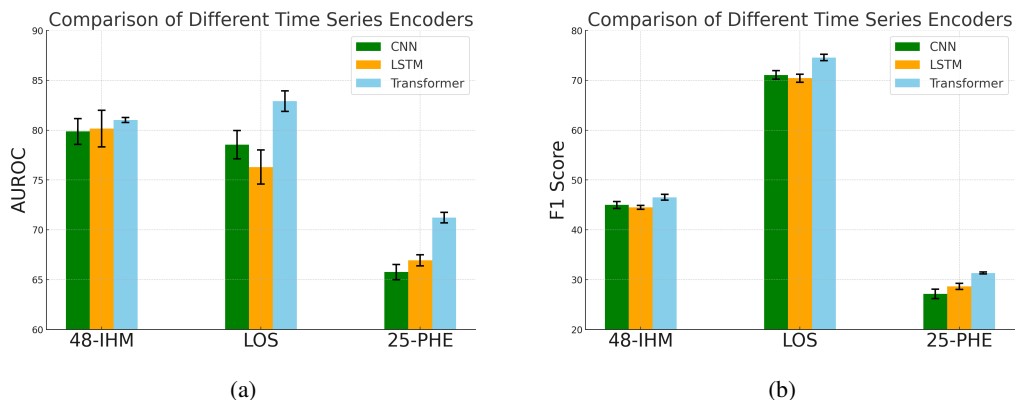

(a)                            (b)

Figure 8: Transformer is more effective in acting as the time-series encoder than CNN and LSTM. The performance outcomes of these approaches are derived from averages over 5 random runs. We utilized the vital signs and clinical notes components of the MIMIC-IV dataset.

In Figure 9, we assess the effect of different CXR encoders on the FuseMoE framework. Currently, the FuseMoE framework incorporates DenseNet-121 as the feature extractor for CXR images before their integration into the mTAND module. This setup is compared with the application of the state-of-the-art vision transformer (ViT-B) [17] as an alternative CXR encoder.

In Figure 10, we evaluate the influence of text encoders on the FuseMoE framework. Currently, FuseMoE incorporates Clinical-Longformer [49] as the text encoder before integrating it into the mTAND module. This setup is compared with other state-of-the-art text encoders: GRU-D [10], FT-LSTM [99], and HierTrans [68].

Finally, in Figure 11, we investigate the effect of the mTAND module on each modality, while we removed mTAND for a particular modality, the rest of FuseMoE's components remained constant.

## H.3   Ablation Study on MoE Architecture

We then conducted ablation studies to explore the efficiency and effectiveness of MoE architecture on model performance. We mainly use MIMIC-IV as our test bed. Figure 12(a) examines the computational efficiency and resource utilization, positioning `FuseMoE` approximately in the middle of

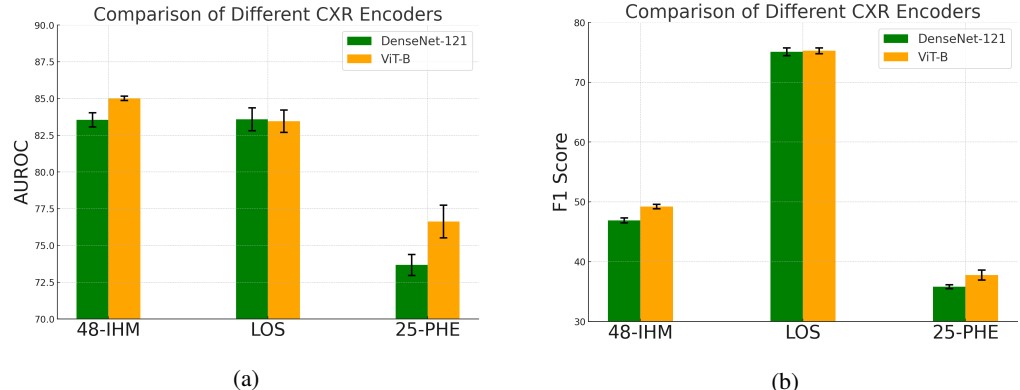

(a)            (b)

Figure 9: ViT could further improve predictive results in some tasks by providing better CXR embeddings. The performance of these approaches is averaged over 5 random runs. We utilized all 4 modalities (vital signs, clinical notes, CXR, and ECG) of MIMIC-IV. Note that while we vary the CXR encoders, the rest of our framework's components remain constant.

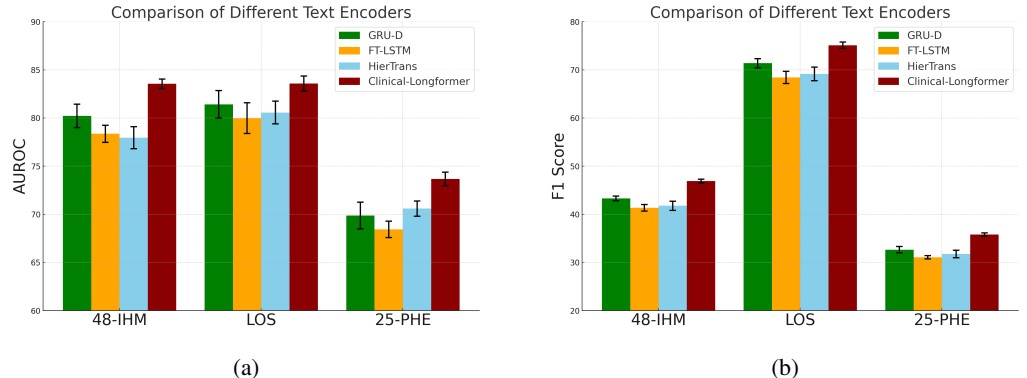

(a)            (b)

Figure 10: Clinical-Longformer as the text encoder achieves the best performance compared with baselines. The performance of these approaches is derived from averages over 5 random runs. We utilized all 4 modalities (vital signs, clinical notes, CXR, and ECG) of the MIMIC-IV dataset, while we varied the text encoders, the rest of our framework's components remained constant.

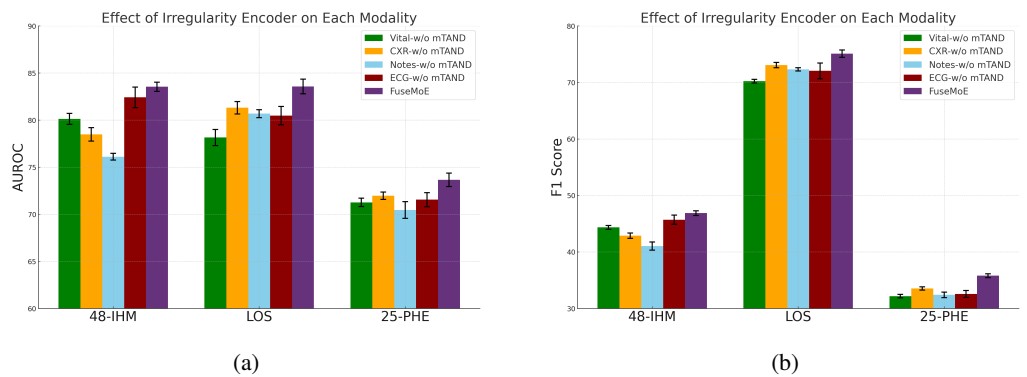

(a)            (b)

Figure 11: Encoding irregularity using the mTAND module improves the overall performance. The positive effect of the irregularity encoder is most evident in vital signs and clinical notes. The performance outcomes of these approaches are averaged over 5 random runs. We utilized all 4 modalities (vital signs, clinical notes, CXR, and ECG) components of the MIMIC-IV dataset.

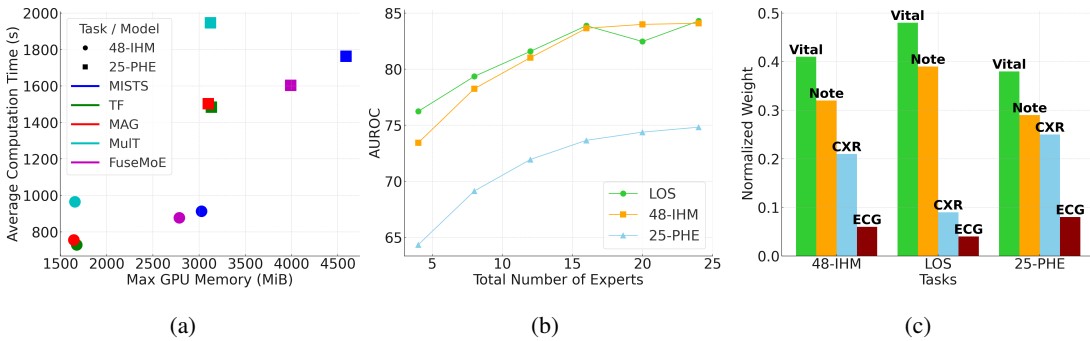

(a)               (b)               (c)

Figure 12: Results of ablation studies on MoE architecture: (a) The computational efficiency and resource utilization of each method when applied to vital signs and clinical notes from the MIMIC-IV dataset; (b) The relationship between the number of experts and task performance across different modalities, including vital signs, clinical notes, and CXR; (c) The impact of each modality on the top-$k$ experts within a disjoint router structure.

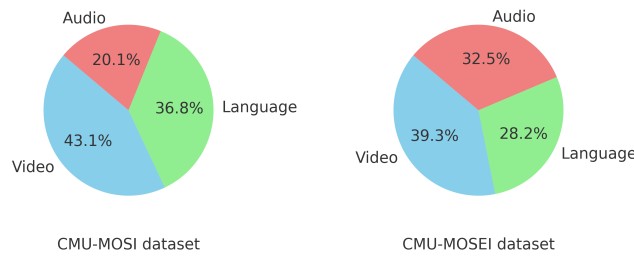

Figure 13: Modality weight composition of CMU-MOSI and CMU-MOSEI datasets.

the comparison. Despite the increase in model parameters due to the incorporation of the MoE layer, its sparse nature does not significantly escalate the computational load. Figure 12(b) illustrates the correlation between the number of experts and task performance across different modalities. Generally, performance improves with the addition of more experts, plateauing once the count exceeds 16. To achieve a compromise between performance and computational expense, we opted to utilize the top 4 experts out of 16 in our experiments. Figure 12(c) and Figure 13 study the influence of each modality on the top-$k$ chosen experts. For every expert selected, we calculate the number of samples that include a specific modality, weighted by corresponding weight factors from the gating functions. The outcomes are subsequently normalized across modalities. The analysis of Figure 12(c) reveals that predictions across all tasks heavily depend on vital signs and clinical notes. This reliance is attributed to the abundant samples in these two modalities. Despite the notably smaller quantity of CXR, they play more significant roles in the 25-PHE and 48-IHM tasks, which aligns with our findings in Table 5. The results in Figure 13 demonstrate that the modality weight distribution in the MOSI and MOSEI datasets is more "spread out", with the audio component carrying a greater weight in the MOSEI dataset.

# I   Details on Numerical Experiments

We conduct multiple numerical experiments to illustrate the theoretical convergence rates of the MLE $\widehat{G}_n$ to the true mixing measure $G_*$ under both exact-specified and over-specified settings.

## I.1   Experimental Setup

**Synthetic Data.** Assume that the true mixing measure $G_* = \sum_{i=1}^{k_*} \exp(\beta_i^*) \delta_{(W_i^*, a_i^*, b_i^*, \nu_i^*)}$ is of order $k_* = 2$. The true parameters for the router, $(W_i^*, \beta_i^*) \in \mathbb{R}^d \times \mathbb{R}$, are drawn independently from an isotropic Gaussian distribution with zero mean and variance $\sigma_r^2 = 0.01/d$ for $1 \le i \le 6$, and otherwise are set to zero. Similarly, the true parameters of the experts, $(a_i^*, b_i^*) \in \mathbb{R}^d \times \mathbb{R}$, are drawn independently of an isotropic Gaussian distribution with zero mean and variance $\sigma_e^2 = 1/d$

for all experts. For the variances $\nu_i^*$, we also sample from the Gaussian distribution $\mathcal{N}(0, \sigma_e^2)$, and then take the absolute value of the sample. These parameters remain unchanged for all experiments.

Then, we generate i.i.d samples $\{(X_i, Y_i)\}_{i=1}^n$ by first sampling $X_i$'s from the uniform distribution Uniform$[0, 1]$ and then sampling $Y_i$'s from the true conditional density $g_{G_*}(Y|X)$ of the Laplace gating Gaussian mixture of experts (MoE) given in equation 4.

**Maximum Likelihood Estimation (MLE).** A popular approach to determining the MLE $\widehat{G}_n$ for each set of samples is to use the EM algorithm [15]. However, since there are not any closed-form expressions for updating the gating parameters $\beta_{0i}, \beta_{1i}$ in the maximization steps, we have to leverage an EM-based numerical scheme, which was previously used in [63]. In particular, we utilize a simple coordinate gradient descent algorithm in the maximization steps. Additionally, we select the convergence criterion of $\epsilon = 10^{-6}$ and run a maximum of 2000 EM iterations.

**Initialization.** For each $k \in \{k_*, k_* + 1\}$, we randomly distribute elements of the set $\{1, 2, ..., k\}$ into $k_*$ different Voronoi cells $\mathcal{A}_1, \mathcal{A}_2, \ldots, \mathcal{A}_{k_*}$, each contains at least one element. Moreover, we repeat this process for each replication. Subsequently, for each $j \in [k_*]$, we initialize parameters $W_i$ by sampling from a Gaussian distribution centered around its true counterpart $W_j^*$ with a small variance, where $i \in \mathcal{A}_j$. Other parameters $\beta_i, a_i, b_i, \nu_i$ are also initialized in a similar fashion.

## I.2  Exact-specified Setting

Under the exact-specified settings, we conduct 5 sample generations for each configuration, across a spectrum of 10 different sample sizes $n$ ranging from $10^3$ to $10^5$. It can be seen from Figure 3 (left) that the MLE $\widehat{G}_n$ empirically converges to the true mixing measure $G_*$ under the Voronoi metric $\mathcal{D}_1$ at the rate of order $\mathcal{O}(n^{-0.49})$, which matches the theoretical parametric convergence rate established in Theorem J.2.

## I.3  Over-specified Setting

Under the over-specified settings, we continue to generate 5 samples of size $n$ for each setting, given 10 different choices of sample size $n \in [10^3, 10^5]$. From Figure 3 (right), we observe that the MLE $\widehat{G}_n$ empirically converges to $G_*$ under the Voronoi metric $\mathcal{D}_2$ at the rate of order $\mathcal{O}(n^{-0.47})$, which aligns with the theoretical parametric convergence rate established in Theorem 3.2.

# J  Exact-Specified Setting

In this appendix, we study the theoretical behaviors of the MLE under the exact-specified setting, i.e., $k = k_*$, of the Laplace gating Gaussian MoE. We demonstrate that under the exact-specified setting, the rate of estimated conditional density function $p_{\widehat{G}_n}$ to $p_{G_*}$ is parametric $\mathcal{O}n^{-1/2})$ (up to some logarithmic factor).

**Theorem J.1.** *The density estimation $p_{\widehat{G}_n}(Y|X)$ converges to the true density $p_{G_*}(Y|X)$ under the Total Variation distance $V$ at the following rate:*

$$\mathbb{E}_X[V(p_{\widehat{G}_n}(\cdot|X), p_{G_*}(\cdot|X))] = \mathcal{O}(\sqrt{\log(n)/n}).$$

The proof of Theorem J.1 can be done similarly to that of Theorem 3.1 in Appendix K.2. The result of Theorem J.1 indicates that as long as we can establish the lower bound of the total variation distance between $p_{\widehat{G}_n}$ and $p_{G_*}$ based on certain loss function between the MLE $\widehat{G}_n$ and the true mixing measure $G_*$, we directly achieve the rate of the MLE under that loss function.

**Voronoi Loss** We now define that loss function between the MLE and the true mixing measure for the exact-specified setting:

$$\mathcal{D}_1(G, G_*) := \sum_{j=1}^{k_*} \Big| \sum_{i \in \mathcal{A}_j} \exp(\beta_i) - \exp(\beta_j^*) \Big| + \sum_{j \in [k_*]:|\mathcal{A}_j|=1} \sum_{i \in \mathcal{A}_j} \exp(\beta_i) \Phi_{ij}(1, 1, 1, 1). \quad (9)$$

Above, for any $(\rho_1, \rho_2, \rho_3, \rho_4) \in \mathbb{R}^4$, we define $\Phi_{ij}(\rho_1, \rho_2, \rho_3, \rho_4) = \|W_i - W_j^*\|^{\rho_1} + \|a_i - a_j^*\|^{\rho_2} + |b_i - b_j^*|^{\rho_3} + |\nu_i - \nu_j^*|^{\rho_4}$ for any $i \in \mathcal{A}_j$ and $j \in [k_*]$. We demonstrate in the following theorem that

the rate of MLE to the true mixing measure under the Voronoi loss function $\mathcal{D}_1$ is $\mathcal{O}(n^{-1/2})$ (up to some logarithmic factor).

**Theorem J.2** (Exact-specified setting). *When $k = k_*$ is known, the following Total Variation bound holds guarantetrue for any $G \in \mathcal{G}_k(\Theta)$:*

$$\mathbb{E}_X[V(p_G(\cdot|X), p_{G_*}(\cdot|X))] \gtrsim \mathcal{D}_1(G, G_*).$$

*Therefore, we have $\mathcal{D}_1(\widehat{G}_n, G_*) = \mathcal{O}(\sqrt{\log(n)/n})$.*

Proof of Theorem J.2 is in Appendix K.1. The convergence rate of MLE under the Voronoi loss function $\mathcal{D}_1$ implies that the rates of estimating the true parameters $W_i^*, a_i^*, b_i^*, \nu_i^*$ are also $\mathcal{O}(n^{-1/2})$ (up to logarithmic factors). These rates are comparable to those under the exact-specified setting of softmax gating Gaussian MoE (cf. Theorem 1 in [64]).

# K    Proof of Theoretical Results

In this appendix, we provide proofs for all theoretical results in the paper. Throughout this appendix, for any vector $v \in \mathbb{R}^d$ and $\alpha := (\alpha_1, \alpha_2, \ldots, \alpha_d) \in \mathbb{N}^d$, we denote $v^\alpha = v_1^{\alpha_1} v_2^{\alpha_2} \ldots v_d^{\alpha_d}$, $|v| := v_1 + v_2 + \ldots + v_d$ and $\alpha! := \alpha_1! \alpha_2! \ldots \alpha_d!$.

## K.1    Proof of Theorem J.2

First of all, we need to establish the following bound:

$$\mathbb{E}_X[V(p_G(\cdot|X), p_{G_*}(\cdot|X))] \gtrsim \mathcal{D}_1(G, G_*).$$

For that sake, it is sufficient to demonstrate two following inequalities:

- **Inequality A.** $\inf_{G \in \mathcal{G}_{k_*}(\Theta): \mathcal{D}_1(G, G_*) \leq \varepsilon'} \dfrac{\mathbb{E}_X[V(p_G(\cdot|X), p_{G_*}(\cdot|X))]}{\mathcal{D}_1(G, G_*)} > 0$;

- **Inequality B.** $\inf_{G \in \mathcal{G}_{k_*}(\Theta): \mathcal{D}_1(G, G_*) > \varepsilon'} \dfrac{\mathbb{E}_X[V(p_G(\cdot|X), p_{G_*}(\cdot|X))]}{\mathcal{D}_1(G, G_*)} > 0$,

for some constant $\varepsilon' > 0$.

**Proof of inequality A**: The inequality A is equivalent to

$$\lim_{\varepsilon \to 0} \inf_{G \in \mathcal{G}_{k_*}(\Theta): \mathcal{D}_1(G, G_*) \leq \varepsilon} \frac{\mathbb{E}_X[V(p_G(\cdot|X), p_{G_*}(\cdot|X))]}{\mathcal{D}_1(G, G_*)} > 0.$$

Assume that the above inequality is not true, then, there exists a sequence of mixing measure $G_n := \sum_{i=1}^{k_*} \exp(\beta_i^n) \delta_{(W_i^n, a_i^n, b_i^n, \nu_i^n)} \in \mathcal{G}_{k_*}(\Theta)$ such that both $\mathcal{D}_1(G_n, G_*)$ and $\mathbb{E}_X[V(p_{G_n}(\cdot|X), p_{G_*}(\cdot|X))]/\mathcal{D}_1(G_n, G_*)$ go to zero as $n \to \infty$. Now, we define

$$\mathcal{A}_j^n = \mathcal{A}_j(G_n) := \{i \in [k_*] : \|\theta_i^n - \theta_j^*\| \leq \|\theta_i^n - \theta_\tau^*\|, \ \forall \tau \neq j\},$$

for any $j \in [k_*]$ as Voronoi cells with respect to the mixing measure $G_n$, where we denote $\theta_i^n := (W_i^n, a_i^n, b_i^n, \nu_i^n)$ and $\theta_j^* := (W_j^*, a_j^*, b_j^*, \nu_j^*)$. In this proof, since our arguments are assymptotic, we can assume without loss of generality (WLOG) that these Voronoi cells does not depend on $n$, that is, $\mathcal{A}_j = \mathcal{A}_j^n$. Next, it follows from the hypothesis $\mathcal{D}_{1n} := \mathcal{D}_1(G_n, G_*) \to 0$ as $n \to \infty$ that each Voronoi cell contains only one element. Therefore, we may assume WLOG that $\mathcal{A}_j = \{j\}$ for any $j \in [k_*]$, which implies that $(W_j^n, a_j^n, b_j^n, \nu_j^n) \to (W_j^*, a_j^*, b_j^*, \nu_j^*)$ and $\exp(\beta_j^n) \to \exp(\beta_j^*)$ as $n \to \infty$. Then, the loss function between $G_n$ and $G_*$ is given by

$$\mathcal{D}_1(G_n, G_*) = \sum_{i=1}^{k_*} \left[ \exp(\beta_i^n)\left( \|\Delta W_i^n\| + \|\Delta a_i^n\| + \|\Delta b_i^n\| + \|\Delta \nu_i^n\| \right) + \left| \exp(\beta_i^n) - \exp(\beta_i^*) \right| \right],$$

where we denote $\Delta \beta_{1i}^n := \beta_{1i}^n - \beta_{1i}^*$, $\Delta a_i^n := a_i^n - a_i^*$, $\Delta b_i^n := b_i^n - b_i^*$ and $\Delta \nu_i^n := \nu_i^n - \nu_i^*$.

Now, we break the rest of our arguments into three steps:

**Stage 1 - Density decomposition**:

In this step, we aim to decompose the term $Q_n := \left[ \sum_{i=1}^{k_*} \exp(-\|W_i^* - X\| + \beta_i^*) \right] \cdot [p_{G_n}(Y|X) - p_{G_*}(Y|X)]$, which can be represented as follows:

$$
\begin{aligned}
Q_n = & \sum_{i=1}^{k_*} \exp(\beta_i^n) \Big[ F(Y|X; W_i^n, a_i^n, b_i^n, \nu_i^n) - F(Y|X; W_i^*, a_i^*, b_i^*, \nu_i^*) \Big] \\
& - \sum_{i=1}^{k_*} \exp(\beta_i^n) \Big[ H(Y|X; W_i^n) - H(Y|X; W_i^*) \Big] \\
& + \sum_{i=1}^{k_*} \Big[ \exp(\beta_i^n) - \exp(\beta_i^*) \Big] \Big[ F(Y|X; W_i^*, a_i^*, b_i^*, \nu_i^*) - H(Y|X, W_i^*) \Big] \\
:= & A_n - B_n + E_n,
\end{aligned}
\tag{10}
$$

where we denote $F(Y|X; W, a, b, \nu) := \exp(-\|W - X\|) f(Y|a^\top X + b, \nu)$ and $H(Y|X; W) = \exp(-\|W - X\|) p_{G_n}(Y|X)$. By applying the first-order Taylor expansion, we can rewrite $A_n$ as

$$
\begin{aligned}
A_n = & \sum_{i=1}^{k_*} \sum_{|\alpha|=1} \frac{\exp(\beta_i^n)}{\alpha!} \cdot (\Delta W_i^n)^{\alpha_1} (\Delta a_i^n)^{\alpha_2} (\Delta b_i^n)^{\alpha_3} (\Delta \nu_i^n)^{\alpha_4} \\
& \qquad\qquad \times \frac{\partial^{|\alpha_1|+|\alpha_2|+\alpha_3+\alpha_4} F}{\partial W^{\alpha_1} \partial a^{\alpha_2} \partial b^{\alpha_3} \partial \nu^{\alpha_4}}(Y|X; W_i^*, a_i^*, b_i^*, \nu_i^*) + R_1(X, Y) \\
= & \sum_{i=1}^{k_*} \sum_{|\alpha|=1} \frac{\exp(\beta_i^n)}{\alpha!} \cdot (\Delta W_i^n)^{\alpha_1} (\Delta a_i^n)^{\alpha_2} (\Delta b_i^n)^{\alpha_3} (\Delta \nu_i^n)^{\alpha_4} \\
& \qquad\qquad \times \frac{\partial^{|\alpha_1|} g}{\partial W^{\alpha_1}}(X; W_i^*) \cdot \frac{\partial^{|\alpha_2|+\alpha_3+\alpha_4} f}{\partial a^{\alpha_2} \partial b^{\alpha_3} \partial \nu^{\alpha_4}}(Y|(a_i^*)^\top X + b_i^*, \nu_i^*) + R_1(X, Y),
\end{aligned}
$$

where $R_1(X, Y)$ is a Taylor remainder that satisfies $R_1(X, Y)/\mathcal{D}_1(X, Y) \to 0$ as $n \to \infty$ and $g(X, W) := \exp(\|W - X\|)$. Recall that $f$ is the univariate Gaussian density, then by denoting $h_1(X; a, b) := a^\top X + b$, we can verify that

$$
\frac{\partial^{\alpha_4} f}{\partial \nu^{\alpha_4}}(Y|(a_i^*)^\top X + b_i^*, \nu_i^*) = \frac{1}{2^{\alpha_4}} \cdot \frac{\partial^{2\alpha_4} f}{\partial h_1^{2\alpha_4}}(Y|(a_i^*)^\top X + b_i^*, \nu_i^*).
$$

Consequently, we get

$$
\begin{aligned}
A_n = & \sum_{i=1}^{k_*} \sum_{|\alpha|=1} \frac{\exp(\beta_i^n)}{2^{\alpha_4} \alpha!} \cdot (\Delta W_i^n)^{\alpha_1} (\Delta a_i^n)^{\alpha_2} (\Delta b_i^n)^{\alpha_3} (\Delta \nu_i^n)^{\alpha_4} \\
& \qquad\qquad \times X^{\alpha_2} \cdot \frac{\partial^{|\alpha_1|} g}{\partial W^{\alpha_1}}(X; W_i^*) \cdot \frac{\partial^{|\alpha_2|+\alpha_3+2\alpha_4} f}{\partial h_1^{|\alpha_2|+\alpha_3+2\alpha_4}}(Y|(a_i^*)^\top X + b_i^*, \nu_i^*) + R_1(X, Y) \\
= & \sum_{i=1}^{k_*} \sum_{|\alpha_1|=0}^{1} \sum_{|\alpha_2|=0}^{1-|\alpha_1|} \sum_{\eta=0}^{2(1-|\alpha_1|-|\alpha_2|)} \sum_{\substack{\alpha_3+2\alpha_4=\eta, \\ 0 \le \alpha_3+\alpha_4 \le 1-|\alpha_1|-|\alpha_2|}} \frac{\exp(\beta_i^n)}{2^{\alpha_4} \alpha!} \cdot (\Delta W_i^n)^{\alpha_1} (\Delta a_i^n)^{\alpha_2} (\Delta b_i^n)^{\alpha_3} (\Delta \nu_i^n)^{\alpha_4} \\
& \qquad\qquad \times X^{\alpha_2} \cdot \frac{\partial^{|\alpha_1|} g}{\partial W^{\alpha_1}}(X; W_i^*) \cdot \frac{\partial^{|\alpha_2|+\eta} f}{\partial h_1^{|\alpha_2|+\eta}}(Y|(a_i^*)^\top X + b_i^*, \nu_i^*) + R_1(X, Y),
\end{aligned}
\tag{11}
$$

where we denote $\eta = \alpha_3 + 2\alpha_4 \in \mathbb{N}$.

Subsequently, we also apply the first-order Taylor expansion to the term $B_n$ defined in equation 10 and get that

$$B_n = \sum_{i=1}^{k_*} \sum_{|\gamma|=1} \frac{\exp(\beta_i^n)}{\gamma!} (\Delta W_i^n)^\gamma \cdot \frac{\partial^{|\gamma|} H}{\partial W^\gamma} (Y|X; W_i^*) + R_2(X, Y)$$

$$= \sum_{i=1}^{k_*} \sum_{|\gamma|=1} \frac{\exp(\beta_i^n)}{\gamma!} (\Delta W_i^n)^\gamma \cdot \frac{\partial^{|\gamma|} g}{\partial W^\gamma} (X; W_i^*) p_{G_n}(Y|X) + R_2(X, Y), \quad (12)$$

where $R_2(X, Y)$ is a Taylor remainder such that $R_2(X, Y)/\mathcal{D}_1(G_n, G_*) \to 0$ as $n \to \infty$.

From the above results, the term $Q_n$ can be rewritten as

$$Q_n = \sum_{i=1}^{k_*} \sum_{|\alpha_1|=0}^{1} \sum_{|\alpha_2|=0}^{1-|\alpha_1|} \sum_{\eta=0}^{2(1-|\alpha_1|-|\alpha_2|)} S_{i,\alpha_1,\alpha_2,\eta}^n \cdot X^{\alpha_2} \cdot \frac{\partial^{|\alpha_1|} g}{\partial W^{\alpha_1}} (X; W_i^*) \cdot \frac{\partial^{|\alpha_2|+\eta} f}{\partial h_1^{|\alpha_2|+\eta}} (Y|(a_i^*)^\top X + b_i^*, \nu_i^*)$$

$$+ \sum_{i=1}^{k_*} \sum_{|\gamma|=0}^{1} T_{i,\gamma}^n \cdot \frac{\partial^{|\gamma|} g}{\partial W^\gamma} (X; W_i^*) p_{G_n}(Y|X) + R_1(X, Y) + R_2(X, Y), \quad (13)$$

in which we respectively define for each $i \in [k_*]$ that

$$S_{i,\alpha_1,\alpha_2,\eta}^n := \sum_{\substack{\alpha_3+2\alpha_4=\eta, \\ 0 \le \alpha_3+\alpha_4 \le 1-|\alpha_1|-|\alpha_2|}} \frac{\exp(\beta_i^n)}{2^{\alpha_4} \alpha!} \cdot (\Delta W_i^n)^{\alpha_1} (\Delta a_i^n)^{\alpha_2} (\Delta b_i^n)^{\alpha_3} (\Delta \nu_i^n)^{\alpha_4},$$

$$T_{i,\gamma}^n := \frac{\exp(\beta_i^n)}{\gamma!} (\Delta W_i^n)^\gamma,$$

for any $(\alpha_1, \alpha_2, \eta) \neq (\mathbf{0}_d, \mathbf{0}_d, 0)$ and $\gamma \neq \mathbf{0}_d$. Otherwise, $S_{i,\mathbf{0}_d,\mathbf{0}_d,0}^n = T_{i,\mathbf{0}_d}^n := \exp(\beta_i^n) - \exp(\beta_i^*)$.

**Stage 2 - Non-vanishing coefficients**:

Moving to the second step, we will show that not all the ratios $S_{i,\alpha_1,\alpha_2,\eta}^n/\mathcal{D}_1(G_n, G_*)$ and $T_{i,\gamma}^n/\mathcal{D}_1(G_n, G_*)$ tend to zero as $n \to \infty$. Assume by contrary that all of them approach zero when $n \to \infty$, then for $(\alpha_1, \alpha_2, \eta) = (\mathbf{0}_d, \mathbf{0}_d, 0)$, it follows that

$$\frac{1}{\mathcal{D}_1(G_n, G_*)} \cdot \sum_{i=1}^{k_*} \left| \exp(\beta_i^n) - \exp(\beta_i^*) \right| = \sum_{i=1}^{k_*} \frac{|S_{i,\alpha_1,\alpha_2,\eta}^n|}{\mathcal{D}_1(G_n, G_*)} \to 0. \quad (14)$$

Additionally, for tuples $(\alpha_1, \alpha_2, \eta)$ where $\alpha_1 \in \{e_1, e_2, \ldots, e_d\}$ with $e_j := (0, \ldots, 0, \underbrace{1}_{j-th}, 0, \ldots, 0)$, $\alpha_2 = \mathbf{0}_d$ and $\eta = 0$, we get

$$\frac{1}{\mathcal{D}_1(G_n, G_*)} \cdot \sum_{i=1}^{k_*} \exp(\beta_i^n) \|\Delta W_i^n\|_1 = \sum_{i=1}^{k_*} \frac{|S_{i,\alpha_1,\alpha_2,\eta}^n|}{\mathcal{D}_1(G_n, G_*)} \to 0.$$

For $(\alpha_1, \alpha_2, \eta)$ where $\alpha_1 = \mathbf{0}_d$, $\alpha_2 \in \{e_1, e_2, \ldots, e_d\}$ and $\eta = 0$, we have

$$\frac{1}{\mathcal{D}_1(G_n, G_*)} \cdot \sum_{i=1}^{k_*} \exp(\beta_i^n) \|\Delta a_i^n\|_1 = \sum_{i=1}^{k_*} \frac{|S_{i,\alpha_1,\alpha_2,\eta}^n|}{\mathcal{D}_1(G_n, G_*)} \to 0.$$

For $(\alpha_1, \alpha_2, \eta)$ where $\alpha_1 = \alpha_2 = \mathbf{0}_d$ and $\eta = 1$, we have

$$\frac{1}{\mathcal{D}_1(G_n, G_*)} \cdot \sum_{i=1}^{k_*} \exp(\beta_i^n) \|\Delta b_i^n\|_1 = \sum_{i=1}^{k_*} \frac{|S_{i,\alpha_1,\alpha_2,\eta}^n|}{\mathcal{D}_1(G_n, G_*)} \to 0.$$

For $(\alpha_1, \alpha_2, \eta)$ where $\alpha_1 = \alpha_2 = \mathbf{0}_d$ and $\eta = 2$, we have

$$\frac{1}{\mathcal{D}_1(G_n, G_*)} \cdot \sum_{i=1}^{k_*} \exp(\beta_i^n) \|\Delta \nu_i^n\|_1 = \sum_{i=1}^{k_*} \frac{|S_{i,\alpha_1,\alpha_2,\eta}^n|}{\mathcal{D}_1(G_n, G_*)} \to 0.$$

As a result, we achieve that

$$\frac{1}{\mathcal{D}_1(G_n, G_*)} \cdot \sum_{i=1}^{k_*} \exp(\beta_i^n)\Big[\|\Delta W_i^n\|_1 + \|\Delta a_i^n\|_1 + |\Delta b_i^n| + |\Delta \nu_i^n|\Big] \to 0.$$

Due to the topological equivalence between norm-1 and norm-2, the above limit implies that

$$\frac{1}{\mathcal{D}_1(G_n, G_*)} \cdot \sum_{i=1}^{k_*} \exp(\beta_i^n)\Big[\|\Delta W_i^n\| + \|\Delta a_i^n\| + |\Delta b_i^n| + |\Delta \nu_i^n|\Big] \to 0. \tag{15}$$

Combine equation 14 with equation 15, we deduce that $\mathcal{D}_1(G_n, G_*)/\mathcal{D}_1(G_n, G_*) \to 0$, which is a contradiction. Consequently, at least one among the ratios $S_{i,\alpha_1,\alpha_2,\eta}^n/\mathcal{D}_1(G_n, G_*)$ and $T_{i,\gamma}^n/\mathcal{D}_1(G_n, G_*)$ does not vanish as $n$ tends to infinity.

**Stage 3 - Fatou's contradiction**:

In this step, we use the Fatou's lemma to point out a contradiction to the results achieved in Step 2. In particular, we denote by $m_n$ the maximum of the absolute values of $S_{i,\alpha_1,\alpha_2,\eta}^n/\mathcal{D}_1(G_n, G_*)$ and $T_{i,\gamma}^n/\mathcal{D}_1(G_n, G_*)$. Since at least one of the previous ratios does not converge to zero, we deduce that $1/m_n \not\to \infty$.

Recall from the hypothesis that $\mathbb{E}_X[V(p_{G_n}(\cdot|X), p_{G_*}(\cdot|X))]/\mathcal{D}_1(G_n, G_*) \to 0$ as $n \to \infty$. According to the Fatou's lemma, we have

$$0 = \lim_{n\to\infty} \frac{\mathbb{E}_X[V(p_{G_n}(\cdot|X), p_{G_*}(\cdot|X))]}{\mathcal{D}_1(G_n, G_*)} \geq \frac{1}{2} \cdot \int \liminf_{n\to\infty} \frac{|p_{G_n}(Y|X) - p_{G_*}(Y|X)|}{\mathcal{D}_1(G_n, G_*)} \mathrm{d}(X, Y) \geq 0.$$

This result indicates that $|p_{G_n}(Y|X) - p_{G_*}(Y|X)|/\mathcal{D}_1(G_n, G_*)$ tends to zero as $n$ goes to infinity for almost surely $(X, Y)$. As a result, it follows that

$$\lim_{n\to\infty} \frac{Q_n}{m_n \mathcal{D}_1(G_n, G_*)} = \lim_{n\to\infty} \frac{|p_{G_n}(Y|X) - p_{G_*}(Y|X)|}{m_n \mathcal{D}_1(G_n, G_*)} = 0.$$

Next, let us denote $S_{i,\alpha_1,\alpha_2,\eta}^n/[m_n \mathcal{D}_1(G_n, G_*)] \to \xi_{i,\alpha_1,\alpha_2,\eta}$ and $T_{i,\gamma}^n/[m_n \mathcal{D}_1(G_n, G_*)] \to \kappa_{i,\gamma}$ with a note that at least one among them is non-zero. From the formulation of $Q_n$ in equation 13, we deduce that

$$\sum_{i=1}^{k_*} \sum_{|\alpha_1|=0}^{1} \sum_{|\alpha_2|=0}^{1-|\alpha_1|} \sum_{\eta=0}^{2(1-|\alpha_1|-|\alpha_2|)} \xi_{i,\alpha_1,\alpha_2,\eta} \cdot X^{\alpha_2} \cdot \frac{\partial^{|\alpha_1|}g}{\partial W^{\alpha_1}}(X; W_i^*) \cdot \frac{\partial^{|\alpha_2|+\eta}f}{\partial h_1^{|\alpha_2|+\eta}}(Y|(a_i^*)^\top X + b_i^*, \nu_i^*)$$

$$+ \sum_{i=1}^{k_*} \sum_{|\gamma|=0}^{1} \kappa_{i,\gamma} \cdot \frac{\partial^{|\gamma|}g}{\partial W^\gamma}(X; W_i^*) p_{G_n}(Y|X) = 0, \tag{16}$$

for almost surely $(X, Y)$. The above equation is equivalent to

$$\sum_{i=1}^{k_*} \sum_{|\alpha_1|=0}^{1} \left[\sum_{|\alpha_2|=0}^{1-|\alpha_1|} \sum_{\eta=0}^{2(1-|\alpha_1|-|\alpha_2|)} \xi_{i,\alpha_1,\alpha_2,\eta} \cdot X^{\alpha_2} \frac{\partial^{\alpha_2+\eta}f}{\partial h_1^{\alpha_2+\eta}}(Y|(a_i^*)^\top X + b_i^*, \nu_i^*) + \kappa_{i,\alpha_1} p_{G_*}(Y|X)\right]$$

$$\times \frac{\partial^{|\alpha_1|}g}{\partial W^{\alpha_1}}(X; W_i^*) = 0,$$

for almost surely $(X, Y)$. It is worth noting that parameters $W_1^*, \ldots, W_K^*$ are pair-wise distinct, thus, the set $\left\{\frac{\partial^{|\alpha_1|}g}{\partial W^{\alpha_1}}(X; W_i^*) : i \in [k_*], \ 0 \leq |\alpha_1| \leq 1\right\}$ is a linearly independent, which implies that

$$\sum_{|\alpha_2|=0}^{1-|\alpha_1|} \sum_{\eta=0}^{2(1-|\alpha_1|-|\alpha_2|)} \xi_{i,\alpha_1,\alpha_2,\eta} \cdot X^{\alpha_2} \frac{\partial^{\alpha_2+\eta}f}{\partial h_1^{\alpha_2+\eta}}(Y|(a_i^*)^\top X + b_i^*, \nu_i^*) + \kappa_{i,\alpha_1} p_{G_*}(Y|X) = 0,$$

for any $i \in [k_*]$, $0 \leq |\alpha_1| \leq 1$ for almost surely $(X, Y)$. Moreover, since $(a_1^*, b_1^*, \nu_1^*), \ldots, (a_{k_*}^*, b_{k_*}^*, \nu_{k_*}^*)$ have pair-wise distinct values, those of $((a_1^*)^\top X + b_1^*, \nu_1^*), \ldots, ((a_{k_*}^*)^\top X + b_{k_*}^*, \nu_{k_*}^*)$ are also pair-wise different. Therefore, the set

$$\Big\{ X^{\alpha_2} \frac{\partial^{\alpha_2 + \eta} f}{\partial h_1^{\alpha_2 + \eta}} (Y|(a_i^*)^\top X + b_i^*, \nu_i^*), \ p_{G_*}(Y|X) :$$

$$0 \leq |\alpha_2| \leq 1 - |\alpha_1|, \ 0 \leq \eta \leq 2(1 - |\alpha_1| - |\alpha_2|) \Big\}$$

is also linearly independent. Consequently, we obtain that $\xi_{i,\alpha_1,\alpha_2,\eta} = \kappa_{i,\gamma} = 0$ for any $i \in [k_*]$, $0 \leq |\alpha_1| + \alpha_2 \leq 1$, $0 \leq \eta \leq 2(1 - |\alpha_1| - |\alpha_2|)$ and $0 \leq |\gamma| \leq 1$, which contradicts the fact that at least one among those terms is different from zero.

Hence, we can find some constant $\varepsilon' > 0$ such that

$$\inf_{G \in \mathcal{G}_{k_*}(\Theta) : \mathcal{D}_1(G, G_*) \leq \varepsilon'} \frac{\mathbb{E}_X[V(p_G(\cdot|X), p_{G_*}(\cdot|X))]}{\mathcal{D}_1(G, G_*)} > 0.$$

**Proof of inequality B**: Assume by contrary that the inequality B does not hold, then there exists a sequence of mixing measures $G_n' \in \mathcal{G}_{k_*}(\Theta)$ such that $\mathcal{D}_1(G_n', G_*) > \varepsilon'$ and

$$\lim_{n \to \infty} \frac{\mathbb{E}_X[V(p_{G_n'}(\cdot|X), p_{G_*}(\cdot|X))]}{\mathcal{D}_1(G_n', G_*)} = 0.$$

This result leads to $\mathbb{E}_X[V(p_{G_n'}(\cdot|X), p_{G_*}(\cdot|X))] \to 0$ as $n \to \infty$. Recall that $\Omega$ is a compact set, therefore, we can replace the sequence $G_n'$ by one of its subsequences that converges to a mixing measure $G' \in \mathcal{G}_{k_*}(\Theta)$. Since $\mathcal{D}_1(G_n', G_*) > \varepsilon'$, this result induces that $\mathcal{D}_1(G', G_*) > \varepsilon'$.

Subsequently, by means of the Fatou's lemma, we achieve that

$$0 = \lim_{n \to \infty} \mathbb{E}_X[2V(p_{G_n'}(\cdot|X), p_{G_*}(\cdot|X))] \geq \int \liminf_{n \to \infty} \Big| p_{G_n'}(Y|X) - p_{G_*}(Y|X) \Big| \, \mathrm{d}(X, Y).$$

It follows that $p_{G'}(Y|X) = p_{G_*}(Y|X)$ for almost surely $(X, Y)$. According to Lemma L.1, the noisy top-K sparse softmax gating Gaussian mixture of experts is identifiable, thus, we obtain that $G' \equiv G_*$. As a consequence, we obtain that $\mathcal{D}_1(G', G_*) = 0$, which contradicts to the fact that $\mathcal{D}_1(G', G_*) > \varepsilon' > 0$.

Hence, the proof is completed.

## K.2   Proof of Theorem 3.1

In this appendix, we employ results for M-estimators in [91] to establish the density estimation rate under the Laplace gating Gaussian mixture of experts (MoE).

Firstly, we introduce some necessary notations and fundamental results. In particular, let $\mathcal{P}_k(\Theta) := \{p_G(Y|X) : G \in \mathcal{G}_k(\Theta)\}$ be the set of all conditional density functions w.r.t mixing measures in $\mathcal{G}_k(\Theta)$. Next, we denote by $N(\varepsilon, \mathcal{P}_k(\Theta), \|\cdot\|_\infty)$ the covering number of metric space $(\mathcal{P}_k(\Theta), \|\cdot\|_\infty)$. Meanwhile, $H_B(\varepsilon, \mathcal{P}_k(\Theta), h)$ stands for the bracketing entropy of $\mathcal{P}_k(\Theta)$ under the Hellinger distance $h$ where $h(p, q) := \left(\frac{1}{2} \int (\sqrt{p} - \sqrt{q})^2 d\mu\right)^{1/2}$ for any probability densities $p, q$ dominated by the Lebesgue measure $\mu$. Then, we provide in the following lemma the upper bounds of those terms.

**Lemma K.1.** *If $\Theta$ is a bounded set, then the following inequalities hold for any $0 < \eta < 1/2$:*

*(i) $\log N(\eta, \mathcal{P}_k(\Theta), \|\cdot\|_\infty) \lesssim \log(1/\eta)$;*

*(ii) $H_B(\eta, \mathcal{P}_k(\Theta), h) \lesssim \log(1/\eta)$.*

Proof of Lemma K.1 is in Appendix K.2.2. Subsequently, we denote

$$\widetilde{\mathcal{P}}_k(\Theta) := \{p_{(G+G_*)/2}(Y|X) : G \in \mathcal{G}_k(\Theta)\};$$

$$\widetilde{\mathcal{P}}_k^{1/2}(\Theta) := \{p_{(G+G_*)/2}^{1/2}(Y|X) : G \in \mathcal{G}_k(\Theta)\}.$$

In addition, for each $\delta > 0$, we define a Hellinger ball centered around the conditional density function $p_{G_*}(Y|X)$ and intersected with the set $\widetilde{\mathcal{P}}_k^{1/2}(\Theta)$ as

$$\widetilde{\mathcal{P}}_k^{1/2}(\Theta, \delta) := \{p^{1/2} \in \widetilde{\mathcal{P}}_k^{1/2}(\Theta) : h(p, p_{G_*}) \leq \delta\}.$$

To capture the size of the above Hellinger ball, [91] suggest using the following quantity:

$$\mathcal{J}_B(\delta, \widetilde{\mathcal{P}}_k^{1/2}(\Theta, \delta)) := \int_{\delta^2/2^{13}}^{\delta} H_B^{1/2}(t, \widetilde{\mathcal{P}}_k^{1/2}(\Theta, t), \|\cdot\|_2) \mathrm{d}t \vee \delta, \tag{17}$$

where $t \vee \delta := \max\{t, \delta\}$. Given those notations, let us recall a standard result for density estimation in [91].

**Lemma K.2** (Theorem 7.4, [91]). *Take* $\Psi(\delta) \geq \mathcal{J}_B(\delta, \widetilde{\mathcal{P}}_k^{1/2}(\Theta, \delta))$ *such that* $\Psi(\delta)/\delta^2$ *is a non-increasing function of* $\delta$. *Then, for some sequence* $(\delta_n)$ *and universal constant* $c$ *which satisfy* $\sqrt{n}\delta_n^2 \geq c\Psi(\delta)$, *we obtain that*

$$\mathbb{P}\left(\mathbb{E}_X\left[h(p_{\widehat{G}_n}(\cdot|X), p_{G_*}(\cdot|X))\right] > \delta\right) \leq c\exp(-n\delta^2/c^2),$$

*for any* $\delta \geq \delta_n$

Proof of Lemma K.2 can be found in [91]. Now, we are ready to provide the proof for convergence rate of density estimation in Theorem J.1 in Appendix K.2.1.

### K.2.1 Main Proof

It is worth noting that for any $t > 0$, we have

$$H_B(t, \widetilde{\mathcal{P}}_k^{1/2}(\Theta, t), \|\cdot\|_2) \leq H_B(t, \mathcal{P}_k(\Omega, t), h).$$

Then, the integral in equation 17 is upper bounded as follows:

$$\mathcal{J}_B(\delta, \widetilde{\mathcal{P}}_k^{1/2}(\Theta, \delta)) \leq \int_{\delta^2/2^{13}}^{\delta} H_B^{1/2}(t, \mathcal{P}_k(\Omega, t), h)\mathrm{d}t \vee \delta \lesssim \int_{\delta^2/2^{13}}^{\delta} \log(1/t)\mathrm{d}t \vee \delta, \tag{18}$$

where the second inequality follows from part (ii) of Lemma K.1.

As a result, by choosing $\Psi(\delta) = \delta \cdot \sqrt{\log(1/\delta)}$, we can verify that $\Psi(\delta)/\delta^2$ is a non-increasing function of $\delta$. Furthermore, the inequality in equation 18 indicates that $\Psi(\delta) \geq \mathcal{J}_B(\delta, \widetilde{\mathcal{P}}_k^{1/2}(\Theta, \delta))$. Next, let us consider a sequence $(\delta_n)$ defined as $\delta_n := \sqrt{\log(n)/n}$. This sequence can be validated to satisfy the condition $\sqrt{n}\delta_n^2 \geq c\Psi(\delta)$ for some universal constant $c$. Therefore, by Lemma K.2, we reach the conclusion of Theorem J.1:

$$\mathbb{P}\left(\mathbb{E}_X[h(p_{\widehat{G}_n}(\cdot|X), p_{G_*}(\cdot|X))] > C\sqrt{\log(n)/n}\right) \lesssim n^{-c},$$

for some universal constant $C$ depending only on $\Theta$.

### K.2.2 Proof of Lemma K.1

**Part (i).** In this part, we will derive the following upper bound for the covering number of metric space $(\mathcal{P}_k(\Theta), \|\cdot\|_\infty)$ for any $0 < \eta < 1/2$ given the bounded set $\Omega$:

$$\log N(\eta, \mathcal{P}_k(\Theta), \|\cdot\|_\infty) \lesssim \log(1/\eta).$$

To start with, we denote $\Omega := \{(a, b, \nu) \in \mathbb{R}^d \times \mathbb{R} \times \mathbb{R}_+ : (\beta, W, a, b, \nu) \in \Omega\}$. As $\Theta$ is a bounded set, the set $\Omega$ is also bounded. Therefore, we can find an $\eta$-cover of $\Omega$, denoted by $\overline{\Omega}_\eta$. Additionally, we also define $\Delta := \{(\beta, W) \in \mathbb{R} \times \mathbb{R}^d : (\beta, W, a, b, \nu) \in \Omega\}$, and $\overline{\Delta}_\eta$ be an $\eta$-cover of $\Delta$. Then, it can be validated that

$$|\overline{\Omega}_\eta| \leq \mathcal{O}(\eta^{-(d+2)k}), \quad |\overline{\Delta}_\eta| \leq \mathcal{O}(\eta^{-(d+1)k}).$$

Next, for each mixing measure $G = \sum_{i=1}^k \exp(\beta_i)\delta_{(W_i, a_i, b_i, \nu_i)} \in \mathcal{G}_k(\Theta)$, we take into account two other mixing measures. The first measure is $G' = \sum_{i=1}^k \exp(\beta_i)\delta_{(W_i, \overline{a}_i, \overline{b}_i, \overline{\nu}_i)}$, where

$(\overline{a}_i, \overline{b}_i, \overline{\nu}_i) \in \overline{\Omega}_\eta$ is the closest points to $(a_i, b_i, \nu_i)$ in this set for all $i \in [k]$. The second one is $\overline{G} := \sum_{i=1}^{k} \exp(\overline{\beta}_i) \delta_{(\overline{W}_i, \overline{a}_i, \overline{b}_i, \overline{\nu}_i)}$ in which $(\overline{\beta}_i, \overline{W}_i) \in \overline{\Delta}_\eta$ for any $i \in [k]$. Next, let us define

$$\mathcal{T} := \{p_{\overline{G}} \in \mathcal{P}_k(\Theta) : (\overline{\beta}_i, \overline{W}_i) \in \overline{\Delta}_\eta, \ (\overline{a}_i, \overline{b}_i, \overline{\nu}_i) \in \overline{\Omega}_\eta, \forall i \in [k]\},$$

then it is obvious that $p_{\overline{G}} \in \mathcal{T}$. Now, we will show that $\mathcal{T}$ is an $\eta$-cover of metric space $(\mathcal{P}_k(\Theta), \|\cdot\|_\infty)$ with a note that it is not necessarily the smallest cover. Indeed, according to the triangle inequality, we have

$$\|p_G - p_{\overline{G}}\|_\infty \le \|p_G - p_{G'}\|_\infty + \|p_{G'} - p_{\overline{G}}\|_\infty. \tag{19}$$

Since the softmax function is no greater than one, the first term in the right hand side can be upper bounded as follows:

$$\begin{aligned}
\|p_G - p_{G'}\|_\infty &\le \sum_{i=1}^{k} \sup_{X \in \mathcal{X}} \operatorname{softmax}(-\|W_i - X\| + \beta_i) \cdot \left| f(Y|a_i^\top X + b_i, \nu_i) - f(Y|\overline{a}_i^\top X + \overline{b}_i, \overline{\nu}_i) \right| \\
&\le \sum_{i=1}^{k} \sup_{X \in \mathcal{X}} \left| f(Y|a_i^\top X + b_i, \nu_i) - f(Y|\overline{a}_i^\top X + \overline{b}_i, \overline{\nu}_i) \right| \\
&\lesssim \sum_{i=1}^{k} \sup_{X \in \mathcal{X}} \left( \|a_i - \overline{a}_i\| + \|b_i - \overline{b}_i\| + \|\nu_i - \overline{\nu}_i\| \right) \\
&= \sum_{i=1}^{k} \left( \|a_i - \overline{a}_i\| + \|b_i - \overline{b}_i\| + \|\nu_i - \overline{\nu}_i\| \right) \\
&\lesssim \eta. \tag{20}
\end{aligned}$$

Subsequently, we bound the second term $\|p_{G'} - p_{\overline{G}}\|_\infty$ as follows:

$$\begin{aligned}
\|p_{G'} - p_{\overline{G}}\|_\infty &\le \sum_{i=1}^{k} \sup_{X \in \mathcal{X}} \left\{ \left| \operatorname{softmax}(-\|W_i - X\| + \beta_i) - \operatorname{softmax}(-\|\overline{W}_i - X\| + \overline{\beta}_i) \right| \right. \\
&\qquad\qquad\qquad\qquad\qquad \left. \times \left| f(Y|\overline{a}_{\tau_i}^\top X + \overline{b}_{\tau_i}, \overline{\nu}_{\tau_i}) \right| \right\} \\
&\le \sum_{i=1}^{k} \sup_{X \in \mathcal{X}} \left| \operatorname{softmax}(-\|W_i - X\| + \beta_i) - \operatorname{softmax}(-\|\overline{W}_i - X\| + \overline{\beta}_i) \right| \\
&\le \sum_{i=1}^{k} \sup_{X \in \mathcal{X}} \left| -\|W_i - X\| + \beta_i + \|W_i - X\| - \beta_i \right| \\
&\le \sum_{i=1}^{k} \sup_{X \in \mathcal{X}} [\|W_i - \overline{W}_i\| + |\beta_i - \overline{\beta}_i|] \\
&\lesssim \eta, \tag{21}
\end{aligned}$$

It follows from the results in equation 19, equation 20 and equation 21 that $\|p_G - p_{\overline{G}}\|_\infty \lesssim \eta$. This result indicates that $\mathcal{T}$ is an $\eta$-cover of the metric space $(\mathcal{P}_k(\Theta), \|\cdot\|_\infty)$. As a consequence, we obtain that

$$N(\eta, \mathcal{P}_k(\Theta), \|\cdot\|_\infty) \lesssim |\overline{\Delta}_\eta| \times |\overline{\Omega}_\eta| \le \mathcal{O}(1/\eta^{(2d+3)k}),$$

which leads to the conclusion of this part: $\log N(\eta, \mathcal{P}_k(\Theta), \|\cdot\|_\infty) \lesssim \log(1/\eta)$.

**Part (ii).** In this part, we provide an upper bound for the bracketing entropy of $\mathcal{P}_k(\Theta)$ under the Hellinger distance $h$:

$$H_B(\eta, \mathcal{P}_k(\Theta), h) \lesssim \log(1/\eta).$$

Since $\Theta$ and $\mathcal{X}$ are bounded sets, there exist positive constants $\gamma, \ell, u$ such that $-\gamma \le a^\top X + b \le \gamma$ and $\ell \le \nu \le u$. Let us define

$$B(Y|X) := \begin{cases} \frac{1}{\sqrt{2\pi\ell}} \exp\left(-\frac{Y^2}{8u}\right), & \text{for } |Y| \ge 2\gamma \\ \frac{1}{\sqrt{2\pi\ell}}, & \text{for } |Y| < 2\gamma \end{cases}$$

Then, it can be validated that $f(Y|a^\top X + b, \nu) \le B(Y|X)$ for any $(X, Y) \in \mathcal{X} \times \mathcal{Y}$.

Next, let $\zeta \le \eta$ which will be chosen later and $\{p_1, \ldots, p_N\}$ be an $\zeta$-cover of metric space $(\mathcal{P}_k(\Theta), \|\cdot\|_\infty)$ with the covering number $N := N(\zeta, \mathcal{P}_k(\Theta), \|\cdot\|_\infty)$. Additionally, we also consider brackets of the form $[\Psi_i^L(Y|X), \Psi_i^U(Y|X)]$ where

$$\Psi_i^L(Y|X) := \max\{p_i(Y|X) - \zeta, 0\}$$
$$\Psi_i^U(Y|X) := \max\{p_i(Y|X) + \zeta, B(Y|X)\}.$$

Then, we can check that $\mathcal{P}_k(\Theta) \subseteq \bigcup_{i=1}^N [\Psi_i^L(Y|X), \Psi_i^U(Y|X)]$ and $\Psi_i^U(Y|X) - \Psi_i^L(Y|X) \le \min\{2\zeta, B(Y|X)\}$.

Let $S := \max\{2\gamma, \sqrt{8u}\} \log(1/\zeta)$, we have for any $i \in [N]$ that

$$\|\Psi_i^U - \Psi_i^L\|_1 = \int_{|Y| < 2\gamma} [\Psi_i^U(Y|X) - \Psi_i^L(Y|X)] \, \mathrm{d}(X, Y) + \int_{|Y| \ge 2\gamma} [\Psi_i^U(Y|X) - \Psi_i^L(Y|X)] \, \mathrm{d}(X, Y)$$

$$\le S\zeta + \exp\left(-\frac{S^2}{2u}\right) \le S'\zeta,$$

where $S'$ is some positive constant. This inequality indicates that

$$H_B(S'\zeta, \mathcal{P}_k(\Theta), \|\cdot\|_1) \le \log N(\zeta, \mathcal{P}_k(\Theta), \|\cdot\|_\infty) \le \log(1/\zeta).$$

By setting $\zeta = \eta/S'$, we obtain that $H_B(\eta, \mathcal{P}_k(\Theta), \|\cdot\|_1) \lesssim \log(1/\eta)$. Finally, due to the inequality $h^2 \le \|\cdot\|_1$, we reach the conclusion of this part:

$$H_B(\eta, \mathcal{P}_k(\Theta), h) \lesssim \log(1/\eta).$$

Hence, the proof is completed.

### K.3 Proof of Theorem 3.2

In order to establish the following Total Variation lower bound under the over-specified settings, i.e. when $k > k_*$ is unknown:

$$\mathbb{E}_X[V(p_G(\cdot|X), p_{G_*}(\cdot|X))] \gtrsim \mathcal{D}_2(G, G_*),$$

we need to prove two following inequalities:

- **Inequality A.** $\inf_{G \in \mathcal{G}_k(\Theta): \mathcal{D}_2(G, G_*) \le \varepsilon'} \dfrac{\mathbb{E}_X[V(p_G(\cdot|X), p_{G_*}(\cdot|X))]}{\mathcal{D}_2(G, G_*)} > 0$;

- **Inequality B.** $\inf_{G \in \mathcal{G}_k(\Theta): \mathcal{D}_2(G, G_*) > \varepsilon'} \dfrac{\mathbb{E}_X[V(p_G(\cdot|X), p_{G_*}(\cdot|X))]}{\mathcal{D}_2(G, G_*)} > 0$,

for some constant $\varepsilon' > 0$. As the inequality B can be achieved in the same fashion as in Appendix K.1, we concentrate on showing the inequality A in this proof. For that purpose, it suffices to prove that

$$\lim_{\varepsilon \to 0} \inf_{G \in \mathcal{G}_k(\Theta): \mathcal{D}_2(G, G_*) \le \varepsilon} \frac{\mathbb{E}_X[V(p_G(\cdot|X), p_{G_*}(\cdot|X))]}{\mathcal{D}_2(G, G_*)} > 0. \tag{22}$$

Assume that the above claim does not hold true, then there exists a sequence of mixing measures $G_n := \sum_{i=1}^{k_n} \exp(\beta_i^n) \delta_{(W_i^n, a_i^n, b_i^n, \nu_i^n)} \in \mathcal{G}_k(\Theta)$ such that both the terms $\mathcal{D}_2(G_n, G_*)$ and $\mathbb{E}_X[V(p_{G_n}(\cdot|X), p_{G_*}(\cdot|X))]/\mathcal{D}_2(G_n, G_*)$ go to zero as $n \to \infty$. Let us recall the formulation of the loss $\mathcal{D}_2(G_n, G_*)$:

$$\mathcal{D}_2(G_n, G_*) = \sum_{\substack{j \in [k_*], \, i \in \mathcal{A}_j \\ |\mathcal{A}_j| > 1}} \exp(\beta_i^n) \left[ \|\Delta W_{ij}^n\|^2 + \|\Delta a_{ij}^n\|^2 + |\Delta b_{ij}^n|^{\bar{r}_j} + |\Delta \nu_{ij}^n|^{\frac{\bar{r}_j}{2}} \right]$$

$$+ \sum_{\substack{j \in [k_*], \, i \in \mathcal{A}_j \\ |\mathcal{A}_j| = 1}} \exp(\beta_i^n) \left[ \|\Delta W_{ij}^n\| + \|\Delta a_{ij}^n\| + |\Delta b_{ij}^n| + |\Delta \nu_{ij}^n| \right] + \sum_{j=1}^{k_*} \left| \sum_{i \in \mathcal{A}_j} \exp(\beta_i^n) - \exp(\beta_j^*) \right|.$$

$$\tag{23}$$

Since $\mathcal{D}_2(G_n, G_*) \to 0$, we deduce that $\sum_{i \in \mathcal{A}_j} \exp(\beta_i^n) \to \exp(\beta_j^*)$ and $(W_i^n, a_i^n, b_i^n, \nu_i^n) \to (W_j^*, a_j^*, b_j^*, \nu_j^*)$ for all $i \in \mathcal{A}_j$ and $j \in [k_*]$.

Now, we reuse the three-step framework in Appendix K.1.

**Stage 1 - Density decomposition**:

Firstly, by abuse of notations, let us consider the quantity

$$Q_n := \Big[ \sum_{j=1}^{k_*} \exp(-\|W_j^* - X\| + \beta_j^*) \Big] \cdot [p_{G_n}(Y|X) - p_{G_*}(Y|X)].$$

Similar to Step 1 in Appendix K.1, we can express this term as

$$Q_n = \sum_{j=1}^{k_*} \sum_{i \in \mathcal{A}_j} \exp(\beta_i^n) \Big[ F(Y|X; W_i^n, a_i^n, b_i^n, \nu_i^n) - F(Y|X; W_j^*, a_j^*, b_j^*, \nu_j^*) \Big]$$

$$- \sum_{j=1}^{k_*} \sum_{i \in \mathcal{A}_j} \exp(\beta_i^n) \Big[ H(Y|X; W_i^n) - H(Y|X; W_j^*) \Big]$$

$$+ \sum_{j=1}^{k_*} \Big[ \sum_{i \in \mathcal{A}_j} \exp(\beta_i^n) - \exp(\beta_j^*) \Big] \Big[ F(Y|X; W_j^*, a_j^*, b_j^*, \nu_j^*) - H(Y|X, W_j^*) \Big]$$

$$:= A_n - B_n + E_n,$$

Next, we proceed to decompose $A_n$ based on the cardinality of the Voronoi cells as follows:

$$A_n = \sum_{j:|\mathcal{A}_j|=1} \sum_{i \in \mathcal{A}_j} \exp(\beta_i^n) \Big[ F(Y|X; W_i^n, a_i^n, b_i^n, \nu_i^n) - F(Y|X; W_j^*, a_j^*, b_j^*, \nu_j^*) \Big]$$

$$+ \sum_{j:|\mathcal{A}_j|>1} \sum_{i \in \mathcal{A}_j} \exp(\beta_i^n) \Big[ F(Y|X; W_i^n, a_i^n, b_i^n, \nu_i^n) - F(Y|X; W_j^*, a_j^*, b_j^*, \nu_j^*) \Big].$$

By applying the Taylor expansions of order 1 and $\bar{r}_j$ to the first and second terms of $A_n$, respectively, and following the derivation in equation 11, we get that

$$A_n = \sum_{j:|\mathcal{A}_j|=1} \sum_{i \in \mathcal{A}_j} \sum_{|\alpha_1|=0}^{1} \sum_{|\alpha_2|=0}^{1-|\alpha_1|} \sum_{\eta=0}^{2(1-|\alpha_1|-|\alpha_2|)} \sum_{\substack{\alpha_3+2\alpha_4=\eta, \\ 0 \le \alpha_3+\alpha_4 \le 1-|\alpha_1|-|\alpha_2|}} \frac{\exp(\beta_i^n)}{2^{\alpha_4} \alpha!} \cdot (\Delta W_{ij}^n)^{\alpha_1} (\Delta a_{ij}^n)^{\alpha_2}$$

$$\times (\Delta b_{ij}^n)^{\alpha_3} (\Delta \nu_{ij}^n)^{\alpha_4} \cdot X^{\alpha_2} \cdot \frac{\partial^{|\alpha_1|} g}{\partial W^{\alpha_1}}(X; W_j^*) \cdot \frac{\partial^{|\alpha_2|+\eta} f}{\partial h_1^{|\alpha_2|+\eta}}(Y|(a_j^*)^\top X + b_j^*, \nu_j^*) + R_3(X, Y)$$

$$+ \sum_{j:|\mathcal{A}_j|>1} \sum_{i \in \mathcal{A}_j} \sum_{|\alpha_1|=0}^{\bar{r}_j} \sum_{|\alpha_2|=0}^{\bar{r}_j-|\alpha_1|} \sum_{\eta=0}^{2(\bar{r}_j-|\alpha_1|-|\alpha_2|)} \sum_{\substack{\alpha_3+2\alpha_4=\eta, \\ 0 \le \alpha_3+\alpha_4 \le \bar{r}_j-|\alpha_1|-|\alpha_2|}} \frac{\exp(\beta_i^n)}{2^{\alpha_4} \alpha!} \cdot (\Delta W_{ij}^n)^{\alpha_1} (\Delta a_{ij}^n)^{\alpha_2}$$

$$\times (\Delta b_{ij}^n)^{\alpha_3} (\Delta \nu_{ij}^n)^{\alpha_4} \cdot X^{\alpha_2} \cdot \frac{\partial^{|\alpha_1|} g}{\partial W^{\alpha_1}}(X; W_j^*) \cdot \frac{\partial^{|\alpha_2|+\eta} f}{\partial h_1^{|\alpha_2|+\eta}}(Y|(a_j^*)^\top X + b_j^*, \nu_j^*) + R_4(X, Y)$$

where $R_i(X, Y)$ is a Taylor remainder such that $R_i(X, Y)/\mathcal{D}_2(G_n, G_*) \to 0$ as $n \to \infty$ for $i \in \{3, 4\}$. Next, we apply the Taylor expansions of order 1 and 2 to the first and second terms of $B_n$, respectively, and following the derivation in equation 12, we get that

$$B_n = \sum_{j:|\mathcal{A}_j|=1} \sum_{i \in \mathcal{A}_j} \sum_{|\gamma|=1} \frac{\exp(\beta_i^n)}{\gamma!} (\Delta W_{ij}^n)^\gamma \cdot \frac{\partial^{|\gamma|} g}{\partial W^\gamma}(X; W_j^*) p_{G_n}(Y|X) + R_5(X, Y)$$

$$\sum_{j:|\mathcal{A}_j|>1} \sum_{i \in \mathcal{A}_j} \sum_{|\gamma|=1}^{2} \frac{\exp(\beta_i^n)}{\gamma!} (\Delta W_{ij}^n)^\gamma \cdot \frac{\partial^{|\gamma|} g}{\partial W^\gamma}(X; W_j^*) p_{G_n}(Y|X) + R_6(X, Y),$$

where $R_5(X,Y)$ and $R_6(X,Y)$ are Taylor remainders such that their ratios over $\mathcal{D}_2(G_n, G_*)$ approach zero as $n \to \infty$. Subsequently, let us define

$$S^n_{j,\alpha_1,\alpha_2,\eta} := \sum_{i \in \mathcal{A}_j} \sum_{\substack{\alpha_3 + 2\alpha_4 = \eta, \\ 0 \le \alpha_3 + \alpha_4 \le \bar{r}_j - |\alpha_1| - |\alpha_2|}} \frac{\exp(\beta^n_i)}{2^{\alpha_4} \alpha!} \cdot (\Delta W^n_{ij})^{\alpha_1} (\Delta a^n_{ij})^{\alpha_2} (\Delta b^n_{ij})^{\alpha_3} (\Delta \nu^n_{ij})^{\alpha_4},$$

$$T^n_{j,\gamma} := \sum_{i \in \mathcal{A}_j} \frac{\exp(\beta^n_i)}{\gamma!} (\Delta W^n_{ij})^{\gamma},$$

for any $(\alpha_1, \alpha_2, \eta) \neq (\mathbf{0}_d, \mathbf{0}_d, 0)$ and $\gamma \neq \mathbf{0}_d$. Otherwise, $S^n_{j,\mathbf{0}_d,\mathbf{0}_d,0} = T^n_{j,\mathbf{0}_d} := \sum_{i \in \mathcal{A}_j} \exp(\beta^n_i) - \exp(\beta^*_j)$. As a consequence, it follows that

$$Q_n = \sum_{j=1}^{k_*} \sum_{|\alpha_1|=0}^{\bar{r}_j} \sum_{|\alpha_2|=0}^{\bar{r}_j - |\alpha_1|} \sum_{\eta=0}^{2(\bar{r}_j - |\alpha_1| - |\alpha_2|)} S^n_{j,\alpha_1,\alpha_2,\eta} \cdot X^{\alpha_2} \cdot \frac{\partial^{|\alpha_1|} g}{\partial W^{\alpha_1}}(X; W^*_j) \cdot \frac{\partial^{|\alpha_2|+\eta} f}{\partial h_1^{|\alpha_2|+\eta}}(Y|(a^*_j)^\top X + b^*_j, \nu^*_j)$$

$$+ \sum_{j=1}^{k_*} \sum_{|\gamma|=0}^{1 + \mathbf{1}_{\{|\mathcal{A}_j|>1\}}} T^n_{j,\gamma} \cdot \frac{\partial^{|\gamma|} g}{\partial W^{\gamma}}(X; W^*_j) p_{G_n}(Y|X) + R_3(X,Y) + R_4(X,Y) + R_5(X,Y) + R_6(X,Y).$$

$$(24)$$

**Stage 2 - Non-vanishing coefficients**:

In this step, we demonstrate that not all the ratios $S^n_{j,\alpha_1,\alpha_2,\eta}/\mathcal{D}_2(G_n, G_*)$ and $T^n_{j,\gamma}/\mathcal{D}_2(G_n, G_*)$ converge to zero as $n \to \infty$. Assume by contrary that all these terms go to zero. Then, by employing arguments for deriving equation 14 and equation 15, we get that

$$\frac{1}{\mathcal{D}_2(G_n, G_*)} \cdot \Big[ \sum_{j=1}^{k_*} \Big| \sum_{i \in \mathcal{A}_j} \exp(\beta^n_i) - \exp(\beta^*_j) \Big|$$

$$+ \sum_{j:|\mathcal{A}_j|=1} \sum_{i \in \mathcal{A}_j} \exp(\beta^n_i) \Big( \|\Delta W^n_{ij}\| + \|\Delta a^n_{ij}\| + |\Delta b^n_{ij}| + |\Delta \nu^n_{ij}| \Big) \Big] \to 0.$$

Taking the summation of $\sum_{j:|\mathcal{A}_j|>1} \frac{|S^n_{j,\alpha_1,\alpha_2,\eta}|}{\mathcal{D}_2(G_n,G_*)}$ for all $(\alpha_1, \alpha_2, \eta)$ where $\alpha_1 \in \{2e_1, 2e_2, \ldots, 2e_d\}$, $\alpha_2 = \mathbf{0}_d$ and $\eta = 0$, we have

$$\frac{1}{\mathcal{D}_2(G_n, G_*)} \cdot \sum_{j:|\mathcal{A}_j|>1} \sum_{i \in \mathcal{A}_j} \exp(\beta^n_i) \|\Delta W^n_{ij}\|^2 \to 0.$$

Taking the summation of $\sum_{j:|\mathcal{A}_j|>1} \frac{|S^n_{j,\alpha_1,\alpha_2,\eta}|}{\mathcal{D}_2(G_n,G_*)}$ for all $(\alpha_1, \alpha_2, \eta)$ where $\alpha_1 = \mathbf{0}_d$, $\alpha_2 \in \{2e_1, 2e_2, \ldots, 2e_d\}$ and $\eta = 0$, we have

$$\frac{1}{\mathcal{D}_2(G_n, G_*)} \cdot \sum_{j:|\mathcal{A}_j|>1} \sum_{i \in \mathcal{A}_j} \exp(\beta^n_i) \|\Delta a^n_{ij}\|^2 \to 0.$$

Combine the above limit with the formulation of $\mathcal{D}_2(G_n, G_*)$ in equation 23, we have that

$$\frac{1}{\mathcal{D}_2(G_n, G_*)} \cdot \sum_{j:|\mathcal{A}_j|>1} \sum_{i \in \mathcal{A}_j} \exp(\beta^n_i) \Big( |\Delta b^n_{ij}|^{\bar{r}_j} + |\Delta \nu^n_{ij}|^{\frac{\bar{r}_j}{2}} \Big) \not\to 0.$$

This result implies that we can find some index $j' \in [k_*] : |\mathcal{A}_{j'}| > 1$ that satisfies

$$\frac{1}{\mathcal{D}_2(G_n, G_*)} \cdot \sum_{i \in \mathcal{A}_{j'}} \exp(\beta^n_i) \Big( |\Delta b^n_{ij'}|^{\bar{r}_{j'}} + |\Delta \nu^n_{ij'}|^{\frac{\bar{r}_{j'}}{2}} \Big) \not\to 0.$$

For simplicity, we may assume that $j' = 1$. Since $S^n_{1,\mathbf{0}_d,\mathbf{0}_d,\eta}/\mathcal{D}_2(G_n, G_*)$ vanishes as $n \to \infty$ for any $1 \le \eta \le \bar{r}_j$, we divide this term by the left hand side of the above equation and achieve that

$$\frac{\sum_{i \in \mathcal{A}_1} \sum_{\substack{\alpha_3 + 2\alpha_4 = \eta, \\ 1 \le \alpha_3 + \alpha_4 \le \bar{r}_1}} \frac{\exp(\beta^n_i)}{2^{\alpha_4} \alpha!} (\Delta b^n_{i1})^{\alpha_3} (\Delta \nu^n_{i1})^{\alpha_4}}{\sum_{i \in \mathcal{A}_1} \exp(\beta^n_i) \Big( |\Delta b^n_{i1}|^{\bar{r}_1} + |\Delta \nu^n_{i1}|^{\frac{\bar{r}_1}{2}} \Big)} \to 0, \tag{25}$$

for any $1 \leq \eta \leq \bar{r}_1$.

Subsequently, we define $M_n := \max\{|\Delta b_{i1}^n|, |\Delta \nu_{i1}^n|^{1/2} : i \in \mathcal{A}_1\}$ and $\pi_n := \max\{\exp(\beta_i^n) : i \in \mathcal{A}_1\}$. As a result, the sequence $\exp(\beta_i^n)/\pi_n$ is bounded, which indicates that we can substitute it with its subsequence that admits a positive limit $z_{5i}^2 := \lim_{n\to\infty} \exp(\beta_i^n)/\pi_n$. Therefore, at least one among the limits $z_{5i}^2$ equals to one. Furthermore, we also denote

$$(\Delta b_{i1}^n)/M_n \to z_{3i}, \ (\Delta \nu_{i1}^n)/(2M_n) \to z_{4i}.$$

From the above definition, it follows that at least one among the limits $z_{3i}$ and $z_{4i}$ equals to either 1 or $-1$. By dividing both the numerator and the denominator of the term in equation 25 by $\pi_n M_n^\eta$, we arrive at the following system of polynomial equations:

$$\sum_{i \in \mathcal{A}_1} \sum_{\substack{\alpha_3 + 2\alpha_4 = \eta, \\ 1 \leq \alpha_3 + \alpha_4 \leq \bar{r}_1}} \frac{z_{5i}^2 \, z_{3i}^{\alpha_3} \, z_{4i}^{\alpha_4}}{\alpha_3! \, \alpha_4!} = 0,$$

for all $1 \leq \eta \leq \bar{r}_1$. Nevertheless, from the definition of $\bar{r}_1$, we know that the above system does not admit any non-trivial solutions, which is a contradiction. Consequently, not all the ratios $S_{j,\alpha_1,\alpha_2,\eta}^n/\mathcal{D}_2(G_n, G_*)$ and $T_{j,\gamma}^n/\mathcal{D}_2(G_n, G_*)$ tend to zero as $n \to \infty$.

**Stage 3 - Fatou's contradiction**:

Recall that $\mathbb{E}_X[V(p_{G_n}(\cdot|X), p_{G_*}(\cdot|X))]/\mathcal{D}_2(G_n, G_*) \to 0$ as $n \to \infty$. Then, by applying the Fatou's lemma, we get

$$0 = \lim_{n\to\infty} \frac{\mathbb{E}_X[V(p_{G_n}(\cdot|X), p_{G_*}(\cdot|X))]}{\mathcal{D}_2(G_n, G_*)} \geq \frac{1}{2} \cdot \int \liminf_{n\to\infty} \frac{|p_{G_n}(Y|X) - p_{G_*}(Y|X)|}{\mathcal{D}_2(G_n, G_*)} \mathrm{d}(X,Y),$$

which implies that $|p_{G_n}(Y|X) - p_{G_*}(Y|X)|/\mathcal{D}_2(G_n, G_*) \to 0$ as $n \to \infty$ for almost surely $(X, Y)$.

Next, we define $m_n$ as the maximum of the absolute values of $S_{j,\alpha_1,\alpha_2,\eta}^n/\mathcal{D}_2(G_n, G_*)$. It follows from Step 2 that $1/m_n \not\to \infty$. Moreover, by arguing in the same way as in Step 3 in Appendix K.1, we receive that

$$Q_n/[m_n \mathcal{D}_2(G_n, G_*)] \to 0 \tag{26}$$

as $n \to \infty$. By abuse of notations, let us denote

$$S_{j,\alpha_1,\alpha_2,\eta}^n/[m_n \mathcal{D}_2(G_n, G_*)] \to \xi_{j,\alpha_1,\alpha_2,\eta},$$
$$T_{j,\gamma}^n/[m_n \mathcal{D}_2(G_n, G_*)] \to \kappa_{j,\gamma}.$$

Here, at least one among $\xi_{j,\alpha_1,\alpha_2,\eta}, \kappa_{j,\gamma}$ is non-zero. Then, by putting the results in equation 24 and equation 26 together, we get

$$\sum_{j=1}^{k_*} \sum_{|\alpha_1|=0}^{\bar{r}_j} \sum_{|\alpha_2|=0}^{\bar{r}_j - |\alpha_1|} \sum_{\eta=0}^{2(\bar{r}_j - |\alpha_1| - |\alpha_2|)} \xi_{j,\alpha_1,\alpha_2,\eta} \cdot X^{\alpha_2} \cdot \frac{\partial^{|\alpha_1|} g}{\partial W^{\alpha_1}}(X; W_j^*) \cdot \frac{\partial^{|\alpha_2|+\eta} f}{\partial h_1^{|\alpha_2|+\eta}}(Y|(a_j^*)^\top X + b_j^*, \nu_j^*)$$

$$+ \sum_{j=1}^{K} \sum_{|\gamma|=0}^{1+\mathbf{1}_{\{|\mathcal{A}_j|>1\}}} \kappa_{j,\gamma} \cdot \frac{\partial^{|\gamma|} g}{\partial W^\gamma}(X; W_j^*) p_{G_n}(Y|X) = 0.$$

Arguing in a similar fashion as in Step 3 of Appendix K.1, we obtain that $\xi_{j,\alpha_1,\alpha_2,\eta} = \kappa_{j,\gamma} = 0$ for any $j \in [k_*]$, $0 \leq |\alpha_1| + |\alpha_2| \leq 2\bar{r}_j$, $0 \leq \eta \leq 2(\bar{r}_j - |\alpha_1| - |\alpha_2|)$ and $0 \leq |\gamma| \leq 1 + \mathbf{1}_{\{|\mathcal{A}_j|>1\}}$. This contradicts the fact that at least one among them is non-zero. Hence, the proof is completed.

## L  Identifiability of the Laplace Gating Gaussian MoE

**Lemma L.1.** *For any mixing measures $G$ and $G_*$ in $\mathcal{G}_k(\Theta)$ that satisfy $p_G(Y|X) = p_{G_*}(Y|X)$ for almost surely $(X, Y) \in \mathcal{X} \times \mathcal{Y}$, we have that $G \equiv G_*$.*

*Proof of Lemma L.1.* First, we assume that two mixing measures $G$ and $G_*$ take the following forms: $G = \sum_{i=1}^{k} \exp(\beta_i)\delta_{(W_i, a_i, b_i, \nu_i)}$ and $G_* = \sum_{i=1}^{k_*} \exp(\beta_i^*)\delta_{(W_i^*, a_i^*, b_i^*, \nu_i^*)}$. Recall that

$p_G(Y|X) = p_{G_*}(Y|X)$ for almost surely $(X, Y)$, then we have

$$\sum_{i=1}^{k} \text{softmax}\left(-\|W_i - X\| + \beta_i\right) \cdot f(Y|a_i^\top X + b_i, \nu_i)$$

$$= \sum_{i=1}^{k_*} \text{softmax}(-\|W_i^* - X\| + \beta_i^*) \cdot f(Y|(a_i^*)^\top + b_i^*, \nu_i^*). \quad (27)$$

Due to the identifiability of the location-scale Gaussian mixtures [85, 86, 87], we get that $k = k_*$ and

$$\left\{\text{softmax}(-\|W_i - X\| + \beta_i) : i \in [k]\right\} \equiv \left\{\text{softmax}(-\|W_i^* - X\| + \beta_i^*) : i \in [k]\right\},$$

for almost surely $X$. WLOG, we may assume that

$$\text{softmax}(-\|W_i - X\| + \beta_i) = \text{softmax}(-\|W_i^* - X\| + \beta_i^*), \quad (28)$$

for almost surely $X$ for any $i \in [k]$. Since the softmax function is invariant to translations, it follows from equation 28 that $W_i = W_i^*$ and $\beta_i = \beta_i^* + v_0$ for some $v_0 \in \mathbb{R}$. Notably, from the assumption of the model, we have $\beta_k = \beta_k^* = 0$, which implies that $v_0 = 0$. As a result, we obtain that $\beta_i = \beta_i^*$ for any $i \in [k_*]$. Then, equation 27 can be rewritten as

$$\sum_{i=1}^{k_*} \exp(\beta_i) \exp(-\|W_i^* - X\|) f(Y|a_i^\top X + b_i, \nu_i)$$

$$= \sum_{i=1}^{k_*} \exp(\beta_i) \exp(-\|W_i^* - X\|) f(Y|(a_i^*)^\top X + b_i^*, \nu_i^*), \quad (29)$$

for almost surely $(X, Y)$. Next, we denote $J_1, J_2, \ldots, J_m$ as a partition of the index set $[k_*]$, where $m \le k_*$, such that $\exp(\beta_i) = \exp(\beta_{i'})$ for any $i, i' \in J_j$ and $j \in [m]$. On the other hand, when $i$ and $i'$ do not belong to the same set $J_j$, we let $\exp(\beta_i) \ne \exp(\beta_{i'})$. Thus, we can reformulate equation 29 as

$$\sum_{j=1}^{m} \sum_{i \in J_j} \exp(\beta_i) \exp(-\|W_i^* - X\|) f(Y|a_i^\top X + b_i, \nu_i)$$

$$= \sum_{j=1}^{m} \sum_{i \in J_j} \exp(\beta_i) \exp(-\|W_i^* - X\|) f(Y|(a_i^*)^\top X + b_i^*, \nu_i^*),$$

for almost surely $(X, Y)$. This results leads to $\{((a_i)^\top X + b_i, \nu_i) : i \in J_j\} \equiv \{((a_i^*)^\top X + b_i^*, \nu_i^*) : i \in J_j\}$, for almost surely $X$ for any $j \in [m]$. Therefore, we have

$$\{(a_i, b_i, \nu_i) : i \in J_j\} \equiv \{(a_i^*, b_i^*, \nu_i^*) : i \in J_j\},$$

for any $j \in [m]$. As a consequence,

$$G = \sum_{j=1}^{m} \sum_{i \in J_j} \exp(\beta_i) \delta_{(W_i, a_i, b_i, \nu_i)} = \sum_{j=1}^{m} \sum_{i \in J_j} \exp(\beta_i^*) \delta_{(W_i^*, a_i^*, b_i^*, \nu_i^*)} = G_*.$$

Hence, we reach the conclusion of this lemma. $\qquad \square$

## M    Broader Impact

This paper presents research aimed at propelling advancements in the broad domain of machine learning. The implications of our findings are wide-ranging, with potential applications in sectors including healthcare, autonomous driving, and recommendation systems. Based on our current understanding, this research does not warrant an ethics review, and a detailed discussion of the potential societal impacts is not required at the current stage.

