# OpenReview forum: "FuseMoE: Mixture-of-Experts Transformers for Fleximodal Fusion"
_NeurIPS.cc/2024/Conference — NeurIPS 2024 poster_

### Official Review · Reviewer_sZSN · 2024-07-11

**Soundness:** 3
**Presentation:** 3
**Contribution:** 2
**Rating:** 6
**Confidence:** 3

**Summary:**

The paper “FuseMoE: Mixture-of-Experts Transformers for Fleximodal Fusion” introduces a novel framework called FuseMoE, designed to tackle challenges associated with multimodal data in machine learning. This framework addresses key issues such as missing elements, temporal irregularity, and sparsity in data, which are prevalent in fields like healthcare where data is often incomplete and irregularly sampled. The core innovation in FuseMoE is its gating function that integrates various modalities efficiently. The framework enhances predictive performance through: 1. Sparse MoE Layers: Incorporating sparsely gated MoE layers to manage tasks and learn optimal modality partitioning. 2. Laplace Gating Function: A novel gating function theoretically proven to ensure better convergence rates than traditional Softmax functions. 3. FlexiModal Data Handling: Efficiently handling scenarios with missing modalities and irregularly sampled data trajectories. The framework is validated through empirical evaluations on diverse prediction tasks, demonstrating its effectiveness in real-world applications.

**Strengths:**

1. Existing demand of a more flexible gating function in the field of multimodal MoE makes the idea of this paper promising. Also, the possible implementations of this new MoE system is promising and may lead to some practical medical tools.
2. The theoretical proof is sufficient and rigorous.
3. The experimental design is comprehensive from the perspectives of scenarios and data, to a certain extent corresponding with the theory. The appendix contains a large amount of supplementary information, which helps verify the completeness and correctness of the article.

**Weaknesses:**

1. The comparison works are all very outdated (the latest one UTDE[1] is publicized in 2023, with over half of the compared works predating 2020). Why not compare with more recent works?
2. Fig.1 lacks clear description, as the data processing workflow is not well explained.
3. Gaussian gating and your proposed form are very similar in format, with formula $h(x)=\operatorname{Top} \mathrm{K}\left(-\|W-x\|_{2}^{2}\right)$. Thus, theoretical comparison with original softmax may be unfair as the form is derived from and similar to gaussian gating. Therefore, although theoretical results may be valid, their importance diminishes due to this aspect.

[1] Improving Medical Predictions by Irregular Multimodal Electronic Health Records Modeling

**Questions:**

Please refer to the weaknesses.
The biggest question at my side is that the similarity with gaussian gating may lead to some theoretical misunderstanding. Any further explanation on this topic could lead to higher score.

**Limitations:**

The authors adequately addressed the limitations, there is some redundancy in the encoder part that heavily draws inspiration from UTDE's work. From the appendix it can be observed that encoder might have a greater impact on overall performance than what gate function alone contributes to improvement; hence further discussion is needed on whether this gate function must necessarily be tied to methods related to mTAND and imputation for obtaining higher-quality features as inputs for downstream MoE layers from an implementation perspective. Of course, the discussion on this topic is optional.

---

> ### Author Rebuttal · Authors · 2024-08-06
>
> Dear Reviewer `sZSN`,
>
> Thank you very much for your constructive suggestions. We are encouraged by your acknowledgment that our idea is promising, the theoretical proof is rigorous, and the experimental design is comprehensive, containing a large amount of supplementary information. Below, we address your concerns in detail.
>
> **W1:** We wish to emphasize that all the baselines we compared are well-known and representative methodologies within specific categories: concatenation, Tensor Fusion, Multimodal Adaptation Gate, and cross-attention-based methods. The MoE-based fusion method is a completely new line of work, and the comparison is essentially within the concept of fusion approaches.
>
> To the best of our knowledge, based on the latest baselines we compared with, there is neither new work that opens up a new line of fusion methods nor any work that is far superior to the method and baselines we presented. However, in response to your comments and those of other reviewers, we have added additional experiments and ablation studies on new datasets and settings to further strengthen our results. These can be found in the attached PDF file.
>
> **W2:** Figure 1 illustrates the example of multimodal electronic health records that possess irregularity and missing modality problems. This illustration reflects the situation of the MIMIC-IV dataset. Detailed task descriptions and data preprocessing workflows are discussed in Appendices A and D. We will add detailed explanations to Fig. 1 in the revised version.
>
> **W3:** Softmax gating is more commonly used than Gaussian gating in real-world problems involving vision, text, and potential multimodal applications, and it also achieves relatively better performance. This is why we mainly compare our method with Softmax gating. Gaussian gating has been compared with Laplace gating in a different work, which illustrates that our proposed Laplace gating is more sample-efficient than Gaussian gating as shown in Nguyen et al. [1].
>
> First, recall that the general form of the Gaussian gating is given by
> $$h(x)=\mathrm{TopK(-(W-x)^{\top}\Gamma^{-1}(W-x)/2)},$$
> where $W$ is a mean vector, and $\Gamma$ is a covariance matrix.
> Then, Nguyen et. al. [1] consider two settings of the mean parameters $W^*_j$. Under the first setting (resp. second setting), Nguyen et. al. [1] argue that due to an interaction between the mean parameters $W^*_j$ and the covariance matrix $\Gamma^*_j$ (resp. the expert parameters $a^*_j$) via the partial differential equation (PDE) in Eq.(8) (resp. Eq.(11)) in [1], the rates for estimating mean parameters $W^*_j$ (resp. $a^*_j$) decrease when the number of their fitted atoms increases, and are no faster than $O(n^{-1/8})$. On the other hand, it can be verified that such parameter interactions do not occur under the FuseMoE. Thus, the estimation rates for the mean parameters $W^*_j$ and the expert parameters $a^*_j$ when using the Laplace gating remain unchanged of order $O(n^{-1/4})$. To strengthen our arguments, we also include a table where we summarize the parameter estimation rates under the Gaussian MoE when using the Laplace gating and the Gaussian gating.
>
> Finally, we explain why the aforementioned PDEs affect the parameter estimation rates. In particular, a key step in our proof is to decompose the density difference $p_{G_n}(Y|X)-p_{G_*}(Y|X)$ into a combination of linearly independent terms using Taylor expansions. However, if those PDEs hold true, then components in that decomposition become linearly dependent, which is undesirable and leads to slow parameter estimation rates.
>
> *Summary of parameter estimation rates under the Gaussian MoE with the Gaussian gating and the Laplace gating. The function $\widetilde{r}(\cdot)$ is defined in Eq.(12) in [1], while the function $\bar{r}(\cdot)$ is defined in Eq.(9) in our paper. Note that $\widetilde{r}(\cdot)\leq\bar{r}(\cdot)$ and $\widetilde{r}(2)=4$, $\widetilde{r}(3)=6$. Additionally, $\mathcal{A}^n_j:=\mathcal{A}_j(\widehat{G}_n)$ is a Voronoi cell defined in Eq.(7) in our paper:*
> |`Gating`|$W_{j}^{*}$|$a_{j}^{*}$|$b^*_{j}$|$\nu_{j}^{*}$|
> |:---:|:----:|:----:|:----:|:----:|
> |Gaussian [1]: setting I|$\mathcal{O}(n^{-1/2\bar{r}(\|\mathcal{A}^n_{j}\|)})$|$\mathcal{O}(n^{-1/4})$| $\mathcal{O}(n^{-1/2\bar{r}(\|\mathcal{A}^n_{j}\|)})$|$\mathcal{O}(n^{-1/\bar{r}(\|\mathcal{A}^n_{j}\|)})$|
> |Gaussian [1]: setting II|$\mathcal{O}(n^{-1/2\widetilde{r}(\|\mathcal{A}^n_{j}\|)})$|$\mathcal{O}(n^{-1/\widetilde{r}(\|\mathcal{A}^n_{j}\|)})$|$\mathcal{O}(n^{-1/2\widetilde{r}(\|\mathcal{A}^n_{j}\|)})$| $\mathcal{O}(n^{-1/\widetilde{r}(\|\mathcal{A}^n_{j}\|)})$|
> |Laplace (Ours)|$\mathcal{O}(n^{-1/4})$|$\mathcal{O}(n^{-1/4})$|$\mathcal{O}(n^{-1/2\bar{r}(\|\mathcal{A}^n_{j}\|)})$|$\mathcal{O}(n^{-1/\bar{r}(\|\mathcal{A}^n_{j}\|)})$|
>
> **Limitations:** We will revise our paper to avoid potential information overlaps with prior UTDE work. However, we wish to emphasize that the encoder part was not proposed by UTDE itself; the authors of that paper also utilized prior works (i.e., mTAND, Time2Vec) for demonstration purposes, similar to our approach.
>
> Regarding the gating function and encoders, we appreciate your insights. Indeed, all the ablation studies on encoders presented in Appendix H.2 are based on the Laplace gating function of our FuseMoE framework. Choosing appropriate encoders is beyond the scope of this paper, which is why we defer this discussion to the Appendix. However, as the reviewer points out, it is beneficial to determine whether gating functions or encoders contribute more significantly to overall performance improvement. Therefore, we have added additional experimental results on mutating different gating functions in conjunction with different encoders and compared the differences in results. The figures can be found in the attached PDF file.
>
> **Reference**
>
> [1] Nguyen et al., (2024). Towards convergence rates for parameter estimation in Gaussian-gated mixture of experts.
>
> Authors

---

> > ### Author Response · Authors · 2024-08-13
> > **Kindly Request for Reviewer's Feedback**
> >
> > Dear Reviewer `sZSN`,
> >
> > We sincerely appreciate the time you have taken to provide feedback on our work, which has helped us greatly improve its clarity, among other attributes. This is a gentle reminder that the **Author-Reviewer Discussion period ends in just around 12 hours from this comment, i.e., 11:59 pm AoE on August 13.** We are happy to answer any further questions you may have before then, but we will be unable to respond after that time.
> >
> > If you agree that our responses to your reviews have addressed the questions you listed, we kindly ask that you consider whether raising your score would more accurately reflect your updated evaluation of our paper. Thank you again for your time and thoughtful comments!
> >
> > Sincerely,
> >
> > Authors

---

### Official Review · Reviewer_6SUH · 2024-07-12

**Soundness:** 3
**Presentation:** 2
**Contribution:** 3
**Rating:** 6
**Confidence:** 3

**Summary:**

This paper proposes an MOE-based model to handle multimodal data fusion. It addresses two challenges: missing modalities and irregularly sampled data trajectories. A Laplace gating function is applied to the MoE Backbone. An entropy regularization loss is proposed to ensure balanced and stable expert utilization. The author validates the method in diverse datasets.

**Strengths:**

1. Strong motivation. Two challenges presented in this paper, missing modalities and irregularly sampled data trajectories, are important in the multi-modal data fusion area. It is reasonable to address them through sparsing encoding and gating networks in the MOE layer.

2. Good theoretical analysis. This paper presents a detailed theoretical analysis of the Laplace gating over the standard Softmax gating in MoE. Other mathematical proofs seem to extensively illustrate the characteristics of the proposed MOE model.

3. Good experimental results. Comprehensive evaluations of FuseMoE on image and video datasets seem to validate its effectiveness.

**Weaknesses:**

Addressing the following weaknesses may improve the paper:

1. The author should clearly distinguish the proposed method and others’ modules. From the paper, the Laplace gating function is proposed as a new one. I am not sure the author made some contributions to the encoder design, router design, and loss design. The author should make more illustrations about the contributions, not just combine other people’s work together.

2. The experimental results are not extensive. The author should clearly demonstrate the data modality of the chosen benchmarks. It seems that other modalities, such as text, are not included. The author should explain this.

3. Please explain more concisely how the gating functions can stabilize the imbalance and sparse multi-modal data. The paper should point this out more concisely and better with experimental results. Too many mathematical proofs and theorems in Sec.3 seem not helpful in illustrating the advantages of the proposed method.

**Questions:**

Please see the weakness part to answer the questions.

**Limitations:**

Yes

---

> ### Author Rebuttal · Authors · 2024-08-06
>
> Dear Reviewer `6SUH`,
>
> We are grateful for your positive feedback and insightful comments. It is particularly encouraging to hear that you found our manuscript to have strong motivation and address important challenges, with good theoretical analysis and comprehensive experimental results. In the following, we address your major concerns in detail.
>
> **Q: The author should clearly distinguish the proposed method and others’ modules. From the paper, the Laplace gating function is proposed as a new one. I am not sure the author made some contributions to the encoder design, router design, and loss design. The author should make more illustrations about the contributions, not just combine other people’s work together.**
>
> **A:** We respectfully disagree with your claim that we are merely combining existing literature. In our work, the MoE fusion layer and missing modalities sections are newly proposed and form important parts of the FuseMoE method. The theoretical insights of the Laplace gating function are another integral and unique aspect of this paper, leading to extensive empirical evaluations under multiple circumstances and applications. Most importantly, the new setting and motivation can potentially be connected to many real-world applications.
>
> While the encoder models we employed are based on existing work, this is primarily for demonstration purposes and does not constitute a significant portion of the context. We would like to emphasize that this level of prior-work reuse has been common in many prior works, including our baselines from [1].
>
> Nevertheless, our intention was not to combine other people’s work but to provide a clearer demonstration. We truly appreciate your advice in pointing out this issue and will ensure to clarify this in the revised version.
>
> **Q: The experimental results are not extensive. The author should clearly demonstrate the data modality of the chosen benchmarks. It seems that other modalities, such as text, are not included. The author should explain this.**
>
> **A:** We have created a table summarizing the modality components and sample size of each dataset in the attached PDF file. These details can also be found in the dataset details section in Appendix B. The datasets we tested include multiple modalities such as time series, text, image, ECG, and video frames.
>
> **Q: Please explain more concisely how the gating functions can stabilize the imbalance and sparse multi-modal data. The paper should point this out more concisely and better with experimental results.**
>
> **A:** We start with the computation of gating functions to demonstrate the advantage of Laplace gating. For each token $\mathbf{h}$, and each expert  $i$ with embedding $\mathbf{e}_i$, the similarity score  $s_i$  is computed as:
>
> $s_i = \mathbf{h} \cdot \mathbf{e}_i,$
>
> which represents the affinity between the token and the expert. These scores are then passed through a Softmax function to obtain the gating probabilities $g_i$:
>
> $g_i = \frac{\exp(s_i)}{\sum_{j} \exp(s_j)},$
>
> where the gating probabilities determine the weight or importance of each expert in contributing to the final decision. **Representation collapse** occurs when the Softmax gating mechanism consistently assigns high probabilities to a small subset of experts, causing these experts to dominate the decision-making process while the others become redundant. This issue arises due to:
>
> - If a few experts have significantly higher similarity scores  $s_i$ than others for most tokens, the Softmax function will amplify these differences, leading to very high gating probabilities for these experts.
> - The exponential nature of the Softmax function makes it sensitive to differences in similarity scores, causing the highest scores to dominate.
>
> As for the Laplace gating function, instead of using the inner product, it computes the similarity score  $s_i$  as the negative L2-distance between a token’s hidden representation  $\mathbf{h}$  and an expert embedding  $\mathbf{e}_i$:
>
> $s_i = -\| \mathbf{h} - \mathbf{e}_i \|_2,$
>
> which is a distance-based similarity measure. The $L_2$-distance measures how far apart the token representation and expert embedding are in the feature space. This approach does not inherently favor any expert based on magnitude, unlike inner product which can be biased towards experts with larger norms. By considering the distance, the Laplace gating function ensures that all experts have a more balanced opportunity to be selected based on how close they are to the token representation, rather than being dominated by a few experts with higher dot products.
>
> When dealing with heterogeneous inputs, such as multimodal data (e.g., text, images, time series), the feature distributions can be very different across modalities. The $L_2$-distance is a more robust measure that can handle these differences without being overly sensitive to the scale and variance of the input features. In addition, it can gracefully degrade in the presence of missing data, rather than causing abrupt changes in gating probabilities that might occur with inner product-based measures.
>
> For experimental results on the Laplace gating function, we have added results on ImageNet using the Vision-MoE framework, which can be found in the attached PDF file. In our paper, we have already tested the Laplace gating function on MIMIC-III, MIMIC-IV, CIFAR-10, and PAM. Additionally, the results presented on the CMU-MOSI and CMU-MOSEI datasets also utilize the Laplace gating function.
>
> Sincerely,
>
> Authors
>
> **Reference**
>
> [1] Zhang et al., (2023) Improving Medical Predictions by Irregular Multimodal Electronic Health Records Modeling, ICML 2023.

---

### Official Review · Reviewer_6Ay7 · 2024-07-13

**Soundness:** 2
**Presentation:** 3
**Contribution:** 3
**Rating:** 7
**Confidence:** 4

**Summary:**

The Paper introduces “FuseMoE”, a novel mixture-of-experts (MoE) framework that can handle multi-modal data even in scenarios with missing elements, sparsity of samples and temporal irregularity.  It proposes an innovative Laplace gating function with theoretical proof to enhance convergence rates and predictive performance across various tasks. The Laplace gating function allows sparser distribution of weights among MoE and encourages more balanced utilization of experts with sharper peaks and heavier tails preventing certain experts to be over dominant. FuseMoE includes modality and irregularity encoder using a discretized multi-time attention (mTAND) module to support generic fleximodal data ingestion with unlimited input modalities instead of other popular pairwise setup that exist in the literature. Paper also outlines various router designs for processing multimodal inputs and discusses the tradeoff between them and demonstrates .

Key:

1. Laplace gating function showed superior performance compared to Softmax gating across multiple tasks
2. Unlike baseline models, FuseMoE demonstrated improved scalability with increasing number of modalities
3. Handles missing modalities with tAND and per modality routers with entropy loss function
4. theoretical analysis of convergence rates for parameter estimation
5. FuseMoe outperformed baseline methods on several fleximodal datasets.

**Strengths:**

1. Sparse MoE backbone with novel Laplace gating function that ensures better convergence rates compared to traditional Softmax with theoretical guarantees
2. Modality and Irregularity Encoder to use mTAND module to descretize irregularly sampled observations to mix with continuous features, Effective management of missing data and irregular sampling
3. MoE fusion layer to integrate embedding from different modalities with different router designs. Flexibility to handle variable number of modalities because of per modal experts routing
4. Novel methods, strong theoretical contributions and comprehensive emperical results and evaluation

**Weaknesses:**

1. Potential over-parameterization when input is small
2. limited discussion of computational needs and scalability to huge datasets
3. Most of the items discussed were already established and proved. Paper seems to be combining existing literature for these datasets.

**Questions:**

1. The paper claims to work with unlimited input modalities. is it really so? how do you anticipate the fuseMoE to handle extremely sparse data across modalities? Any thoughts/estimation on how far we can go here? How does the complexity of setup scale with number of modalities?
2. Is Laplace only tested with CIFAR-10 dataset? Why was it not tested with other bigger datasets?
3. Have you thought about or explored the interpretability side of expert assignement process?
4. Few other paper discusses and introduces Laplace instead of gaussian for MoE. Can you expand more on how FuseMoE is differnt from these?

   - Nguyen, Hien D., and Geoffrey J. McLachlan. "Laplace mixture of linear experts." *Computational Statistics & Data Analysis* 93 (2016): 177-191.

   - Wu, Lc., Zhang, Sy. & Li, Ss. Heteroscedastic Laplace mixture of experts regression models and applications. *Appl. Math. J. Chin. Univ.* **36**, 60–69 (2021). https://doi.org/10.1007/s11766-021-3591-2

**Limitations:**

Potential limitation of over-parameterization of networks especially when input data/size is small is adequately discussed in the paper.

---

> ### Author Rebuttal · Authors · 2024-08-06
>
> **W1**: Yes, we mentioned this as one of our limitations in line 316 of Section 5. This issue was observed during our empirical evaluation, where the Time2Vec method we employed transforms a univariate time series into a high-dimensional vector. This transformation helps capture trend/seasonality components and long-term dependency. However, when the number of samples is small and lacks sufficient information, transforming to a high-dimensional vector can be redundant and lead to over-parameterization.
>
> This problem only occurred in a few scenarios among the numerous experiments we conducted. As mentioned in Section 5, we plan to address this issue in future work.
>
> **W2**: We have included a discussion comparing the computational efficiency of various methods in Figure 12 of Section H.3. Additionally, we have described the computational resources used in Section G.1.
>
> We have also tested FuseMoE on large-scale multimodal datasets, including MIMIC-III, MIMIC-IV, and CMU-MOSEI. As shown in Table 1 of the MultiBench paper [1], these datasets are all featured as large-scale multimodal datasets, with MIMIC-III containing 36,212 samples, MIMIC-IV containing 73,173 samples, and CMU-MOSEI containing 22,777 samples. Even though CMU-MOSI is relatively small-scale with 2,199 samples, our experiments have overall tested a sufficient number of large-scale multimodal datasets compared to most prior works (e.g. [2]). These datasets span various data types, including text, images, time series, ECG, video frames, and audio signals, demonstrating strong generalization capacity across domains.
>
> Please find tables summarizing the information of these datasets in the attached PDF file.
>
> **W3**: We respectfully disagree that we are merely combining existing literature. In our work, the MoE fusion layer and missing modalities sections are newly proposed and form important parts of FuseMoE. The theoretical insights of Laplace gating are another integral aspect of this paper, leading to extensive empirical evaluations under multiple circumstances. Most importantly, the new setting and motivation can potentially be connected to many real-world applications.
>
> While the encoder models we employed are based on existing work, this is primarily for demonstration purposes and does not constitute a significant portion of the context. We want to emphasize that this level of prior-work reuse has been common in many prior works, including our baselines from [2]. We truly appreciate your advice in pointing out this issue and will ensure to clarify this in the revised version.
>
> **Q1**: We wish to first emphasize that “unlimited” does not mean infinite. According to [3], most real-world multimodal problems do not exceed four modalities. FuseMoE can scale to recently introduced high-modality problems, which combine modalities from different tasks. The scalability advantage is particularly prominent when compared with the pair-wise cross-attention-based approach.
>
> FuseMoE is designed to address real-life multimodal datasets with missing modality or irregularity issues, which can be caused by malfunctioning equipment, different patient conditions, human-level decisions, etc. In principle, the current method can be applied to arbitrary sparsity or modality combinations, as we have only tested on available real-world multimodal datasets. However, we did not manually create sparse data to meet the potential “extremely sparse” criteria. If such situations occur, we can employ additional strategies to enhance the robustness of the current framework, such as leveraging hierarchical MoE where a specific subgroup of experts is assigned to handle missing modalities, and applying regularization or dropout during training to make the model robust to missing modalities.
>
> Given that the model architecture remains the same, as the number of modalities N increases, the FuseMoE-based method scales linearly in $\mathcal{O}(N)$, whereas the cross-attention method scales in $\mathcal{O}(N^2)$.
>
> **Q2**: We have added additional results on ImageNet using the Vision-MoE framework for the Laplace gating, this can be found in the attached PDF file. In our paper, we have also tested the Laplace gating on MIMIC-III, MIMIC-IV, and PAM. Additionally, the results presented on the CMU-MOSI and CMU-MOSEI datasets utilize the Laplace gating function.
>
> **Q3**: Yes, please see Figure 12 (c) in Appendix H, which visualizes the modality contribution of the MIMIC dataset. Additionally, we have included the modality contributions of the CMU-MOSI and CMU-MOSEI datasets in the attached PDF file.
>
> **Q4**: There are three main differences between our FuseMoE and the Laplace MoE considered in the referenced papers:
>
> 1. Conditional Distributions:
> Under the FuseMoE model, the conditional distribution of $Y | X$, with the density given in Equation (4), where $X$ is an input and $Y$ is an output, is a mixture of Gaussian distributions. In contrast, the Laplace MoE model utilizes a mixture of Laplace distributions.
>
> 2. Sparse MoE versus Dense MoE:
> In FuseMoE, only a subset of Gaussian distributions is activated for each input $X$. Specifically, the conditional distribution of $Y | X$ is a mixture of K Gaussian distributions, where K is often set to one or two in practice. This mechanism is enabled by the Top K operator described below line 154. Meanwhile, in the Laplace MoE model, all Laplace distributions are activated for each input.
>
> 3. Gating Kernels:
> In FuseMoE, the mixture weights in Equation (4) use the Laplace kernel with the scale parameter set to one. In contrast, the gating kernel in the Laplace MoE model is an exponential value of a linear kernel.
>
> **References**
>
> [1] MULTIBENCH: Multiscale Benchmarks for Multimodal Representation Learning
>
> [2] Improving Medical Predictions by Irregular Multimodal Electronic Health Records Modeling
>
> [3] High-Modality Multimodal Transformer: Quantifying Modality \& Interaction Heterogeneity for High-Modality Representation Learning

---

> > ### Comment · Reviewer_6Ay7 · 2024-08-13
> > **Official Comment from reviewer  6Ay7**
> >
> > I thank the authors for the rebuttal and for taking the time to conduct additional experiments and comparisons.
> >
> > The rebuttal and the additional clarification address most of my concerns so I am happy to increase my score from 6 to 7.
> > Cheers!

---

> > > ### Author Response · Authors · 2024-08-13
> > > **Thank You!**
> > >
> > > Dear Reviewer `6Ay7`,
> > >
> > > Thank you so much for your positive response! Please feel free to reach out if you have any further questions or thoughts. We would be delighted to discuss them with you to improve our paper!
> > >
> > > Sincerely,
> > >
> > > Authors

---

### Official Review · Reviewer_P6x6 · 2024-07-14

**Soundness:** 3
**Presentation:** 3
**Contribution:** 2
**Rating:** 5
**Confidence:** 3

**Summary:**

This papers proposes a novel MoE arcchitecture for handling and fusing multiple modalities, along with two core contributions:
- a novel router design that can handle missing modalities;
- a laplace gating function that is theoretically proven to ensure better convergence.

**Strengths:**

- 1 This paper is generally well written and easy to follow; tables and figures are neat and informative;
- 2 The motivations of core designs are well elaborated;
- 3 Theoretical proofs are providied to further justify its effectiveness;
- 4 Extensive experiments, along with diverse modalities. are conducted to validate the effectiveness of the proposed approach.
- 5 The capabilty of handling missing inputs is interesting (Fig. 4 c).

**Weaknesses:**

- 1 The proposed method achieves promising results on the tested benchmark, which are still comparably small scale datasets, e.g, CMU-MOSI is with 2000+ samples. I am wondering if this work can be applicable to training/fine-tuning with larger scale data.
- 2 Again, for vision task, only CIFAR-10 is used. This cannot fully justify its effectiveness for more general vision or multi-modal tasks;
- 3 The capability in handling missing modalities is promising. The auhor is encouraged to explain why, in some scenarios, mising modalities + the proposed method surposses the variant with full modalities (Fig 4 c).  I am still wondering its generalizability under more scenarios.
- 4 The visualization and quanlitative analysis of the learned gaing weight would help the reader better understand the method.

**Questions:**

- 1 In table 5, Laplace gaiintg shows limited superiority against other variants, any insight for it?
- 2 In table 4c, why the method achieves better results with less modaltieis? Any further insight or ablations?

**Limitations:**

N

---

> ### Author Rebuttal · Authors · 2024-08-06
>
> Dear Reviewer `P6x6`,
>
> We deeply appreciate your insightful comments and positive feedback. We are heartened by your recognition that our paper is well-written, with well-elaborated motivations. We are also pleased that you acknowledge the theory and extensive experiments we provided to justify the effectiveness of our method. Below, we address your questions and concerns in detail.
>
> **Q: The proposed method achieves promising results on the tested benchmark, which are still comparably small scale datasets, e.g, CMU-MOSI is with 2000+ samples. I am wondering if this work can be applicable to training/fine-tuning with larger scale data.**
>
> **A:** Yes, we have tested FuseMoE on large-scale multimodal datasets including MIMIC-III, MIMIC-IV, and CMU-MOSEI. As shown in Table 1 of the MultiBench paper [1], these datasets are all featured as large-scale multimodal datasets, with MIMIC-III containing 36,212 samples, MIMIC-IV containing 73,173 samples, and CMU-MOSEI containing 22,777 samples. Even though CMU-MOSI is relatively small-scale with 2,199 samples, our experiments overall have tested a sufficient number of large-scale multimodal datasets. These datasets span various data types including text, images, time series, ECG, video frames, and audio signals, demonstrating strong generalization capacity across different domains.
>
> Please find tables summarizing the information of these datasets in the attached PDF file.
>
> **Q: Again, for vision task, only CIFAR-10 is used. This cannot fully justify its effectiveness for more general vision or multi-modal tasks**
>
> **A:** As explained above, several of the multimodal datasets we tested include vision components, such as chest X-rays in MIMIC and video frames in CMU-MOSI and MOSEI. Additionally, we have conducted further evaluation of the ImageNet dataset. Detailed information can be found in the attached PDF document.
>
> **Q: The capability to handle missing modalities is promising. The author is encouraged to explain why, in some scenarios, missing modalities + the proposed method surpasses the variant with full modalities (Fig 4 c). I am still wondering about its generalizability under more scenarios.**
>
> **A:** Full modalities mean that patient records with incomplete modalities will be discarded. In contrast, incorporating data with missing modalities allows us to access a broader array of samples, resulting in the utilization of all available patient data. As mentioned in Table 6 of Appendix B.1, for instance, when examining the 48-IHM or LOS tasks in the MIMIC dataset, the total number of patient samples is 35,129. However, the number of patients with clinical notes is only 32,038, and with chest X-ray (CXR) only 8,731. If we strictly enforce the inclusion of complete modalities involving CXR, the total number of samples available is constrained by this bottleneck.
>
> Therefore, being able to utilize incomplete modalities is a critical step in making use of the vast amount of patient data without full modalities. This approach will greatly improve generalization due to the increased number of available samples.
>
> **Q: The visualization and qualitative analysis of the learned gating weight would help the reader better understand the method.**
>
> **A:** We have included the visualization and qualitative analysis of the learned gating weights in Figure 12 of Appendix I. To better address your request, we have also added additional results on the CMU-MOSI and MOSEI datasets in the attached PDF file.
>
> **Q: In table 5, Laplace gating shows limited superiority against other variants, any insight for it?**
>
> **A:** In the results of Table 5, Laplace gating is still close to the best-performing method. We wish to emphasize that the advantage shown in Table 4 comes from the MoE architecture improvement. Most fusion methods listed in Table 4 are only suited to two-modality problems and cannot be extended as we scale to more modalities in Table 5. The comparison between Softmax, Gaussian, and Laplace gating functions in Table 5 reflects improvements within the same MoE architecture.
>
> These two types of improvements are distinct: sometimes one factor (e.g., the MoE architecture) is more important than the other (e.g., the gating function). Additionally, the performance advantage of each method also depends on the modality combination. Adding or removing modalities can influence the effectiveness of the Laplace gating mechanism.
>
> **Q: In table 4c, why the method achieves better results with less modalities? Any further insight or ablations?**
>
> **A:** Based on the context, we believe you mean Figure 4c instead of Table 4c. We would like to refer you to our answer to your first question for more details on this matter.
>
> Please let us know if there are specific aspects or sections you would like us to elaborate on further, or if you have any additional questions. Thank you!
>
> Sincerely,
>
> Authors
>
> **Reference**
>
> [1] Liang et al., (2021) MULTIBENCH: Multiscale Benchmarks for Multimodal Representation Learning, NeurIPS 2021.

---

### Author Rebuttal · Authors · 2024-08-07

Dear Area Chairs and Reviewers,

We want to thank you for your valuable feedback and insightful reviews, which have greatly contributed to improving our paper. The following endorsements are truly motivating:

- Writing: Our paper is well-written and easy to follow, with informative tables and figures (Reviewer `P6x6`).
- Motivation: The motivation of our paper is promising and well-elaborated (Reviewers `P6x6`, `6SUH`, `sZSN`).
- Method: FuseMoE is novel and performs better than baselines (Reviewer `6Ay7`).
- Theory: Our paper is theoretically sound (Reviewers `6Ay7`, `6SUH`, `sZSN`).
- Experiments: The experiments in our paper are extensive and comprehensive (Reviewers `P6x6`, `6Ay7`, `6SUH`, `sZSN`).

To better address the reviewers’ concerns, we have added additional results in the attached PDF file. We summarize the results below:
- **Table 1**: Summary of information on all the datasets we tested in the experiment section, including important categories such as modalities and sample sizes. These results complement our answers to Reviewers `P6x6`, `6Ay7`, and `6SUH`.
- **Figure 1**: ImageNet classification accuracy results using Vision-MoE-Base and Vision-MoE-Large models. For Vision-MoE-Base, we used 8 experts, 512 hidden dimensions, and 12 layers; for Vision-MoE-Large, we used 16 experts, 768 hidden dimensions, and 16 layers. The models are trained for 300 epochs. These results complement our answers to Reviewers `P6x6`, `6Ay7`, `6SUH`, and `sZSN`.
- **Figure 2**: Visualization of modality weight on the top-$k$ chosen experts using CMU-MOSI and CMU-MOSEI datasets. For every expert selected, we calculate the number of samples that include a specific modality, weighted by corresponding weight factors from the gating functions. The outcomes are subsequently normalized across modalities. The results for the MIMIC dataset can be found in Figure 12(c) of Appendix H.3. These results complement our answers to Reviewers `P6x6`, `6Ay7`, and `sZSN`.
- **Tables 2 - 5**: Experimental results on mutating different gating functions in conjunction with different encoders of FuseMoE. All results are averaged over 5 random runs. These results complement our answers to Reviewer `sZSN`.

We hope that our responses below and additional results will satisfactorily address your questions and concerns. We sincerely appreciate the time and effort you have dedicated to reviewing our submission, along with your invaluable suggestions.

Sincerely,

Authors

---

### Author Response · Authors · 2024-08-13
**Summary After Author-Reviewer Discussion**

Dear Area Chairs and Reviewers,

As the Author-Reviewer Discussion period comes to an end, we would like to provide a brief summary of our paper at this point for your reference. We have carefully reviewed all comments and have provided detailed responses to each. These will be comprehensively addressed in the final version of the paper. We extend our sincere gratitude to all reviewers for their insightful feedback and suggestions.

## **Motivation**:
We introduced FuseMoE, a novel MoE framework specifically designed to enhance multimodal fusion for inputs with arbitrary missingness or irregularity. Multiple reviewers (`P6x6`, `6SUH`, `sZSN`) have **agreed that the problem we are addressing is well-motivated and represents a promising direction for investigation**.

## **Method**:
FuseMoE incorporates sparsely gated MoE layers in its fusion component, which routes each modality to designated experts specializing in those specific data types while mitigating the influence of missing modalities. **This capability is critical to our method, as it allows for leveraging a vast amount of patient data without relying solely on complete modalities.** As we explained to Reviewer `P6x6`, most other methods that only leverage full modality data result in the exclusion of patient records with incomplete modalities. FuseMoE, by contrast, greatly improves generalization by accessing a broader array of samples.

Additionally, FuseMoE employs various router design mechanisms tailored for multimodal inputs with distinct combinations or interactions, giving users the flexibility to choose based on their specific applications.

## **Theory**:
FuseMoE integrates a novel Laplace gating function, which is particularly well-suited for multimodal input data, especially in large-scale experiments, compared to the commonly used Softmax gating function. Theoretically, it ensures better convergence rates than Softmax, as discussed in detail with high-level intuitive explanations in our rebuttal to Reviewer `6SUH`. Additionally, we have illustrated the superiority of our gating function over the Gaussian gating function in our response to Reviewer `sZSN`.

## **Experiments**:
We demonstrate that FuseMoE significantly outperforms existing well-known and representative methodologies within specific categories, in integrating diverse input modality types:
- `Text`: MIMIC-III, MIMIC-IV, CMU-MOSI, CMU-MOSEI
- `Images`: MIMIC-IV, CIFAR-10, ImageNet (new)
- `Time series`: MIMIC-III, MIMIC-IV, PAM
- `ECG waveform`: MIMIC-IV
- `Video frames`: CMU-MOSI, CMU-MOSEI
- `Audio signals`: CMU-MOSI, CMU-MOSEI

with varying missingness and irregular sampling on challenging tasks:

- `Mortality prediction`: MIMIC-III, MIMIC-IV
- `Length-of-stay`: MIMIC-III, MIMIC-IV
- `Phenotyping`: MIMIC-IV
- `Sentiment analysis`: CMU-MOSI, CMU-MOSEI
- `Activity recognition`: PAM
- `Image classification`: CIFAR-10, ImageNet (new)

**The majority of the multimodal datasets we tested are considered large-scale**, including MIMIC-III with 36,212 samples, MIMIC-IV with 73,173 samples, and CMU-MOSEI with 22,777 samples. These datasets cover a sufficient number of benchmarks and research areas, particularly in a context where benchmarks are not abundant.

**We also conducted comprehensive ablation studies and visualizations of FuseMoE** to assess the effectiveness of each building block and enhance the interpretability of our proposed method. These studies revealed that our model architecture consistently outperforms other alternatives and insights into how much each input modality contributes to our tasks of interest.

## **Reproducibility**:
We have submitted (1) implementations of FuseMoE and its baselines, (2) scripts to process and generate multimodal input data as part of our supplementary material.

Thank you again for your efforts in reviewing this submission. We appreciate all the valuable feedback, which has significantly contributed to improving our manuscript. **We look forward to continuing the discussion with any reviewers who still have questions during the remaining discussion period.**

Sincerely,

Authors

---

### Decision · Program_Chairs · 2024-09-25

**Decision:**

Accept (poster)

**Comment:**

This paper presents work on multi-modal data, particularly including time series data.  A mixture of experts framework is developed, with a Laplace gating function.  A variety of fusion/gating strategies is explored as examples of the general MoE / gating approach.

The reviewers appreciated the problem formulation and contribution of the Laplace gating function with its accompanying theoretical analysis.  Concerns were raised over the computation needs, scalability, and effectiveness of the routing/gating mechnanism versus the modality-specific encoders.

After considering the authors' response the reviewers evaluated their positions.  The additional notes on experiments, comparisons, and clarificiations resolved most of the reviewers' concerns.

Overall, the paper presents interesting work on learning from multi-modal data with a solid contribution in the form of a novel gating function and promising results.  As such, it is recommended for acceptance in NeurIPS.